



# A Microphysics Guide to Cirrus – Part II: Climatologies of Clouds and Humidity from Observations

Martina Krämer[1,2], Christian Rolf[1], Nicole Spelten[1], Armin Afchine[1], David Fahey[3], Eric Jensen[4], Sergey Khaykin[5], Thomas Kuhn[6], Paul Lawson[7], Alexey Lykov[8], Laura L. Pan[4], Martin Riese[1], Andrew Rollins[3], Fred Stroh[1], Troy Thornberry[4], Veronika Wolf[6], Sarah Woods[7], Peter Spichtinger[2], Johannes Quaas[9], and Odran Sourdeval[10]

[1]Institute for Energy and Climate Research (IEK-7), Research Center Jülich, Jülich, Germany
[2]Institute for Atmospheric Physics (IPA), Johannes Gutenberg University, Mainz, Germany
[3]NOAA ESRL CSD, Boulder, USA;
[4]NCAR, Atmospheric Chemistry Observations and Modeling Laboratory, Boulder, USA;
[5]LATMOS/IPSL, UVSQ, Sorbonne Université, CNRS, Guyancourt, France
[6]Luleå University of Technology, Division of Space Technology, Kiruna, Sweden
[7]SPEC Inc., Boulder, CO, USA;
[8]Central Aerological Observatory (CAO), Department of Upper Atmospheric Layers Physics, Moscow, Russia
[9]Leipzig Institute for Meteorology (LIM), Universität Leipzig, Leipzig, Germany
[10]Univ. Lille, CNRS, UMR 8518 - LOA - Laboratoire d'Optique Atmosphérique, F-59000 Lille, France

*Correspondence to:* M. Krämer (m.kraemer@fz-juelich.de)





**Abstract.** This study presents airborne in-situ and satellite remote sensing climatologies of cirrus clouds and humidity. The climatologies serve as a guide to the properties of cirrus clouds, with the new in-situ data base providing detailed insights into boreal mid-latitudes and the tropics, while the satellite-borne data set offers a global overview.

To this end, an extensive, quality checked data archive, the Cirrus Guide II in-situ data base, is created from airborne in-situ measurements during 150 flights in 24 campaigns. The archive contains meteorological parameters, IWC, $N_{ice}$, $R_{ice}$, $RH_{ice}$ and $H_2O$ for each of the flights (IWC: ice water content, $N_{ice}$: number concentration of ice crystals, $R_{ice}$: ice crystal mean mass radius, $RH_{ice}$: relative humidity with respect to ice, $H_2O$: water vapor mixing ratio). Depending on the specific parameter,

the data base has extended by about a factor of 5-10 compared to earlier studies. One result of our investigations is, that across all latitudes, the thicker liquid origin cirrus predominate at lower altitudes, while at higher altitudes the thinner in-situ cirrus prevail. Further, exemplary investigations of the radiative characteristics of in-situ and liquid origin cirrus show that the in-situ origin cirrus only slightly warm the atmosphere, while liquid origin cirrus have a strong cooling effect.

An important step in completing the Cirrus Guide II is the provision of the global cirrus $N_{ice}$ climatology, derived by means of the retrieval algorithm DARDAR-Nice from ten years of cirrus remote sensing observations from satellite. The in-situ data base has been used to evaluate and adjust the satellite observations. We found that the global median $N_{ice}$ from satellite observations is almost two times higher than the in-situ median and increases slightly with decreasing temperature. $N_{ice}$ me-

dians of the most frequentl occuring cirrus sorted by geographical regions are highest in the tropics, followed by austral/boreal mid-latitudes, Antarctica and the Arctic. Since the satellite climatologies enclose the entire spatial and temporal $N_{ice}$ occurrence, we could deduce that half of the cirrus are located in the lowest, warmest cirrus layer and contain a significant amount of liquid origin cirrus.

     A specific highlight of the study is the in-situ observations of tropical tropopause layer (TTL)

cirrus and humidity in the Asian monsoon anticyclone and the comparison to the surrounding tropics. In the convectively very active Asian monsoon, peak values of $N_{ice}$ and IWC of 30 ppmv and 1000 ppmv are detected around the cold point tropopause (CPT). Above the CPT, ice particles that are convectively injected can locally add a significant amount of water available for exchange with the stratosphere. We found IWCs of up to 8 ppmv in the Asian monsoon in comparison to only 2 ppmv

in the surrounding tropics. Also, the highest $RH_{ice}$ inside of the clouds as well as in clear sky (120-150%) are observed around and above the CPT. We attribute this to the high amount of $H_2O$ (3-5 ppmv) in comparison to 1.5-3 ppmv in other tropical regions. The supersaturations above the CPT suggest that the water exchange with the stratosphere is 10-20% higher than expected in regions of weak convective activity and up to about 50% in the Asian monsoon.





## 1 Introduction

In Part 1 of the study (Krämer et al., 2016), a detailed guide to cirrus cloud formation and evolution is provided, compiled from extensive model simulations covering the broad range of atmospheric conditions and portrayed in the same way as field measurements in the Ice Water Content-Temperature (IWC-T) parameter space. The study was motivated by the continuing lack of understanding of the microphysical and radiative properties of cirrus clouds, which remains one of the greatest uncertainties in predicting the Earth's climate (IPCC, 2013). An important result is the classification of two types of cirrus clouds that differ by formation mechanisms and microphysical properties: rather thin in-situ origin cirrus that form on-site below -38°C, and predominantly thick cirrus originating from liquid clouds that are uplifted from warmer layers farther below.

Since then, a number of studies has been published that shed further light on the exploration of the high ice clouds. For example, some new studies, mostly based on aircraft or lidar observations, provide overviews and climatologies of cirrus cloud properties (Kienast-Sjögren et al., 2016; Petzold et al., 2017; Heymsfield et al., 2017a, b; Woods et al., 2018; Lawson et al., 2019) while others present a more specific view (Urbanek et al., 2017, 2018). Overviews of cirrus' properties from global satellite remote sensing observations were also recently enhanced to include ice crystal number concentrations (Sourdeval et al., 2018; Gryspeerdt et al., 2018; Mitchell et al., 2018). Several studies make use of the concept of in-situ and liquid origin cirrus, e.g. Wernli et al. (2016), investigating the occurrence of in-situ and liquid origin cirrus over the North Atlantic by analyzing ERA-interim data, Gasparini and Lohmann (2016) simulating, amongst other things, the global distribution of liquid origin cirrus (they name them 'detrained ice'). Wolf et al. (2018) studied the microphysical properties of Arctic in-situ and liquid origin cirrus from balloon-borne observations, and Wolf et al. (2019) provide a cirrus parametrization demonstrating the dependence on the origin of the clouds.

The wealth of earlier (see e.g. references in Krämer et al., 2016) and new studies provide more and more insights into formation processes, life cycles and appearance of cirrus. Nevertheless, there are still gaps that need to be filled, on the one hand in the ice process understanding and on the other hand to improve the representation of cirrus clouds in climate prediction. A way to accomplish this task requires large and high quality observational data bases that can serve, for example, to evaluate global models or other data sets and to derive parametrizations of cirrus clouds reliably representing the different types of cirrus (see e.g. Wolf et al., 2019). In addition, such data bases allow detailed studies of special types of cirrus that are still poorly understood, e.g. cirrus in fast updrafts as orographic cirrus or cirrus in strong convection.

In this study, we approach these requirements as follows: we first compile a data archive of airborne in-situ observations which is extended with respect to earlier versions (Schiller et al., 2008; Krämer et al., 2009; Luebke et al., 2013; Krämer et al., 2016) in terms of the size of the data set





that contains all parameters needed for the desired studies, i.e. meteorological parameters, ice water content (IWC), number concentration of ice crystals ($N_{ice}$), ice crystal mean mass radius ($R_{ice}$[1] ), relative humidity with respect to ice ($RH_{ice}$) and water vapor mixing ratio ($H_2O$). However, though
airborne in-situ measurements best represent detailed microphysical properties of cirrus and their environment, they are always snapshots of specific situations that are also limited by the possibilities of the flight patterns and thus not suitable to derive spatial geographical or seasonal views of cirrus clouds. For this purpose, a globally complete dataset of remote sensing observations from satellite observations are the better option. Hence, as next step of the study we use in-situ climatologies to
evaluate cirrus $N_{ice}$ from satellite observations and, based on this, derive a global climatology of cirrus $N_{ice}$. From the portrayal of the two Cirrus Guide II data sets emerging from this study together with some more detailed analyses, we show that the combined evaluation of airborne in-situ and satellite remote sensing observations enhances the insights in cirrus properties. The in-situ observations are best suitable for the investigation of specific, smaller scale phenomena and for the
evaluation of satellite observations or model simulations. Satellite-borne observations, on the other hand, allow a view of the larger scale and seasonal properties.

The article is structured as follows: The Cirrus Guide II in-situ data bases and the used methods are described in  Section 2 and Appendix A. As an overview, in  Section 3 we portray the in-situ
cirrus cloud and humidity data base with respect to altitude for the latitudes covered by the observations. The usefulness of the data set is shown by discussing the characteristics and occurrences of in-situ and liquid origin cirrus.  Section 4 first presents cirrus and humidity climatologies of the extended Cirrus Guide II in-situ data base in terms of temperature in comparison to the earlier studies mentioned above. Further, characteristic properties of mid-latitude and tropical climatologies are
presented.

In  Section 5 we show another example of a specific analysis extracted from the Cirrus Guide II in-situ data base: the data set includes recent unique measurements in the tropical tropopause layer (TTL) region of the Asian monsoon anticyclone, where cirrus clouds and humidity are of special interest and observations are rare. The topic is briefly introduced before the special observations
in the Asian monsoon are presented and compared with the conditions found in the surrounding tropical regions.

The last part of the study (Section 6) is the step to global climatology of cirrus $N_{ice}$ from satellite remote sensing observations. For this purpose, the Cirrus Guide II in-situ data base is used to evaluate remote sensing cirrus observations. Based on this, a global $N_{ice}$ climatology is derived and first
analyses of the global and also regional $N_{ice}$ are presented.

---

[1] mean mass radius $R_{ice} = \left( \frac{3 \cdot IWC}{4\pi\rho \cdot N_{ice}} \right)^{1/3}$ with $\rho$= 0.92 g/cm$^3$





## 2 Data bases

### 2.1 In-situ data set

The observations presented here include the ice water content IWC, the ice crystal number concentration $N_{ice}$, the mean mass radius $R_{ice}$ in-cloud as well as the clear sky $RH_{ice}$ and clear sky water vapor volume mixing ratio $H_2O$. The complete in-situ data set comprises 24 field campaigns: the 17 experiments shown in Part I of this study (Krämer et al., 2016, ; the campaigns were performed between 1999 and 2014 over Europe, Africa, Seychelles, Brazil, Australia, USA and Costa Rica), extended by the field campaigns SPARTICUS 2010 and START 2008 over Central US, LTU 2012-2018 over Kiruna, CONTRAST and ATTREX in 2014 and POSIDON 2016 over the tropical Pacific as well as StratoClim 2017 out of Nepal.

A map of flights during the various campaigns (extended map of Cirrus Guide: Part I) is shown in Figure 1. In the Appendix A, a summary of the field campaigns and deployed instrumentation is given in Table 4; Table 5 lists all campaigns and the measured parameters. Also, a discussion of new data evaluation methods, data quality and data coverage is presented. An overview of each campaign is given in the Supplementary Material (SM in the following). Twenty campaigns are chosen to be included in the climatologies; four campaigns (marked in Table 5), where the data volume is very low (START 2008, LTU 2012-2018) or very massive (SPARTICUS in 2010, CONTRAST 2014), so that their contribution to frequency occurrences is either negligible or dominant, are shown in the overview of measurements only (SM).

The climatologies are advanced in several aspects in comparison to the compilations of IWC by Schiller et al. (2008), Luebke et al. (2013) and Krämer et al. (2016) and $N_{ice}$, $R_{ice}$ and $RH_{ice}$ by Krämer et al. (2009):

- The number of flights and total time in cirrus increased from 104 flights / 94 hours (Krämer et al., 2016) to a total of 150 flights / 168 hours, i.e. the data base disporportionally has extended by about a factor of 5-10 depending on the specific parameter.

- For IWC, a new data product has been developed that increases the observed data volume (Appendix A2.1).

- For $N_{ice}$, observations from advanced and extended instrumentation could be added to the data base; further, a new correction of the occurrence frequencies is applied (Appendix A2.2).

- The geographical spread of the observations has broadened, so that a portrayal of cirrus and humidity with respect to the geographical regions mid-latitude and tropics, and also latitude/altitude seems worthwhile.

- As in the earlier climatologies, all data underwent strict quality control.





## 2.2 Satellite data set

DARDAR-Nice provides observational-based estimates of $N_{ice}$ obtained from CALIPSO and Cloud-Sat measurements (Sourdeval et al., 2018). This unique approach uses the sensitivity of lidar and radar measurements to small and large particles, respectively, to constrain two parameters of a particle size distribution (PSD) parametrization and $N_{ice}$ is subsequently estimated by direct integration, from a minimum threshold size set in this study to 5 μm. DARDAR-Nice uses the parametrization

by (Delanoë et al., 2005), in which two normalization parameters (a slope parameter $N_0^\star$ and the volume-weighted diameter $D_m$) are used to predict the shape of a PSD. Sourdeval et al. (2018) have demonstrated that the method of Delanoë et al. (2005) is theoretically capable of predicting $N_{ice}$ from recent in-situ campaigns, by comparing its prediction based on in-situ $N_0^\star$ and $D_m$ measurements to the actual in-situ $N_{ice}$ measurement. Good agreements were found, except for temperatures higher

than about -50°C where an overestimation in DARDAR-Nice due to the inability of the modified gamma distribution to match the frequently bi-modal shape of the measured PSDs.

This evaluation framework is here repeated on the basis of 5 in-situ campaigns archived in the Cirrus Guide II in-situ data base: COALESC2011, ACRIDICON2014, ATTREX2014, MLCIR-RUS2014, STRATOCLIM2017. Based on the agreement between DARDAR-Nice and the in-situ

observations, a global $N_{ice}$ climatology is derived from 10 years of satellite observations. Regional $N_{ice}$ climatologies for the Arctic (90N - 67.7N), Northern mid-latitudes (67.7N-23.3N), Tropics (23.3N - 23.3S), Southern mid-latitudes (23.3S - 67.7S) and Antarctica (67.7S - 90S) are analysed in more detail. The results are presented in Section 6.

## 3 Vertical distribution of cirrus and humidity

As an introduction, atmospheric temperature profiles in the Arctic, at mid-latitudes and in the tropic are shown in Figure 2, (left panel, adopted from Schiller et al., 2008). Exemplarily, vertical profiles inside of cirrus clouds are shown for 28 flights, using blueish colors for Arctic, greenish for mid-latitude and reddish for tropical observations. It can be seen that in the tropics, with the strongest warming of the Earth's surface by the sun, the air is much warmer at higher altitudes than at mid-

latitudes or in the Arctic. The coldest atmospheric temperatures are found at the points where the slopes of the temperature profile reverses: the cold point tropopause (CPT). This is the region where the transition from the troposphere to stratosphere occurs. Above the CPT, in the stratosphere, almost no cirrus clouds are observed because it is too dry for ice formation. This can be seen in the right panel of Figure 2, (adapted from Schiller et al., 2009, with annotations), where total $H_2O$ (= water

vapor + evaporated ice crystals) is plotted versus the potential temperature $\Theta$[2] for the tropical field campaign TROOCINOX (Brazil, 2005), color coded by total $H_2O$ expressed as relative humidity

---

[2] $\Theta$ is often used in upper troposphere–lower stratosphere (UT/LS) research, since it allows a clear assignment of air parcels to the associated atmospheric layer, in contrast to the temperature, whose course reverses above the CPT. The relations





over ice ($RH_{ice}$). Data points right of the clear sky $H_2O$ represent a few overshooting convective tropical cirrus clouds (see Section 5), where the total water is enhanced by evaporating ice crystals. In clear sky conditions above the CPT, $H_2O$ is very low and the air is correspondingly dry. Below the
CPT, $H_2O$ together with humidity are rapidly increasing with decreasing $\Theta$.

### 3.1 Latitude - altitude distributions

As first portrayal of the Cirrus Guide II data set the distribution of cirrus clouds and humidity is shown with respect to latitude and altitude in Figure 3. Plotted are IWC (color coded by volume mixing ratio), $N_{ice}$ (color coded by concentration), $R_{ice}$ (color coded by size), in-cloud as well as the
clear sky humidity (color coded by $RH_{ice}$) and $H_2O$ (color coded by volume mixing ratio). The data were collected in the latitude range from about 70°North to around 20°South, i.e. the northern mid-latitudes and the tropics are covered by the observations. The altitude range is between about 5 to 20km. The times of data sampling are displayed in the respective panels; a more detailed description of the data base is given in Section 4.

Cirrus clouds are found at lower altitudes at mid-latitudes and reach higher levels in the tropical region (Figure 2). Though this structure in the measurements is influenced by the maximum or minimum height the engaged aircraft can reach, it corresponds well with the CALIPSO latitudinal height distribution cirrus clouds, which is largely caused by the decrease in tropopause height with increasing latitude (shown in the review article by Heymsfield et al., 2017a, original figure by Sassen
et al. (2008)).

#### 3.1.1 In-situ and liquid origin cirrus

Krämer et al. (2016), Luebke et al. (2016) and Wernli et al. (2016) describe two different cirrus types: (1) in-situ origin cirrus that form (by heterogeneous or homogeneous ice nucleation on ice nucleating particles, INP, or soluble solution aerosol particles, respectively) from water vapor di-
rectly as ice at $T < 235K$, $RH_{ice} > 100 \%$ and $RH_w < 100 \%$; (2) liquid origin cirrus that evolve (also heterogeneously or homogeneously) from freezing of liquid drops in clouds at $T \gtrsim 235K$ and $RH_w \sim 100 \%$ [3] . In other words, in-situ origin cirrus are observed at the altitudes where they are formed, whereas liquid origin cirrus are glaciated liquid clouds from further below which are lifted
to the cirrus temperature region where liquid water no longer exist.

*Microphysical characteristics:* In the following, the typical characteristics of the cirrus types are briefly described and summarized in tabular form in Table 1.

---

between $\Theta$ and temperature as well as altitude and $\Theta$ are shown in Figure 2, middle left and middle right panel for the different geographical regions.

[3]For a more detailed description of the freezing mechanisms see Vali et al. (2015); Heymsfield et al. (2017a) and references therein.



IN-SITU ORIGIN cirrus split in two sub-classes, depending on the strength of the updraft: in *slow updrafts*, in-situ cirrus form mostly heterogeneously and are rather thin with lower IWCs and few but large ice crystals. We like to note here that in an atmosphere free of INPs, cirrus clouds appearing homogeneously would have similar characteristics, since also only few ice crystals are nucleated homogeneously at slow updrafts (see e.g. Spreitzer et al., 2017). Hence, in this regime, the freezing process is not relevant for the cirrus properties. In *fast updrafts*, homogeneous freezing is mostly triggered regardless of the presence of INPs, since fast updrafts cause $RH_{ice}$ to reach the homogeneous freezing threshold even after heterogenous freezing. The in-situ cirrus that emerge in these situations are thicker with higher IWCs and more but smaller ice crystals.

LIQUID ORIGIN cirrus stem from lower altitudes, where more water is available, so they generally consist predominantly of thicker cirrus with higher IWC, together with more and larger ice crystals that are frozen heterogeneously at T > 235K in *slower updrafts*. In *fast updrafts*, liquid origin cirrus with very high $N_{ice}$ appear. The reason is that in such updrafts $RH_w$ and $RH_{ice}$ are both > 100 % and thus the Wegener-Bergeron-Findeisen process, where liquid drops evaporate and ice crystals remain at $RH_w$ < 100 % and $RH_{ice}$ > 100 %, is suppressed (Korolev, 2007). Thus, the numerous supercooled drops reach the altitude (temperature $\sim$ 235K) where they freeze homogeneously (Costa et al., 2017).

The meteorological situations where slow updraft in-situ cirrus frequently occur (see Krämer et al., 2016) are low- and high-pressure systems (frontal/synoptic cirrus). The warm conveyor belt in low pressure systems can also produce liquid origin cirrus. Fast updraft in-situ and liquid origin cirrus occur in -often orographically induced- gravity waves, in jet streams and in convective systems and anvils.

As outlined in the Cirrus Guide I (Krämer et al., 2016), to a certain extent cirrus types can be identified by their typical characteristics. This applies to the initial stage, after which the clouds lose the signature of the formation process e.g. by sedimentation: as long as an updraft prevails (corresponding $RH_{ice}$> 100%), smaller ice crystals in the size range $\lesssim$ 20μm grow to larger sizes on a time scale of tens of minutes. The larger ice crystals sediment to lower altitudes, thus removing ice surface from the cloud volume which consequently reduces the depletion of $H_2O_{gas}$ ($\propto RH_{ice}$) by water vapor transport to the ice (for more detail see Spichtinger and Cziczo, 2010). The ice crystals that have fallen out of the layer deepen the cirrus extent to lower altitudes (fall streaks can extend the cirrus to several km below the nucleation level Jensen et al., 2012; Murphy, 2014), while at the same time, large ice crystals from above could sediment into the cloud volume. Altogether, the cirrus evolution is a dynamical process and the cirrus properties change in the course of a cirrus lifetime. At the final cirrus stage, i.e. when the temperature increases, the environment becomes subsaturated ($RH_{ice}$< 100%) and ice crystals evaporate, disappearing the faster the smaller they are (timescales of growth and evaporation of ice crystals are shown in Kübbeler et al., 2011, their Figure 12).



Cirrus types with the most striking features (high IWC / $N_{ice}$) are the easiest to identify. They could be liquid or in-situ origin cirrus in fast updrafts or also liquid origin cirrus in slow updrafts. This will be shown on the basis of Figure 3 and Figure 4.

In Figure 4, the relation between $N_{ice}$, $R_{ice}$ and IWC is shown for ∼87 h of cirrus cloud observations. It can be nicely seen that in case $N_{ice}$ is high, the ice crystals are small – because the numerous

ice crystals consumed all of the available vapor, thereby suppressing further growth – while low $N_{ice}$ are related to large ice crystals for a given IWC. The IWC (in mg/m$^3$) can be recognized by the color code and result from the combination of both $N_{ice}$ and $R_{ice}$. The thin black lines are isolines of IWC (in ppmv) that appear in the order shown the legend. The scheme at the right side illustrates the partitioning of the clouds between liquid and in-situ origin: the thickest cirrus (blue points) are of

liquid origin, the thinnest (yellow points) of in-situ origin. As the thickness of the cirrus decreases, the portion of liquid origin cirrus becomes smaller and smaller while more and more in-situ origin cirrus appear.

So most of the highest IWCs ($> 10$ mg/m$^3$, dark red and blue diamonds in Figure 3, top left panel) are of liquid origin. They appear in the lower parts of the clouds, i.e. they are uplifted from farther

below. A good example of this is the field campaign SPARTICUS in 2010, which is separately plotted in Figure 6 of the SM (right panel). About 23 hours of sampling in mostly liquid origin clouds were performed over Central USA. The clouds are recorded in the temperature range 210-240 K, which corresponds to altitudes between 5-10 km, i.e. they are rather low cirrus clouds. The IWCs are mostly high (red to blue colors) and, as shown by Muhlbauer et al. (2014), ice particles up to thousand μm

diameter and more are present in most measurements, which is indicative for liquid origin cirrus (see Table 1). In the tropics (Figure 3), some blue points are also detected at high altitudes of about 17 km. They are observed above the Asian monsoon in strong convective updrafts (see also Section 5) and consist of many ice crystals (0.1-10 cm$^{-3}$) with medium $R_{ice}$ (20-70 μm).

Generally, the IWC roughly shows a vertical structure of decreasing IWC with increasing altitude.

This is caused on the one hand by the amount of available water that decreases with decreasing temperature, but also because cirrus of liquid origin predominate in lower layers, whereas cirrus with in-situ origin become more abundant at higher altitudes, i.e. colder temperatures. This is in accordance with the findings of Luebke et al. (2016); Wernli et al. (2016) and Wolf et al. (2018), where Luebke et al. (2016) and Wolf et al. (2018) experimentally investigated the two cloud types

in mid-latitudes and in the Arctic, respectively, while Wernli et al. (2016) analyzed 12 years ERA-interim data in the North Atlantic region (see also Section 6.2).

As the IWC is an indication for the cirrus type, $N_{ice}$ (Figure 3, middle left panel) can give hints on the freezing mechanism, either heterogeneous or homogeneous: high ice crystal numbers $N_{ice}$ ($\gtrsim 0.5$ cm$^{-3}$, dark red and blue diamonds, middle left panel of Figure 3) are an indicator for homogeneous

ice formation in fast updrafts caused by waves or convection, both for in-situ as well as for liquid origin cirrus. At mid-latitudes, they are found e.g. at the tops of mountain wave clouds (the ice



nucleation zone), which were observed for example behind the Norwegian mountains at around 62°North. High $N_{ice}$ values have also been observed in tropical deep convection around 10°North, corresponding to measurements reported by Jensen et al. (2009).

Another source of mid-latitude high $N_{ice}$ are young contrails. Note however, that these high $N_{ice}$ are transient, that means they do not exist for a long time. This is because high ice crystal numbers are associated with small ice crystal sizes (blue diamonds in the bottom left panel, see also Figure 4) that grow/evaporate quickly. At lower altitudes, $N_{ice}$ tends to be lower and the crystals are larger, because large crystals having lower concentrations sediment out from the cloud tops and, as mentioned above,

liquid origin cirrus with characteristic large ice crystals are also common in this altitude range. In the tropics, high $N_{ice}$ at high altitudes are induced by in-situ homogeneous freezing in convection or gravity waves (see alo Jensen et al., 2013a). At cloud bases, such high $N_{ice}$ are most probably of liquid origin, initiated by homogeneous freezing of supercooled drops.

A more detailed discussion of the microphysical properties of cirrus including frequencies of oc-

currence of specific signatures which can not be seen from Figure 3, is given in Section 4.

*Radiative characteristics:* A motivation to study cirrus clouds is to investigate the radiation properties on the basis of the findings on their microphysical properties. In the Cirrus Guide I, Krämer et al. (2016) speculated that the physically and optically thinner in situ slow updraft cirrus cause a

warming effect, while the thicker in situ fast updraft and particularly the thick liquid origin cirrus have the potential to cool. Here, we show a first estimate of the radiative forcing of typical in-situ slow and fast updraft as well as liquid origin cirrus (Figure 5). To this end, radiative transfer calculations for idealized scenarios are realized, at noon at the equinox, which are briefly described in the caption of Figure 5.

In the left panel of the Figure, the radiative forcing of the slow (light green) and fast (dark green) in-situ origin cirrus is displayed with respect to the optical depth. This panel is zoomed into the right panel, where the forcing of the liquid origin cirrus is shown. Obviously, the net radiative forcing of the in-situ origin cirrus is much smaller than that of liquid origin and, moreover, change the sign from warming to cooling.

In more detail, the slow in-situ origin cirrus have only small optical depth ($\tau$) between 0.001 - 0.05, resulting in a slight net warming effect of not larger than about 1.5 W/m$^2$. The optical depth of fast in-situ origin cirrus is larger ($\tau$: 0.05 - 1), but most of them are also warming (2-10 W/m$^2$). However, the thickest fast in-situ origin cirrus at the lowest altitudes change the sign of their net forcing, they switch to a slight cooling effect. The reason is the warmer temperature at lower altitude

that reduces the warming effect of the longwave infrared radiation. The liquid origin cirrus, mostly found in the warmest cirrus layers, have large optical depths ($\tau$: 1 - 12), which is larger than the range of cirrus optical depth reported in many studies (the maximum optical depth is often assumed as 3, e.g. Sassen et al., 2008; Mitchell et al., 2018, ; note that this is likely because the lidar technique, often use to investiagte cirrus cloud optical properties, has restrictions in the detection of thicker





ice clouds). A consequence of the large optical thickness is a quite strong net cooling effect (- 15 to -250 W/m$^2$) of liquid origin cirrus. These values are of the same order of magnitude as reported from direct measurements inside of cirrus clouds (Wendisch et al., 2007; Joos , 2019).

Thus, from this first and very idealized simulations, we can conclude that in-situ formed cirrus clouds are most likely warm the atmosphere, whereas liquid origin ice clouds have the potential for

strong cooling. Note here, that we only investigate local time 12 h, where the cooling is probably most pronounced. For lower sun position (i.e. larger zenith angle) the cooling is probably reduced and during night time, cirrus clouds can only warm the atmosphere (due to the thermal greenhouse effect). Thus, the net effect of cirrus clouds averaged over the whole daily cycle is not yet clear. Such investigations go beyond the scope of this study and are subject of future work.

### 325 3.1.2 Humidity

The distribution of in-cloud and clear sky $RH_{ice}$ as well as water vapor $H_2O$ with latitude and altitude is shown in the right column of Figure 3. It can nicely be seen how the amount of $H_2O$ decreases with altitude (bottom panel). The clear sky $RH_{ice}$ (middle panel) ranges from very dry conditions ($<$ 70% green and orange diamonds) up to highly supersaturated regions ($>$ 130%, dark red and

blue diamonds), which mainly are found at high altitudes in the tropics. Such high supersaturations are also found inside of the tropical cirrus clouds (top panel). The behavior of $RH_{ice}$ will be further discussed in Sections 4 and 5.

## 4 In-situ climatologies

The Cirrus Guide II data set shown in Figure 3 is now displayed as frequencies of occurrence in

dependence on temperature (binned in 1 K intervals, Figure 6) and discussed in comparison to the earlier in-situ climatologies presented by Schiller et al. (2008), Krämer et al. (2009) and Luebke et al. (2013) (Figure 7). Further, as mentioned above, the new data set is large enough to split it in mid-latitude and tropical cirrus. The differing cirrus cloud properties are presented in Figures 8 and the respective clear sky and in-cloud $RH_{ice}$ in Figures 9.

### 340 4.1 Entire in-situ climatologies

#### 4.1.1 Ice water content (IWC-T)

Figure 6 (top left panel) depicts the IWC. The black solid and dotted lines represent the median, minimum and maximum IWC of the core IWC band, that is the envelope of the most frequent IWC ($>$5% per IWC-T bin, Schiller et al., 2008).

The number of hours spend in cirrus clouds raised from 27 hours in Schiller et al. (2008), 38 hours in Luebke et al. (2013) and 94 hours in (Krämer et al., 2016) to 168 hours in the new extended data set. Part of the additional data is due to the new IWC data product that is applied to all campaigns and





combines the IWC from total water measurements as well as the IWC derived from cloud particle
size distributions (see Appendix A2.1).

However, the median IWC and the core IWC band -decreasing with temperature as described by
Schiller et al. (2008)- is still valid, showing that the IWC measurement techniques are robust and
that the IWC is a stable parameter describing cirrus clouds. Note here that below about 200 K data
points below the lower dotted line are not unambiguously identified as clouds, while above about
200 K this threshold is 0.05 ppmv. For more detail see also Appendix A2.1.

### 4.1.2  Ice crystal number ($N_{ice}$-T)

About 90 hours of $N_{ice}$ observations are shown in  Figure 6 (middle left panel [4] ), which is an
increase of about a factor of ten in comparison to the data set of Krämer et al. (2009), who complied
8.5 hours (Figure 7). For $N_{ice}$, the picture has greatly changed when comparing the old and the new
data sets. This is on the one hand due to an extension of the lower detection limit of $N_{ice}$ from

360    $4 \cdot 10^{-3}$ to $10^{-4}$ $cm^{-3}$ (see Appendix A2.2), but also because the new data set represents a better
mixture of different meteorological situations. For example, the higher $N_{ice}$ at warmer temperatures
in the Krämer et al. (2009) data set (Figure 7) were caused by flights where lee wave cirrus behind
the Norwegian mountains were probed (see also  Figure 3, blue diamonds at around 60°North).
Also, at temperatures colder than about 200 K, $N_{ice}$ was most often very low. Further, the enhanced

365    occurrence frequencies at the lowest concentrations seen in the earlier data set are corrected in the
new data evaluation procedures (Figures 17 and 7, middle left panel).

90 hours of aircraft $N_{ice}$ observations within of cirrus clouds is a tremendous amount when taking
into account the necessary effort. However, this is still far from being representative for the distribu-
tion of $N_{ice}$ in the atmosphere. We nevertheless calculated 10, 25, 50, 75 and 90% percentiles, which

are shown as thin, dotted and solid lines in  Figure 6 (middle left panel). Note that the 10 and 90%
percentiles enclose the core region of $N_{ice}$, i.e. the envelope of the most frequent $N_{ice}$ (>5% per $N_{ice}$-
T bin). Fits through these percentiles and the median $N_{ice}$ reveal no temperature dependence of $N_{ice}$
(10%, median and 90% $N_{ice}$: 0.002, 0.03 and 0.3 $cm^{-3}$). This is different to the slight decrease with
temperature of the minimum, middle and maximum $N_{ice}$ lines shown by Krämer et al. (2009), which

was, as discussed above, cause by two flights with high $N_{ice}$ at comparatively warm temperatures.

It is a question why the $N_{ice}$ do not increase with decreasing temperature, as would be expected be-
cause the homogeneous ice nucleation rate then increases. This has been investigated by Gryspeerdt
et al. (2018) based on a 10 year global data set retrieved from satellite observations (DARDAR-$N_{ice}$,
Sourdeval et al., 2018, see also Section 6). Gryspeerdt et al. (2018) analyzed the $N_{ice}$ (>5µm) only at

cloud tops and also those throughout the cirrus clouds. From the cloud top analysis, a clear increase
of $N_{ice}$ with temperature was obvious, while integrating throughout the cirrus this temperature de-

---

[4]Note that for $N_{ice}$ -and thus $R_{ice}$- the number of hours spent in clouds in  Figure 6 is less than in  Figure 3. For details
see figure caption.





pendence becomes much weaker, though it is still present. Gryspeerdt et al. (2018) propose that the missing temperature dependence in the in-situ results could be due to a lack of in-situ measurements near the cloud top, where the temperature dependence is strongest. Another reason could be that the
higher $N_{ice}$ are short-lived (see Section 3.1.1) and thus not easy to trace by aircraft.

A further consideration of $N_{ice}$ frequencies of occurrence on a global as well as regional scales derived from satellite remote sensing will be presented in Section 6.

### 4.1.3 Ice crystal mean mass radius ($R_{ice}$-T)

The ice crystal mean size is calculated as mean mass radius $R_{ice}$ as shown in Footnote 1. $R_{ice}$ is
close to the common effective cloud particle radius $R_{eff}$ ($\sim$ ice Volume/Area), but can be calculated without knowing details of the ice particle size distribution (see also Krämer et al., 2016).

A total of 84 hours of observations are compiled in Figure 6 (bottom left panel). The $R_{ice}$ core band (frequencies $\gtrsim$ 5% per $R_{ice}$-T bin) decreases slightly with decreasing temperature which is caused by the decrease of the IWC core band, since the $N_{ice}$ band is not dependent on temperature
(see previous section).

The 10, 25, 50, 75 and 90% percentiles of the data set are plotted as thin, dotted and solid red lines. The black lines represent the minimum, middle and maximum $R_{ice}$ shown by Krämer et al. (2009) (see Figure 7). The range of $R_{ice}$ has shifted slightly to larger sizes in the new data set, which is caused by the $N_{ice}$ range extended towards lower concentrations that mainly consist of larger ice
crystals.

Remarkable is the drop of the most frequent $R_{ice}$ from larger to smaller ice crystals seen at around 215 K. This is probably the approximate temperature up to which liquid origin clouds are detected (see Luebke et al., 2016; Sourdeval et al., 2018, and also Section 6.2), that are characterized by larger ice crystals than in-situ cirrus (Krämer et al., 2016). At higher temperatures where both liquid
and in-situ origin cirrus prevail, higher IWCs - and thus also a larger $R_{ice}$ - occur more often than at lower temperature where only the thinner in-situ origin clouds exist. This is especially true at mid-latitudes, in the tropics the liquid origin cirrus reach also lower temperatures and thus no sudden drop of $R_{ice}$ is observed (see Figure 8, left and right bottom panel). The bifurcated structure of the most frequent $R_{ice}$ which ca be seen for temperatures $\lesssim$ 190 K is discussed in Section 4.2.1 (vii).

### 410 4.1.4 Clear sky and in-cloud $RH_{ice}$ ($RH_{ice}$-T)

The new $RH_{ice}$ clear sky and in-cloud data sets are displayed in the middle and top right panel of Figure 6. The respective earlier data sets of Krämer et al. (2009) are shown in Figure 7. The overall picture of the $RH_{ice}$ distributions has not changed, though the in-cloud data base has increased from 10 hours of measurements to about 116 hours and the clear sky observation time from 16
to even 320 hours. The only slight change is found in the in-cloud $RH_{ice}$, maybe caused by the larger number of data: below about 200 K, high supersaturations (>120%) occur more often and low





subsaturations (<80%) less often. At higher temperatures, the peak of $RH_{ice}$ frequencies at 100% is more pronounced.

For the new data set, we show for information in addition to the clear sky $RH_{ice}$ also the absolute

water vapor volume mixing ratio $H_2O$ in the bottom right panel of Figure 6. To guide the eye, water vapor saturation wrt ice, $H_2O_{sat,ice}$, is drawn as black solid line. The decrease of $H_2O$ with temperature is nicely seen in this panel and clear sky supersaturations appear in this portrayal as data points above $H_2O_{sat,ice}$.

### 4.2  Mid-latitudes and tropics

#### 4.2.1  Mid-latitude and tropical cirrus clouds

The data sets of mid-latitude/tropical cirrus consist of 67/101 hours of IWC, 29/61 hours of $N_{ice}$ and 28/56 hours of $R_{ice}$ measurements. This section describes some of some pronounced characteristics of mid-latitudes and tropical cirrus, which are displayed in Figure 8 (greenish/ reddish colors repre-

sent mid-latitude/tropical cirrus).

(i) *Temperature ranges:* Comparing mid-latitude and tropical cirrus, the first obvious difference -as expected when looking at the temperature profiles in Figure 2- is the temperature range. The observed mid-latitude cirrus rarely occur below 200 K, while tropical cirrus are detected down to

temperatures of 182 K. The core IWC/$N_{ice}$/$R_{ice}$range of both mid-latitude and tropical cirrus corresponds to the total climatology (see Figure 6).

(ii) *Mid-latitude WCBs and MCS:* At European mid-latitudes, the most frequent cirrus can be assigned to slow updrafts frontal systems (WCBs: warm conveyor belts) containing both liquid and

in-situ origin cirrus. High IWCs stem mostly from liquid origin WCB cirrus. Above the US continent, mesoscale convective systems (MCS) with faster updrafts are more frequent. The resulting liquid origin cirrus are thicker than the European, i.e. the ice crystals are larger and the IWC is higher (see also Krämer et al., 2016).

(iii) *Contrails:* A striking feature in the cirrus observations are $N_{ice}$ of up to several hundreds per $cm^3$, that are found at mid-latitudes in the temperature range of about 210-220 K, which corresponds to about 10 km altitude (see Figure 2), the typical cruising level of passenger aircraft. They can be attributed to young contrails, which were a topic of investigation during COALESC 2011 (Jones et al., 2012) and also ML-CIRRUS 2014 (Voigt et al., 2017). Higher mid-latitude $N_{ice}$ at higher tem-

peratures are most probably in-situ origin cirrus caused by stronger updrafts in e.g. mountain waves.

(iv) *Drop freezing:* High $N_{ice}$ between ten and hundredper $cm^3$ or even more are found above 235 K. Such high concentrations together with small $R_{ice}$ are typical for frozen supercooled liquid drops, i.e. are an indication for liquid origin cirrus caused by homogeneous drop freezing in tropical or



mid-latitude convective systems with fast updrafts.

(v) *Tropical convection:* However, in tropical cirrus, in particular for temperatures $\gtrsim$ 220 K, high IWCs and $N_{ice}$ above the core range -corresponding to convective liquid origin cirrus- become more frequent, while $R_{ice}$ tends to be smaller. This is most likely caused by the fast convective vertical ve-

locities that often let clouds in the mixed-phase temperature regime rise to the cirrus altitude range.

(vi) *Tropical deep convection:* Also remarkable is that in the tropics, massive convective liquid origin cirrus carrying a high IWC -often accompanied by high $N_{ice}$- are detected down to very cold temperatures ($< 200$ K), which corresponds to high altitudes up to 17 km. The thickest cirrus with

very high IWC and $N_{ice}$ can also be seen in Figure 3 (upper and middle left panel: distribution of cirrus with latitude; blue data points represent liquid origin with a high IWC/$N_{ice}$) at around 25° northern latitude and 16-18 km altitude. These exceptional thick and cold cirrus at high altitudes were observed in the Asian monsoon tropical tropopause layer (TTL). The observed many small $N_{ice}$ are generated most likely by in-situ homogeneous ice nucleation, triggered either by fast updrafts

in gravity waves (see also Spichtinger and Krämer, 2013; Jensen et al., 2017) or deep convection. They often occur in the tops of to massive liquid origin cirrus with very high IWC; a theoretical description of such clouds is given by Jensen and Ackerman (2006).

(vii) *TTL cirrus:* The discussion of cirrus clouds in the tropical tropopause layer is presented in

Section 5.

### 4.2.2 Mid-latitude and tropical humidity

The mid-latitude/tropical humidity data sets include 35/81 hours of in-cloud $RH_{ice}$ and 94/226 hours of clear sky $RH_{ice}$/$H_2O$ measurements, which are displayed in Figure 9 with the same color code as in Figure 8.

In clear sky at temperatures higher than about 200 K, $RH_{ice}$ is most often below saturation and randomly distributed in both mid-latitudes and tropics (Figure 9, middle row). Clear sky supersaturations occurs less frequently, simply because they only take place in those periods when moist air parcels are cooled towards the ice nucleation thresholds (heterogeneous or homogeneous), which are rare compared to drier conditions of the atmosphere.

Below about 200 K, i.e. in the TTL (see also Section 5), the clear sky $RH_{ice}$ distribution looks very different. In this region, $H_2O$ is low and its variability is only small (Figure 9, bottom row). We plotted lines of constant $H_2O$ (1.5, 3 and 5 ppmv) in the clear sky $RH_{ice}$ panels to illustrate that for constant $H_2O$, $RH_{ice}$ increases only due to the decrease in temperature, i.e. $H_2O_{sat,ice}$. Thus it can be seen that at mid-latitudes $RH_{ice}$ mostly represent $H_2O$ values around 3 ppmv and in the tropics

between 1.5 and 3 ppmv. Since in the tropics much colder temperatures are reached, the respective $RH_{ice}$ ranges from 10 up to about 150% or even more.





Clear sky $RH_{ice}$ above the homogeneous freezing line are under discussion, because they would indicate that no supercooled liquid aerosol particles are present to initiate freezing, or that the homogeneous freezing is prevented, for example by organic material contained in the aerosol particles.

More probably, these few data points are outliers; note also that the uncertainty of $RH_{ice}$ rises from approximately 10% at warmer temperatures to about 20% at colder temperatures (Krämer et al., 2009).

Inside of cirrus, the peak of the $RH_{ice}$ frequencies is mostly around the thermodynamical equi-
librium value of 100% (saturation) at mid-latitudes as well as in the tropics (Figure 9, upper row). However, in the TTL, at the coldest prevailing temperatures ($\lesssim$ 190 K), supersaturation increasingly becomes the most common condition, which is discussed in more detail in Section 5. High supersaturations at low temperatures were also reported by (Krämer et al., 2009) and Jensen et al. (2013a), and the reason given for the existence of such high supersaturation is low $N_{ice}$ concentra-
tions, which were mostly present at low temperatures in these observations (Figure 7). But, Jensen et al. (2013a) also showed that $RH_{ice}$ rapidly drops to saturation in the presence of many ice crystals. As can be seen from Figure 8, middle right panel (and discussed in Section 4.1.2), in the new data set $N_{ice}$ cover a broader concentration range in comparison to the earlier data (Figure 7), while $RH_{ice}$ is supersaturated in most cases. This is not straightforward understandable because of the complex
relation between $RH_{ice}$ and $N_{ice}$. We will investigate the TTL supersaturations at in cirrus clouds in a follow-up study.

## 5 TTL in-situ climatologies
### in- and outside of the Asian monsoon anticyclone

The tropical tropopause layer is the region above the upper level of main convective outflow, where
the transition from the troposphere to stratosphere occurs. It is placed at temperatures <205 K between ~150 hPa/355 K potential temperature/14 km and 70 hPa/425 K/18.5 km (Fueglistaler et al., 2009, see also Figure 2). The coldest temperatures are found here at the point where the slope of the temperature profile reverses (cold point tropopause, CPT). In the TTL, the prevailing dynamical conditions are very slow large-scale updrafts superimposed by a spectrum of high-frequency gravity
waves (i.e. Spichtinger and Krämer, 2013; Dinh et al., 2015; Jensen et al., 2017; Podglajen et al., 2017). In addition, deep convection with fast updrafts can overshoot into the TTL occasionally.

Cirrus clouds and humidity in the TTL deserve a special consideration, because this region represents the main pathway by which water vapor enters the upper troposphere and lower stratosphere (UT/LS) where it is further distributed over long distances (e.g. Brewer, 1949; Rolf et al., 2018;
Vogel et al., 2016; Ploeger et al., 2013). This is of importance, because water vapor is a greenhouse gas that has a significant impact on the surface climate of the Earth, especially in the tropical UT/LS (Solomon et al., 2010; Riese et al., 2012), but also in the LS at high latitudes where it is being trans-



ported from the tropics. Cirrus clouds are of particular relevance as regulators of the partitioning of $H_2O$ between gas and ice phase. Furthermore, they have a climate feedback themselves by influenc-
ing the Earth's radiation balance (e.g. Boucher et al., 2013). Thus, the simultaneous observation and analysis of both cirrus clouds and humidity ($H_2O$) in this climatically crucial region are of special interest.

     This section specifically reports cirrus clouds together with humidity (from the climatologies pre-
sented in Section 4), that was recently observed for the first time in the TTL in the Asian monsoon anticyclone (June to September) in comparison to observations in the surrounding tropical regions. As the Asian monsoon is characterized by strong convective activity, this comparison will show the difference to less active, calmer areas. Observations of Asian monsoon TTL cirrus clouds and humidity are of particular importance because, amongst other trace species, large amounts of $H_2O$
and also cloud particles are convectively transported upwards from far below, where the additional $H_2O$ in turn often cause cirrus formation (e.g. Ueyama et al., 2018, and references therein). Directly injected $H_2O$ or $H_2O$ from sublimated ice crystals can then be mixed up into the stratosphere. Thus, the Asian monsoon anticyclone represents the major gateway for $H_2O$ between UT and LS (e.g. Fueglistaler et al., 2009; Ploeger et al., 2013) and currently it is under discussion to what extent
cirrus cloud particles contribute to the amount of $H_2O$ entering the stratosphere (e.g. Ueyama et al., 2018, and references therein).

     The airborne measurements in the Asian monsoon (see Figure 1 and Tables 4 and 5) are per- formed during July-August 2017 out of Khatmandu, Nepal, during a field campaign embedded in the StratoClim project (http://www.stratoclim.org/). An overview of the observations is given in
Figures 10, 11 and 12, where the frequencies of IWC, $N_{ice}$, $R_{ice}$ and in-cloud, clear sky $RH_{ice}$ as well as the clear sky $H_2O$ volume mixing ratio are shown in the temperature and also the potential temperature $\Theta$ parameter space. [5] Most of the measurements during StratoClim are performed at temperatures $\lesssim 205$ K, corresponding to potential temperatures $\gtrsim 355$ K and altitudes about $\gtrsim 14$ km, i.e. in the TTL (marked in the middle panels of the Figures).
The surrounding, in most (but not all) cases calmer tropical TTL regions are represented by ob- servations during the campaigns shown in Figure 1 and listed in Tables 4 and 5. The majority of the data are sampled during ATTREX_2014 and POSIDON_2016. The TTL measurements in the temperature parameter space are shown in Figures 8 and 9; in these plots, the Asian monsoon obser- vations are included, but since the measurements represent only a small part of all TTL observations,
excluding the StratoClim campaign only slightly change the frequency distributions. Thus, the Fig- ures are representative for the TTL outside of the Asian monsoon anticyclone, so we refrain from showing an additional Figure. The measurements in the $\Theta$ parameter space (StratoClim excluded) are presented in the left panels of Figures 11 and 12.

---

[5] The additional $\Theta$ portrayal provides a more detailed impression of the distribution of cirrus clouds and humidity around the CPT, i.e. at the transition between troposphere and stratosphere (see Figure 2).





### 5.1 TTL cirrus clouds

#### 5.1.1 Temperature parameter space

In the tropics outside of the Asian monsoon (Figure 8, right column), cirrus IWC and $N_{ice}$ range from very low to quite high values. We want to draw attention to a special feature of the most frequently occurring cirrus $N_{ice}$ (middle right column). Two main branches of most frequent $N_{ice}$ are found, one at very low ($\sim 10^{-3}$ cm$^{-3}$) and the other at moderate ($\sim 5 \cdot 10^{-2}$ cm$^{-3}$) $N_{ice}$. These two branches are

also reflected in $R_{ice}$, where the larger ice crystals are found together with lower concentrations and vice versa, as shown in Figure 4. More precisely, cirrus with $N_{ice} \lesssim 1.5 \cdot 10^{-2}$ cm$^{-3}$ consist of ice particles larger than about 20 μm [6] . That means that the ice crystal spectra of the low $N_{ice}$ cirrus most likely represent aged clouds (in-situ or liquid origin) where the smaller nucleation mode ice crystals are either grown to larger sizes in supersaturations or evaporated in case of subsaturated conditions

(see also Section 3.1.1). The middle $N_{ice}$ cirrus containing ice crystals smaller than 20 μm are most likely young cirrus that have formed in-situ, because these small crystals quickly (on a time scale of 10 - 20 minutes) grow to larger sizes. It is impossible to speculate if they have formed homo- or heterogeneously, since both pathways might produce such $N_{ice}$ in the slow updrafts prevailing in the TTL. The reason that the aged cirrus marked in Figure 8 (middle right column) become

visible only in the TTL, though they certainly occur in all cirrus regions, is probably the calmer dynamic environment with -in comparison to lower altitudes- less frequently occurring temperature fluctuations. These fluctuations likely cause new ice nucleation events superimposing the aged cirrus.

The clouds observed in the Asian monsoon include in-situ formed cirrus as well as cirrus clouds from overshooting deep convection. In the much uneasier Asian monsoon conditions, IWC and $N_{ice}$

are most frequently above the median lines derived from the entire climatology (Figure 10, left column) and also high in comparison to the tropical climatology (Figure 8, right column). Here, the highest observed values (at temperatures <205 K) are reached with IWC mixing ratios of up to 1000 ppmv and a maximum $N_{ice}$ as high as 30 cm$^{-3}$ (note that the ice crystal shattering was significantly minimized, see Section A2.2). Also, the ice crystals mean mass size $R_{ice}$ is above the

median, especially at very low temperatures, which means that large ice crystals are found around the cold point. These exceptional findings are recorded during flights in strong convection, where liquid origin clouds from far below are detected in the upper part of the Asian monsoon anticyclone simultaneously with freshly homogeneously nucleated ice crystals. The observations were possible due to the pilot of the Geophysica aircraft, who dared to fly into the strong updrafts. Because of the

dangerous nature of measurements under such conditions, the frequency of convective – and also orographic wave cirrus – is under-represented in the entire in-situ climatology.

---

[6]It should be noted that concentrations $\lesssim 0.1$ cm$^{-3}$ of cloud particles $\lesssim 20$ μm are below the detection limit of cloud spectrometers, i.e. small ice crystals with such low concentrations could be present (see Appendix A2.2).



### 5.1.2 Potential temperature Θ parameter space

The Asian monsoon cirrus clouds in the Θ parameter space are shown in the right panel of Figure 11.
TTL average upper and lower boundaries and the CPT are marked in the middle panel of the Figure
following the definition of Fueglistaler et al. (2009). In the left panel of Figure 11, the climatologies
of the tropical observations without Asian monsoon are shown.

From Figure 11, upper row, it can be nicely seen how steeply the IWC increases during the
transition from the TTL to the free troposphere. In the tropics outside of the Asian monsoon, the
maximum Θ where ice is detected is about 420 K ($\sim$ 19 km). The mixing ratios of these highest
cirrus are about 0.05 ppmv. At about 380 K (CPT, $\sim$ 16-18 km), the range of most frequent IWCs
broadens, ranging between about 0.005 and 1 ppmv. Some higher IWCs are also detected in the
upper TTL, indicating that overshooting convection (liquid origin cirrus) is also embedded in these
measurements. Below about 380 K the most frequent IWC increases steadily up to values of around
10 ppmv at 355 K and 1000 ppmv at 340 K and farther below.

In the Asian monsoon, the maximum Θ where ice is detected is about 415 K ($\sim$0.5 ppmv) and
at 400 K the IWC ranges between 0.05 - 0.1 ppmv, as outside of the Asian monsoon. But, at 380 K
(CPT), the range of most frequent IWCs rises to 0.5 - 2 ppmv, then steadily increasing below about
380 K to values of around 50 - 500 ppmv at 355 K. Obviously, in the Asian monsoon TTL the IWC
is enhanced by about a factor of ten and more in the Asian monsoon around the CPT and below
by injection of liquid origin cirrus in convective overshooting events. The highest TTL IWCs (up to
1000 ppmv) are detected in the Asian monsoon up to 390 K.

These overshooting cirrus around and below the CPT are also seen in high ice crystal numbers $N_{ice}$
($\gtrsim$0.5 cm$^{-3}$) when comparing the Asian monsoon with the other tropical regions (Figure 11 middle
panels). Striking are again the high $N_{ice}$ up to 30 cm$^{-3}$ in the Asian monsoon, already discussed with
respect to the temperature parameter space. Here it is visible that this burst of in-situ homogeneous
ice nucleation in a strong convective event (theoretically described by Jensen and Ackerman, 2006)
took place below the CPT.

Above 390 K, at the transition to the stratosphere, $N_{ice}$ is very low, mostly lower than 0.1 cm$^{-3}$.
Recall that these low concentrations only contain particles $> 20\,\mu$m (Footnote 6 and Appendix A2.2).
This means that in the Asian monsoon (middle right panel of Figure 11) low concentrations of larger
ice crystals (together wit higher IWCs) are more often present at such altitudes in comparison to the
surrounding tropics.

In the TTL outside of the Asian monsoon, the two branches of more frequent $N_{ice}$ (and $R_{ice}$) –
discussed with respect to the temperature parameter space (Figure 8, right middle panel) – are also
very clearly visible. The smaller/larger ice crystals with higher/lower concentrations were identified
as young and aged cirrus clouds, which most probably have formed in-situ. Farther below, $N_{ice}$
further increases with decreasing Θ/altitude and more and more small liquid origin cloud particles
(frozen or liquid) with higher concentrations appear.





### 5.2 TTL humidity

#### 5.2.1 Temperature parameter space

Considering the in-cloud and clear sky $RH_{ice}$ in the Asian monsoon (Figure 10, right column) in comparison to the entire tropical climatologies (Figure 9, right column), significant differences are visible. Below about 200 K, the most frequent in-cloud $RH_{ice}$ are found in supersaturation in the Asian monsoon cirrus. In the entire climatology (Figure 9, right column) this occurs only below about 185 K. The same is seen in clear sky $RH_{ice}$: higher supersaturations occur frequently already at higher temperatures in the Asian monsoon comparison to the entire tropical climatology. These higher supersaturations reflect a higher amount of water vapor in the Asian monsoon TTL, visible through the dashed lines in the middle right panel of Figure 10. They show the increase of $RH_{ice}$ caused by the decrease of $H_2O_{sat,ice}$ during cooling of air at a constant $H_2O$ mixing ratio. These lines correspond to $H_2O$ between 3 and 5 ppmv in the Asian monsoon, while in the total tropical climatology the most frequent $H_2O$ ranges only between 1.5 and 3 ppmv (Figure 9; compare also the bottom panels of the Figures, where $H_2O$ is plotted). Inevitably, in case more water is present while cooling of air, supersaturation is achieved already at higher temperatures. Implications of the observed supersaturations are further discussed in Section 5.3.

#### 5.2.2 Potential temperature $\Theta$ parameter space

The high $RH_{ice}$ in and outside of cirrus clouds in the Asian monsoon are also visible in the $\Theta$ representation of the humidity (see Figure 12, top and middle right panels). In the tropical climatology outside of the Asian monsoon (top left panel of Figure 12), $RH_{ice}$ most frequently center around saturation inside of the cirrus clouds. In the cloud free TTL outside of the Asian monsoon (middle left panel of Figure 12), a humidification of the layer between $\sim$ 360 and 380 K to $RH_{ice}$ around 90% can be seen (for more detail see Schoeberl et al., 2019).

In the Asian monsoon, on the other hand, the in-cloud $RH_{ice}$ in the TTL exceeds saturation (top right panel of Figure 12). Especially, beween 380 and 400 K the most frequent $RH_{ice}$ is around 130%. Also outside of clouds (middle right panel of Figure 12), around the CPT supersaturation is frequently detected and in general the humidification is higher than in the surrounding tropical regions. This is in agreement with Schoeberl et al. (2019), who reported high $RH_{ice}$ coincident with the Himalaya Monsoon during summer and closely associated with convection. The higher amount of water vapor in the Asian monsoon in comparison to the other tropical regions can also be seen in the $H_2O$ volume mixing ratios (bottom panels of Figure 12). In the tropics without Asian monsoon, the most frequent $H_2O$ between 365-410 K is 1.5-4 ppmv, while in the Asian monsoon we found 3-8 ppmv, as also seen in the temperature parameter space. This finding is in accordance to other studies, but has been observed in-situ from aircraft directly in the Asian monsoon for the first time. For example, analyzing the air mass histories of higher in-situ $H_2O$ observations at other locations





(highest values of 8 ppmv), Schiller et al. (2009) found that those air masses has passed the Asian monsoon region. Also, Ueyama et al. (2018) (and references therein) reported 5-7 ppmv at 100 hPa from MLS observations and extensive model simulations over the Asian summer monsoon region.

### 5.3 $H_2O$ and IWC for transport to the stratosphere

The *$H_2O$ transport to the stratosphere* is regulated by the coldest temperature an air parcel expe-
riences during transition through the tropopause region. The water amount passing the tropopause is set by the freeze-drying process associated with this transition (Jensen and Pfister, 2004) and is discussed to be as low as $H_2O$ saturation at the minimum temperature (e.g. Schiller et al., 2009).

However, Rollins et al. (2016) showed for the ATTREX 2014 observations, that the water vapor at the stratospheric entry point is higher by ∼10%, because the water vapor depletion by ice crystals
becomes increasingly inefficient at temperatures below 200 K (note that this is a rough estimate of the excess water vapor at the stratospheric entry point based on the actual temperature, which cloud be somewhat different from the minimum temperature of the air parcel's back trajectory). This is of importance, since already small amounts of $H_2O$ can influence the stratospheric radiation budget (Solomon et al., 2010; Riese et al., 2012).

High supersaturations at the coldest points of the TTL are also discussed in Section 5.2.1. The AT-TREX 2014 and also POSIDON 2016 observations are included in Figure 9, right panels, where the most frequent in cloud and clear $RH_{ice}$ at these temperatures can be seen to increase with decreasing temperature as discussed by Rollins et al. (2016). But, in the Asian Monsoon measurements (Figure 10, right panels), $RH_{ice}$ at the coldest temperatures is even higher above saturation than in the
calmer tropics outside, the saturation is exceeded by about 20-50% below 195 K.

Therefore, we extend the conclusion of Rollins et al. (2016): taking saturation at the stratospheric entry of an air mass as set point for water vapor transport to the stratosphere, the transport is underes-timated by ∼10% in regions of weak convective activity. In convective regions the underestimation increased to 20-50% in our observations.

As mentioned earlier, the question of how much *$H_2O$ from convectively injected ice crystals* is transported from above the CPT further into the lower stratosphere is subject of recent research. In the study of Ueyama et al. (2018), it is concluded that over the Asian monsoon at 100 hPa, convection is the dominant driver of the localized $H_2O$ and that nearly all of the convective enhancements in
$H_2O$ is due to the effect of convective humidification, while convectively detrained ice crystals have negligible impact.

From our measurements in and above the Asian monsoon anticyclone, however, it is difficult to estimate to what extent cirrus cloud particles contribute to the amount of $H_2O$ that might enter the stratosphere. Nevertheless, from Section 5.1 we know that the most frequent IWCs between about
365-400 K are 10 to 0.5 ppmv. Above 400 K, up to 415 K some overshoots with IWCs between 0.5 to 8 ppmv are detected (Figure 11, upper right panel). These IWCs are similar to the amount



of gas phase $H_2O$, which is between 3 to 8 ppmv in this region above the CPT (see last Section). Such amounts of $H_2O$ and IWC indicate that the air masses originate from an altitude of about 15 km Figure 3, which corresponds to the lower bound of the TTL (see Figure 2).

The question is if the ice particles found in the upper TTL will further grow and sediment out in supersaturated conditions (dehydration) or evaporate and add $H_2O$ to the gas phase $H_2O$ in subsaturation (hydration, see Jensen et al., 2007; Schoeberl et al., 2018). Inspecting the in-cloud $RH_{ice}$ (see Figure 12, upper right panel) related to the respective IWCs, it can be seen that the air in the overshoots $\gtrsim 400$ K with the fairly high IWCs (0.5 to 8 ppmv) is subsaturated, while at lower

altitudes both super- and subsaturation occur. This means that convective overshoots in the Asian monsoon can locally contribute a significant amount of water that might be further transported into the stratosphere. Unfortunately, we can not say anything about the frequency of these events in this study.

Above the CPT in the surrounding tropics outside of the Asian monsoon, the in-cloud $RH_{ice}$ at

$\gtrsim 400$ K are also subsaturated. Here, the IWCs range only between 0.02 and 2 ppmv, with some single data points up to about 2 ppmv. The related $H_2O$ is 1.5 to 3 ppmv.

Comparing gas phase $H_2O$ and IWC above the Asian monsoon CPT (both 3-8 ppmv) with the surrounding tropical regions (1.5-3/0.02-2 ppmv) might be an indication by in-situ observations that,

driven by overshooting convection, the Asian monsoon could by an important source for transport of $H_2O$ to the stratosphere.

## 6  Global and regional cirrus $N_{ice}$ climatologies
   from satellite remote sensing

Though the new in-situ cirrus climatologies presented in Section 4 represent a considerable data set

from research aircraft from which substantial insights are gained, it is still a mixture of different meteorological situations encountered during various field campaigns that does not necessarily display a statistically representative overall picture of the distribution of cirrus cloud properties. For this purpose, long-term global satellite remote sensing observations would be the method of choice, though retrievals of cirrus microphysical properties also have their own difficulties and limitations. This is

especially true for the ice crystal number $N_{ice}$, which is particularly challenging to estimate from satellite remote sensing. The new retrieval method for $N_{ice}$ introduced in Section 2.2, DARDAR-Nice, has however demonstrated to be able to satisfactorily reproduce in-situ ice concentrations (Sourdeval et al., 2018).

In this section, we show a global climatology of $N_{ice}$ derived from DARDAR-Nice in the same

presentation as the in-situ $N_{ice}$ climatology. Median $N_{ice}$, 25/75[th] and 10/90[th] percentiles are provided for a global data set. In addition, median $N_{ice}$ are discussed for the Arctic region, the northern mid-latitudes, the tropics, the southern mid-latitudes and Antarctica.





## 6.1 Comparison of remote sensing and in-situ $N_{ice}$

As a base for reliable $N_{ice}$ climatologies from satellite remote sensing, in-situ measurements of PSDs
(ice particle size distributions) from five campaigns of the Cirrus Guide II are used for an evaluation
of the DARDAR-Nice retrieval algorithm (see Section 2.2). This directly follows the evaluation of
DARDAR-Nice presented in Sourdeval et al. (2018), where the same campaigns were used. Figure 13 (top panel) shows the $N_{ice}$ frequencies of occurrence in 1K temperature bins of the in-situ
measurements, the corresponding frequencies obtained from DARDAR-Nice are shown in the bottom panel. The latter are derived on the basis of two input parameters which are extracted from the
in-situ PSDs of the five campaigns listed in the figure caption, rather than being constrained from
lidar-radar measurements during usual retrievals. Note that DARDAR-Nice provides concentrations
of ice crystals $> 5\mu m$, as it has not been evaluated for smaller sizes, while $N_{ice}$ from in-situ observations includes sizes $> 3\mu m$; however, the resulting difference in the ice concentrations is negligible.
Also, the 1-Hz in-situ PSDs are here sampled into 10-s averages to simulate the 1.7-km horizontal
resolution of the DARDAR-Nice retrievals (assuming a flight speed of about $170\,\mathrm{m\,s^{-1}}$).

The black solid and dotted lines of the in-situ climatology indicate the 50, 25 and 75[th] percentiles,
which agree well with those of the entire in-situ climatology (see Figure 6, middle left panel). This
demonstrates that the selected subset is statistically representative of the entire database (indeed, the
entire $N_{ice}$ climatology contains only one additional campaign, see Table 5). A satisfactory agreement
between DARDAR-Nice and the in-situ observations is also seen, demonstrating that DARDAR-Nice $N_{ice}$ retrievals very well match the in-situ data base for known size distibutions.

A detailed comparison of the DARDAR-Nice percentiles (black solid and dashed lines) with those
of the in-situ observations, however, yields a small offset by a factor of 1.73, which is visualized in
Figure 13 by the average medians of DARDAR-Nice (straight black line, $0.064\,\mathrm{cm^{-3}}$) and in-situ
(straight blue line, $0.037\,\mathrm{cm^{-3}}$). One reason for this offset lies in the method of the retrieval. The
two parameters, $N_0^\star$ and $D_m$, calculated here from the in-situ PSDs (usually from lidar-radar observations) are fed into a predefined four-parameter gamma-modified function to calculate ice particle
size distributions (PSDs, see Section 2.2). $N_{ice}$ is then calculated by summing up individual ice concentrations over a grid of size bins distributed over the gamma-shaped PSD for the entire range of
observed ice crystal sizes (usually, a continuous integration of the PSD is performed). However, in
the in-situ measurements, there are often PSDs that do not contain ice particles $< 20\,\mu m$, i.e. these
size bins are empty. Such PSDs represent aged cirrus after the ice nucleation phase where the smaller
ice crystals have grown to larger sizes (see Section 4.2.1). DARDAR-Nice, however, assumes a modified Gamma distribution including all size bins, which partly explains the described positive offset.
This behavior was also found by Wolf et al. (2019) who parameterized in-situ and liquid origin cirrus from balloon-borne measurements by Gamma functions. It is to note that when subtracting the
ice concentrations of the 'empty bins' of in-situ PSDs from the retrieved DARDAR-Nice in a direct
intercomparison, the agreement of $N_{ice}$ is improving (not shown here). This is an important finding,





as gamma functions are often used to represent cirrus PSDs, both in remote sensing retrieval algorithms and in global models. On the other hand it has to be noted that an offset on the order of a factor of 1.73 is tolerable given the variability of $N_{ice}$ (six orders of magnitude) and other possible error sources in the measurements, both in-situ and remote sensing.

    Higher $N_{ice}$ in DARDAR-Nice towards low temperatures are also not surprising, as Sourdeval

et al. (2018) reported that the PSD parametrization used by DARDAR-Nice predicts higher concentrations of small particles ($D_{ice} < 25\,\mu m$) than the in-situ comparative measurements. The overestimation increases with decreasing temperature, as the small ice particles dominate the PSDs more and more. This might be caused by a sharper representation of small ice concentrations in the PSD parametrization in comparison to the in-situ small ice particle measurements below $\sim$210 K.

A further expected difference between the data sets are their detection limits. The lowest $N_{ice}$ that can be detected by the in-situ instruments is $10^{-4}\mathrm{cm}^{-3}$, for the highest $N_{ice}$ there is no limitation. Detection limits for DARDAR-Nice depend on lidar-radar sensitivity but are also influenced by the instrumental resolution, which may cause specific features to remain undetected. This effect, however, is represented in Fig. 13 by the 10-s sampling of 1-Hz in-situ PSDs. Indeed, DARDAR-

Nice misses the high $N_{ice}$ in the temperature range 210-220 K, but these are young, line shaped contrails which are too small scale to be detected from satellite. Also, part of the thinnest cirrus are not represented by DARDAR-Nice. This will be further discussed in the next section describing the global $N_{ice}$ distribution from DARDAR-Nice.

    Overall, following the above-mentioned arguments, an overestimation within a factor of 2 in

DARDAR-Nice by comparison to the in-situ data set is expected, although there is still some uncertainty as to how this overestimation will propagate globally and over various ice cloud regimes. The ratio of 1.73 in $N_{ice}$ between DARDAR-Nice and the in-situ data set, found from Figure 13, should therefore be thought of as a minimum expected bias. Finally, the bias is expected to be stronger at low temperatures than at high temperatures, due to the increased likelihood of missing bins as well

as the higher importance of the representation of the small ice particle mode in the PSDs.

### 6.2  Global cirrus $N_{ice}$

The global frequency distribution of $N_{ice}$ from ten years of satellite observations is shown in Figure 14. The data was collected twice a day, approximately at midday and at midnight (satellite equator-crossing time is 1.30 am/pm), from June 2006 to December 2016. Overall, the global satel-

lite data set consists of nearly $2 \times 10^{10}$ $N_{ice}$ retrievals. The color code represent frequencies of occurrence, the black contours the 25, 50 and 75$^{th}$ percentiles.

    From the median (solid black line), a slight increase of $N_{ice}$ with decreasing temperature is visible, which is somewhat different to the in-situ median $N_{ice}$ (solid blue line, from Figure 6, middle left panel), where no temperature dependence is found. The DARDAR-Nice temperature dependence

was already noted and discussed by Gryspeerdt et al. (2018). Another difference is that at $T \gtrsim 210$



K, the maximum $N_{ice}$ of DARDAR-Nice reaches up to $100\,cm^{-3}$, while the maximum in-situ $N_{ice}$ is only about $10\,cm^{-3}$ (outside of young contrails).

We attribute the slightly increasing median $N_{ice}$ with decreasing temperature to homogeneous ice nucleation events, because homogeneous ice nucleation rates increase with decreasing temperature, but their appearance in space and time is transient, as discussed in Section 4.1.2. Thus, such events are difficult to find by research aircraft. Hence, these events are likely underrepresented in the aircraft observations.

The thinnest cirrus with low $N_{ice}$ are represented by DARDAR-Nice for temperatures $\gtrsim 190$ K. At $\lesssim 190$ K, however, a decreasing detectability of thin cirrus becomes visible in the DARDAR-Nice climatology: in contrast to the in-situ climatology, lesser or no thin cirrus are measured with decreasing temperature. The reason is that the colder the cirrus clouds, the smaller are the ice crystals and the lesser their IWC (Figure 10, top and bottom left panel). In the in-situ $N_{ice}$ climatology, in the TTL ($\lesssim 205$ K), an increased occurrence of very thin cirrus with $N_{ice}$ around $0.001\,cm^{-3}$ is reported in Section 4.2.1 and explained as aged cirrus consisting of only larger ice crystals. Though this type of clouds is partly missing in the DARDAR-Nice climatology, the median $N_{ice}$ decreases again at these temperatures. This might reflect the frequent presence of aged thin cirrus. Note, however, that the statistics at these temperatures is based on a lower number of observations than at higher temperatures (see Figure 15).

The median $N_{ice}$ across all temperatures of the 10 years DARDAR-Nice climatology is about $0.100\,cm^{-3}$ (Figure 14, plain cyan line). Adjusting this number to the offset factor of 1.73 between DARDAR-Nice and the in-situ observations reported in the previous section yields an in-situ average median of $0.056\,cm^{-3}$ (dashed cyan line, see also Table 3). It is nevertheless important to note that such median is by construction computed assuming that all temperature bins have equal weight. When considering their actual relative occurrence as function of temperature (see Figure 15 and later discussion), the in-situ adjusted global DARDAR-Nice median falls down to $0.046\,cm^{-3}$. This is still larger than the average median of $0.03\,cm^{-3}$ reported from the $N_{ice}$ in-situ climatology (blue line; from Figure 6). Nevertheless, in the light of the offset between the data sets, the decreasing detectability of thin cirrus by DARDAR-Nice and other sources of error in both methods, the agreement between the DARDAR-Nice and the in-situ $N_{ice}$ climatology is good.

Altogether, both data sets have advantages and disadvantages: DARDAR-Nice has the advantage of the long, global time series where all atmospheric situations above the detection limit showing up at the times of observations are sampled. However, the thinnest cirrus at cold temperatures are not detected. Further, an offset is found between DARDAR-Nice and the in-situ observations. On the other hand, the in-situ $N_{ice}$ climatology is based on a much smaller data set which is randomly distributed between different atmospheric situations. But, nearly the complete range of possible $N_{ice}$ can be detected by the in-situ instruments.



For comparisons with other N$_{ice}$ data sets or results from global models, the percentiles of the adjusted DARDAR-Nice as well as the average in-situ N$_{ice}$ are shown in Figure 15 and listed in Table 2 (note that an in-situ data set slightly smaller than the data set presented here, but extended with regard to Krämer et al. (2009) is already used for this purpose by Bacer et al., 2018; Penner et al., 2018; Righi et al., 2019).

The DARDAR-Nice data set does not only provide the N$_{ice}$ distribution with temperature as shown in Figure 14, but, as it contain the entire spatial and temporal N$_{ice}$ information, also the actual appearance of cirrus clouds in the N$_{ice}$-T parameter space. This is shown in Figure 15, where the occurrences of DARDAR-Nice N$_{ice}$ retrievals are indicated by a the colored background. It can be noticed that, in regions of high frequency of occurrence (for T $\gtrsim$ 210 K), the agreement between DARDAR-Nice and the in situ data set is best.

From this portrayal it can be seen that the cirrus clouds do not spread evenly across all temperatures, but that about 50% of the cirrus appear at temperatures above about 225 K (see also Table 3, temperature range of global 'most frequent' median N$_{ice}$). This clustering of cirrus at warmer temperatures is likely because in-situ and liquid origin cirrus clouds are both found in this warmest and lowest cirrus layer, while at colder temperatures and higher altitudes only in-situ cirrus are present (see also Section 3.1 and Figure 3). This vertical structure of the cloud types is also reported from observations by Krämer et al. (2016); Luebke et al. (2016); Wolf et al. (2018), from global modeling by Gasparini and Lohmann (2016) (they name liquid origin cirrus 'detrained ice') and also from 12 years ERA-interim data analysis in the North Atlantic region by Wernli et al. (2016). Wernli et al. (2016) also provides the relative frequencies of liquid origin cirrus with respect to pressure, which decreases from about 55% of the cirrus at 500 hPa (roughly 240 K), to 50, 30, 5% at 400, 300, 200 hPa (roughly 230, 215, 200 K; see Figure 2).

The analysis presented here provides the information that globally, half of the cirrus are in the lowest, warmest cirrus layer that contains a significant part of liquid origin cirrus. This is of importance because of the differing radiative properties of in-situ and liquid origin cirrus shown in Section 3.1.1 (see Figure 5): if the thick liquid origin cirrus clouds occur with such a high frequency, their strong cooling effect might exceed the much smaller warming effect of the in-situ origin cirrus which could lead to a general cooling by cirrus.

### 6.3 Regional cirrus N$_{ice}$

Lastly, we provide an impression of the regional variations of N$_{ice}$ (see Table 3). To this end, subsets of the DARDAR-Nice data for five regional latitude bands between 90°, 67.7°, 23.3°, -23.3°, -67.7°, -90°, corresponding to the Arctic, Mid-lat North, the Tropics, Mid-lat South and Antarctica, are considered, and three different N$_{ice}$ medians are computed for each region and for the global data set. A first median (labeled "all Temp.") corresponds to the median N$_{ice}$ value across all temperatures and considering all temperature bins to be equiprobable, i.e. the same as used in Figure 14. A second





(labeled "T-int weighted") represents the median using the relative occurrence of each temperature
interval as a weight. The last ("most frequent") shows the most frequently appearing $N_{ice}$, consid-
ering those temperature intervals that contain 50% of the total $N_{ice}$ occurrence; T-range denotes the
respective minimum and maximum temperatures (note that in the in-situ data set all three methods
would yield to an identical median due to the flat temperature dependence).

Comparing the three median $N_{ice}$, it is obvious that in most cases the 'T-int weighted' and 'most
frequent' medians are smaller than those for 'all temperatures'. This behavior is found because the
cirrus clouds appear often at warmer temperatures (see previous subsection) where $N_{ice}$ is moderate,
which can be seen particularly in the 'most frequent' medians, where the temperature range of the
cirrus occurrence is given.

The global 'most frequent' median $N_{ice}$ is 0.031 cm$^{-3}$ (in comparison to 0.056 and 0.046 cm$^{-3}$ for
'all temperatures' and 'T-int weighted') in the temperature range between 224-242 K (note that, to
avoid representing mixed-phase clouds in the DADAR-Nice analysis, 242 K is chosen as maximum
temperature). The lowest $N_{ice}$ are found in the Arctic ('most frequent' median 0.016 cm$^{-3}$ between
230-242 K), which is most probably because the updrafts in the Arctic regions are generally lower
than in other regions.In Antarctica, the 'most frequent' median is higher (0.029 cm$^{-3}$ between 224
and 242 K). This could be traced back to stronger orographic uplifts in austral winter. The boreal and
austral mid-latitude cirrus clouds are similar to the global, with 'most frequent' medians of 0.030
and 0.031 cm$^{-3}$ in the temperature range of 227-242 K. This points to moderate updrafts on average,
though regionally orography and convection can cause higher updrafts and high $N_{ice}$, which is also
shown by Sourdeval et al. (2018).

The tropical cirrus clouds are different from those in the other regions. The 'most frequent' cirrus
have the highest median (0.074 cm$^{-3}$) and appear at much colder temperatures (197-221 K), i.e.
higher altitudes. This finding corresponds to the in-situ observations presented in Section 3.1 (see
Figure 3) and also the DARDAR-Nice analyses of the vertical distribution of $N_{ice}$ shown by Sour-
deval et al. (2018). The reason is the strong convection prevailing in tropical regions, lifting liquid
origin clouds up to high altitudes and cold temperatures while homogeneously nucleating a large
number of in-situ ice crystals in addition (see also Section 5).

For all regions, except the tropics, more than half of the cirrus clouds are found at temperatures
warmer than about 225 K and contain a considerable amount of liquid origin cirrus, as discussed in
the previous subsection. The same is true in the tropics, but for colder temperatures.





### 7  Summary and Conclusions

The Cirrus Guide II aims to represent cirrus clouds and their environment from the perspective of in-situ and satellite remote sensing observations. To this end, an in-situ data base is created, which is based on measurements with state-of the art instrumentation and extended in comparison to earlier studies (Schiller et al., 2008; Krämer et al., 2009; Luebke et al., 2013; Krämer et al., 2016). The in-situ data base consists now of about 168 hours of ice water content IWC, 99/88 hours of ice crystal number concentration $N_{ice}$ and $R_{ice}$, 116/320 hours of in-cloud and clear sky $RH_{ice}$ and, correspondingly, 320 hours of clear sky water vapor $H_2O$. The measurements span the altitude range between 5 and 20 km and cover the latitude band from 75 °North to 20 °South. The remote sensing data base includes 10 years (2006-2016) of global recording of ice crystal number concentrations $N_{ice}$. The main results form the evaluation of the two data sets are summarized in the following.

**1. Characteristics and distribution of in-situ origin and liquid origin cirrus** (Section 3)

From the extended Cirrus Guide II in-situ data set, we generally *confirm the typical characteristics of the cirrus types* introduced by Krämer et al. (2016) with some additions (see  Table 1):

- In-situ as well as liquid origin cirrus consist of two sub-classes determined by the updraft (1:slow updrafts - few large ice crystals form from heterogeneous freezing; 2:fast updrafts - many small ice crystals nucleate homogeneously).
  New in this concept is that also liquid origin cirrus differ in the two updraft regimes. Often, the two subclasses occur in succession and form a bimodal size distribution.

- Liquid origin cirrus are thicker (higher IWC) than in-situ cirrus and are usually characterized by larger ice crystals.

- The differences between the cirrus types are most pronounced in the formation phase of the clouds and are blurred with increasing lifetime due to ice crystal growth and sedimentation or additional ice formation.

In addition, we present a *picture of the distribution of cirrus with respect to altitude and latitude*, including an impression of the vertical structure of liquid origin and in-situ origin cirrus (Figure 3):

- Across all latitudes, the thicker liquid origin cirrus predominate at lower altitudes, while at higher altitudes the thinner in-situ cirrus prevail. In between, the two cirrus types overlap. This finding is in agreement with Luebke et al. (2016) and Wernli et al. (2016) for mid-latitude and Wolf et al. (2018) for Arctic cirrus.

Finally, a *first estimate of the radiative characteristics of typical idealized in-situ and liquid origin cirrus scenarios* is given (Figure 5):



- slow in-situ origin cirrus have a small optical depth ($\tau$: 0.001 - 0.05), resulting in a slight net warming effect of not larger than about 1.5 W/m$^2$.

- the optical depth of fast in-situ origin cirrus is larger ($\tau$: 0.05 - 1), but most of them are also warming (2-10 W/m$^2$). The thickest fast in-situ origin cirrus at the lowest altitudes can change the sign of their net forcing, they switch to a slight cooling effect.

- liquid origin cirrus have large optical depths ($\tau$: 1 - 12), and consequently exhibit a quite strong net cooling effect (-15 to -250 W/m$^2$).

**2. Cirrus and humidity in the tropical tropopause layer (TTL)** (Section 5)

The new in-situ data set is extended by observations in the tropical TTL outside, but also, for the first time, inside of the Asian monsoon anticyclone. Therefore, we put special emphasis on the analysis of the TTL environment.

*TTL cirrus clouds* (Figure 11):

- Two types of most likely in-situ formed cirrus are identified in slow large scale updrafts at low temperatures (T $\lesssim$ 205 K, $\Theta \gtrsim$ 355 K). The first is interpreted as young cirrus (N$_{ice}$ around 0.1-1 cm$^{-3}$ $\gtrsim$ 3 µm diameter), the second as aged cirrus (N$_{ice}$ around 0.001 cm$^{-3}$ $\gtrsim$ 20 µm diameter) where the smaller ice crystals have grown to larger sizes.

- The highest N$_{ice}$ (up to 30 cm$^{-3}$ around the cold point tropopause, CPT) and IWC (up to 1000 ppmv around the CPT) are found in deep convective systems in the Asian monsoon. Such systems represent massive liquid origin cirrus (very high IWC) superimposed by a fresh strong homogeneous in-situ ice nucleation event (many small ice crystals $\lesssim$ 20 µm) in fast updrafts.

*TTL humidity* (Figure 12):

- In the Asian monsoon, in-cloud and clear sky RH$_{ice}$ is higher (often supersaturated around and above the CPT) than in the surrounding tropics. This is caused by a higher amount of H$_2$O in the Asian monsoon (most frequently 3 to 5 ppmv) in comparison to the tropics outside (1.5 to 3 ppmv).

- Taking saturation at the stratospheric entry of an air mass as set point for water vapor transport to the stratosphere, the transport is underestimated by $\sim$ 10% in regions of weak convective activity (see also Rollins et al., 2016). In convective regions, the underestimation increased to 20-50% in our observations in the Asian monsoon.

- Convectively injected ice over the Asian monsoon CPT ($\Theta \gtrsim$ 400 K, RH$_{ice}$ $\lesssim$ 100%) can locally contribute a significant amount of water (up to an IWC of 8 ppmv, in comparison to only 2 ppmv in the surrounding tropics) that might be evaporated and further transported into the stratosphere.






### 3. In-situ and satellite climatologies (Sections 4 and 6)

*Cirrus Guide II in-situ cirrus and humidity climatologies*:

– The entire extended data set (Figure 6) is compared to the earlier studies listed above (Figure 7).

IWC: the median IWC and the core IWC band is the same in both data sets. showing that the in-situ IWC measurement techniques are robust and that the IWC is a stable parameter describing cirrus clouds.

$RH_{ice}$: the overall picture of the in-cloud and clear sky $RH_{ice}$ distributions has also not changed, demonstrating that high altitude water vapor measurements that were under discussion earlier have improved and stabilized.

$N_{ice}$: an extended view is presented for $N_{ice}$, which is due to the better lower $N_{ice}$ detection limit and a better mixture of dynamical situation. The new $N_{ice}$-T percentiles are lower and show no distinct temperature dependence (average 10%, median and 90% $N_{ice}$ percentiles: 0.002, 0.03

and 0.3 $cm^{-3}$) in comparison to the earlier observations, that show a slight decrease of $N_{ice}$ with temperature and an average median $N_{ice}$ of about 0.1 $cm^{-3}$.

– The in-situ data set is subdivided into mid-latitude and tropical climatologies (Figures 8 and 9) and typical cirrus and humidity characteristics of the respective regions are presented.

*DARDAR-Nice satellite global cirrus $N_{ice}$ climatology*
(Figures 14 and 15):

– A global climatology of $N_{ice}$ from 10 years (2006-2016) of satellite observations is provided that can be used for comparison with global models or other data sets.

The $N_{ice}$ from satellite observations are validated by and adjusted to in-situ measurements

from a subset of five campaigns of the Cirrus Guide II.

– The global median $N_{ice}$ from satellite observations is almost 2 times higher than the in-situ median and increases slightly with decreasing temperature.

– $N_{ice}$ medians sorted by geographical regions are highest in the tropics, followed by austral/boreal mid-latitudes, Antarctica and the Arctic.

– In the satellite climatologies of $N_{ice}$, half of the cirrus are located in the lowest, warmest cirrus layer and contain a significant amount of liquid origin cirrus. Their global median $N_{ice}$ is 0.031 $cm^{-3}$.

– Regarding the frequent appearance of liquid orign cirrus together with strong cooling effect is a motivation to investigate their influence on the overall cirrus radiative feedback on climate

in future studies.



**Acknowledgments**

This paper is dedicated to my colleague Cornelius Schiller, who passed away much too early in 2012. He was the initiator of the StratoClim project, including the tropical aircraft campaign. Through this campaign, he intended to complement his work on hydration and dehydration in the tropical tropopause layer, based on observations over Brazil, Australia and Africa (Schiller et al., 2009) with measurements over Asia. The campaign took place successfully out of Kathmandu, Nepal, in Summer 2017, after several years of planning and many hurdles that had to be overcome.





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

Contributions of Stratospheric Water Vapor to Decadal Changes in the Rate of Global Warming, SCIENCE,
327, 1219–1223, doi:10.1126/science.1182488, 2010.

Sourdeval, O., Gryspeerdt, E., Krämer, M., Goren, T., Delanoë, J., Afchine, A., Hemmer, F., and Quaas, J.:
Ice crystal number concentration estimates from lidar–radar satellite remote sensing – Part 1: Method and
evaluation, Atmospheric Chemistry and Physics, 18, 14 327–14 350, doi:10.5194/acp-18-14327-2018, 2018.

Spichtinger, P. and Cziczo, D. J.: Impact of heterogeneous ice nuclei on homogeneous freezing events, Journal
of Geophysical Research: Atmospheres, 115, D14208, doi:10.1029/2009JD012168, 2010.

Spichtinger, P. and Krämer, M.: Tropical tropopause ice clouds: a dynamic approach to the mystery of low
crystal numbers, Atmos. Chem. Phys., 13, doi:10.5194/acp-13-9801-2013, 2013.

Spreitzer, E. J., Marschalik, M. P., and Spichtinger, P.: Subvisible cirrus clouds – a dynamical system
approach, Nonlinear Processes in Geophysics, 24, 307–328, doi:10.5194/npg-24-307-2017, https://www.
nonlin-processes-geophys.net/24/307/2017/, 2017.

Thornberry, T., Rollins, A., Avery, M., Woods, S., Lawson, R., Bui, T., and Gao, R.-S.: Ice water content-
extinction relationships and effective diameter for TTL cirrus derived from in situ measurements during AT-
TREX 2014, Journal of Geophysical Research: Atmospheres, 122, 4494–4507, doi:10.1002/2016JD025948,
2017.

Thornberry, T. D., Rollins, A. W., Gao, R. S., Watts, L. A., Ciciora, S. J., McLaughlin, R. J., and Fahey, D. W.:
A two-channel, tunable diode laser-based hygrometer for measurement of water vapor and cirrus cloud ice
water content in the upper troposphere and lower stratosphere, Atmospheric Measurement Techniques, 8,
211–224, doi:10.5194/amt-8-211-2015, https://www.atmos-meas-tech.net/8/211/2015/, 2015.

Ueyama, R., Jensen, E. J., and Pfister, L.: Convective Influence on the Humidity and Clouds in the Tropical
Tropopause Layer During Boreal Summer, Journal of Geophysical Research: Atmospheres, 123, 7576–7593,
doi:10.1029/2018JD028674, 2018.

Urbanek, B., Groß, S., Schäfler, A., and Wirth, M.: Determining stages of cirrus evolution: a cloud classi-
fication scheme, Atmospheric Measurement Techniques, 10, 1653–1664, doi:10.5194/amt-10-1653-2017,
https://www.atmos-meas-tech.net/10/1653/2017/, 2017.

Urbanek, B., Groß, S., Wirth, M., Rolf, C., Krämer, M., and Voigt, C.: High Depolarization Ratios of Naturally
Occurring Cirrus Clouds Near Air Traffic Regions Over Europe, Geophysical Research Letters, 45, 13,166–
13,172, doi:10.1029/2018GL079345, 2018.

Vali, G., DeMott, P. J., Möhler, O., and Whale, T. F.: Technical Note: A proposal for ice nucleation termi-
nology, Atmospheric Chemistry and Physics, 15, 10 263–10 270, doi:10.5194/acp-15-10263-2015, https:
//www.atmos-chem-phys.net/15/10263/2015/, 2015.

Vogel, B., Günther, G., Müller, R., Grooß, J.-U., Afchine, A., Bozem, H., Hoor, P., Krämer, M., Müller, S.,
Riese, M., Rolf, C., Spelten, N., Stiller, G. P., Ungermann, J., and Zahn, A.: Long-range transport path-
ways of tropospheric source gases originating in Asia into the northern lower stratosphere during the Asian



monsoon season 2012, Atmospheric Chemistry and Physics, 16, 15 301–15 325, doi:10.5194/acp-16-15301-2016, https://www.atmos-chem-phys.net/16/15301/2016/, 2016.

1310 Voigt, C., Schumann, U., Minikin, A., Abdelmonem, A., Afchine, A., Borrmann, S., Boettcher, M., Bucuchholz, B., Bugliaro, L., Costa, A., Curtius, J., Dollner, M., Doernbrack, A., Dreiling, V., Ebert, V., Ehrlich, A., Fix, A., Forster, L., Frank, F., Fuetterer, D., Giez, A., Graf, K., Grooss, J.-U., Gross, S., Heimerl, K., Heinold, B., Hueneke, T., Jaervinen, E., Jurkat, T., Kaufmann, S., Kenntner, M., Klingebiel, M., Klimach, T., Kohl, R., Krämer, M., Krisna, T. C., Luebke, A., Mayer, B., Mertes, S., Molleker, S., Petzold, A., Pfeilsticker, K., Port, 1315 M., Rapp, M., Reutter, P., Rolf, C., Rose, D., Sauer, D., Schaefer, A., Schlage, R., Schnaiter, M., Schneider, J., Spelten, N., Spichtinger, P., Stock, P., Walser, A., Weigel, R., Weinzierl, B., Wendisch, M., Werner, F., Wernli, H., Wirth, M., Zahn, A., Ziereis, H., and Zöger, M.: ML-Cirrus the airborne experiment on natural cirrus and contrail cirrus with the high-altitude long-range research aircraft HALO, Bulletin of the American Meteorological Society, 98, 271–288, doi:10.1175/BAMS-D-15-00213.1, 2017.

1320 Wendisch, M., Yang, P., and Pilewskie, P.: Effects of ice crystal habit on thermal infrared radiative properties and forcing of cirrus, Journal of Geophysical Research: Atmospheres, 112, doi:10.1029/2006JD007899, 2007.

Wendisch, M., Poeschl, U., Andreae, M. O., Machado, L. A. T., Albrecht, R., Schlager, H., Rosenfeld, D., Martin, S. T., Abdelmomonem, A., Afchine, A., Araujo, A. C., Artaxo, P., Aufmhoff, H., Barbosa, H. M. J., Borrmann, S., Braga, R., Buchholz, B., Cecchini, M. A., Costa, A., Curtius, J., Dollner, M., Dorf, M., Dreil-1325 ing, V., Ebert, V., Ehrlich, A., Ewald, F., Fisch, G., Fix, A., Frank, F., Futterer, D., Heckl, C., Heidelberg, F., Hueneke, T., Jakel, E., Jarvinen, E., Jurkat, T., Kanter, S., Kaestner, U., Kenntner, M., Kesselmeier, J., Klimach, T., Knecht, M., Kohl, R., Koelling, T., Krämer, M., Krueger, M., Krisna, T. C., Lavric, J. V., Longo, K., Mahnke, C., Manzi, A. O., Mayer, B., Mertes, S., Minikin, A., Molleker, S., Munch, S., Nillius, B., Pfeilsticker, K., Pohlker, C., Roiger, A., Rose, D., Rosenowow, D., Sauer, D., Schnaiter, M., Schneider, J., 1330 Schulz, C., de Souza, R. A. F., Spanu, A., Stock, P., Vila, D., Voigt, C., Walser, A., Walter, D., Weigel, R., Weinzierl, B., Werner, F., Yamasoe, M. A., Ziereis, H., Zinner, T., and Zoeger, M.: ACRIDICON-CHUVA CAMPAIGN Studying Tropical Deep Convective Clouds and Precipitation over Amazonia Using the New German Research Aircraft HALO, BULLETIN OF THE AMERICAN METEOROLOGICAL SOCIETY, 97, 1885–1908, doi:10.1175/BAMS-D-14-00255.1, 2016.

1335 Wernli, H., Boettcher, M., Joos, H., Miltenberger, A. K., and Spichtinger, P.: A trajectory-based classification of ERA-Interim ice clouds in the region of the North Atlantic storm track, Geophysical Research Letters, 43, 6657–6664, doi:10.1002/2016GL068922, https://agupubs.onlinelibrary.wiley.com/doi/abs/10.1002/2016GL068922, 2016.

Wolf, V., Kuhn, T., Milz, M., Voelger, P., Krämer, M., and Rolf, C.: Arctic ice clouds over northern Swe-1340 den: microphysical properties studied with the Balloon-borne Ice Cloud particle Imager B-ICI, Atmospheric Chemistry and Physics, 18, 17 371–17 386, doi:10.5194/acp-18-17371-2018, 2018.

Wolf, V., Kuhn, T., and Krämer, M.: On the dependence of cirrus parametrizations on the cloud origin, Geophysical Research Letters, 46, 12 565–12 571, doi:10.1029/2019GL083841, 2019.

Woods, S., Lawson, R. P., Jensen, E., Bui, T. P., Thornberry, T., Rollins, A., Pfister, L., and Avery, M.: Micro-1345 physical Properties of Tropical Tropopause Layer Cirrus, Journal of Geophysical Research: Atmospheres, 123, 6053–6069, doi:10.1029/2017JD028068, 2018.



Zondlo, M. A., Paige, M. E., Massick, S. M., and Silver, J. A.: Vertical cavity laser hygrometer for the National
Science Foundation Gulfstream-V aircraft, Journal of Geophysical Research: Atmospheres, 115, D02309,
doi:10.1029/2010JD014445, 2010.

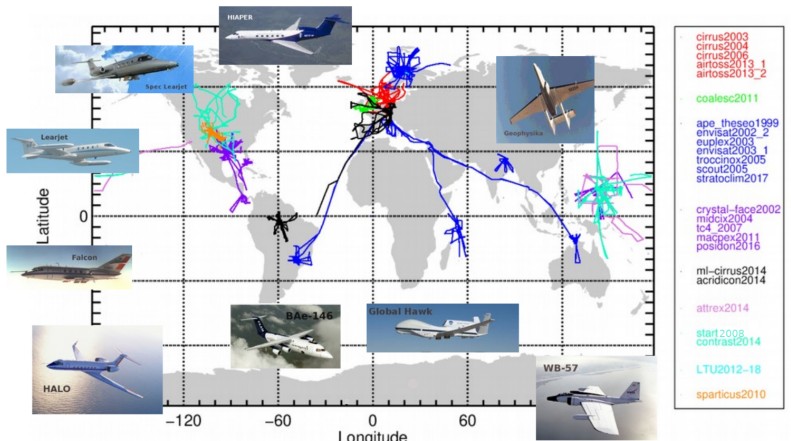

**Figure 1.** Aircraft flight paths during the 24 campaigns listed in Table 5. 185 flights, 192 h IWC measurements, 90 h $N_{ice}$, 84 h $R_{ice}$, 116/331 h $RH_{ice}$ in/outside cirrus (campaign names: red – GfD Learjet, green – BAe-146, blue – Geophysica, purple – WB-57, black – HALO, light-purple – Global Hawk, cyan – GV HIAPER, orange – Spec Learjet). In comparison: Schiller et al. (2008) / Krämer et al. (2009) – 52 flights, 27 h IWC measurements, 8.5 h $N_{ice}$, 8.5 h $R_{ice}$, 10/16 h $RH_{ice}$ in/outside cirrus.





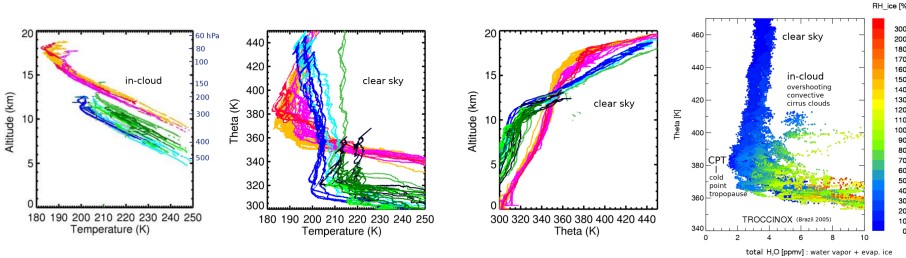

**Figure 2. Left panel:** Temperature vs. altitude inside of cirrus clouds (adopted from Schiller et al., 2008), **middle left panel:** Temperature vs. potential temperature $\Theta$, **middle right panel:** clear sky $\Theta$ vs. altitude. Plotted are 28 flights from the data base, with blueish colors for Arctic, greenish colors for mid-latitude and reddish colors for tropical observations. **right panel:** Total $H_2O$ (= water vapor + evaporated ice crystals) vs. $\Theta$ for the tropical field campaign TROOCINOX (Brazil, 2005), color coded by total $H_2O$ expressed as $RH_{ice}$ (adopted from Schiller et al., 2009).



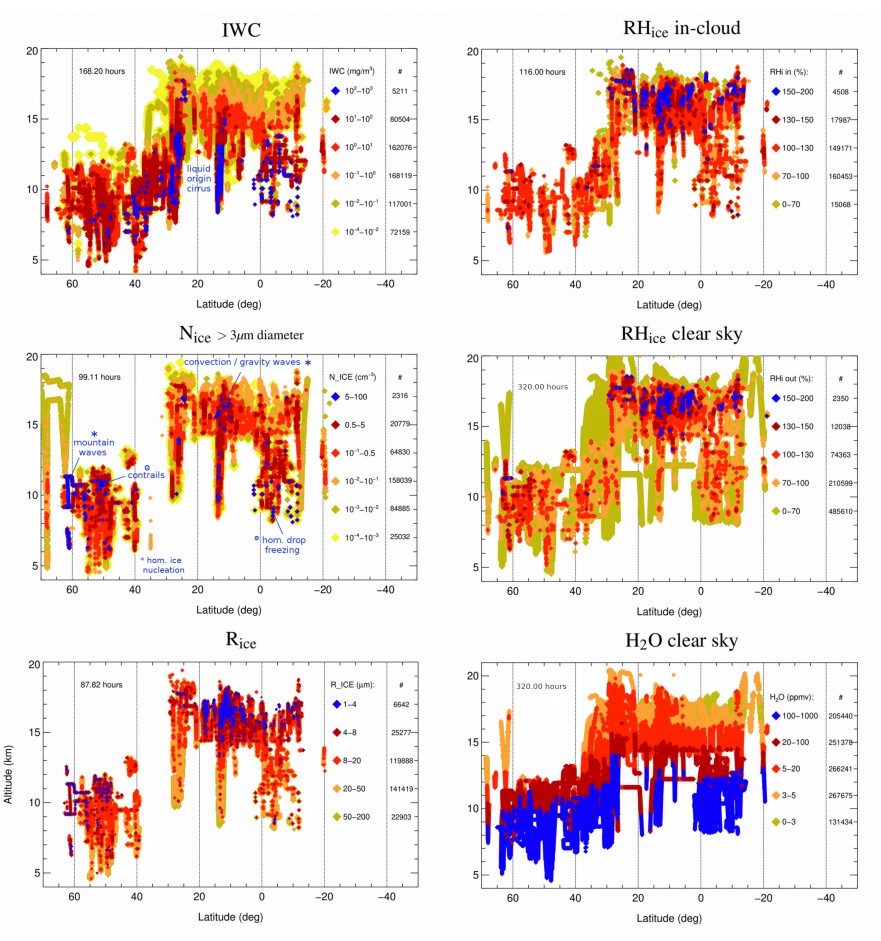

**Figure 3.** Cirrus cloud distribution with latitude and altitude of **Left column:** Ice water content (IWC, top panel), ice crystal number ($N_{ice}$, size $> 3~\mu$m diameter, middle panel) and mean mass radius ($R_{ice}$, calculated from IWC/$N_{ice}$, bottom panel). **Right column:** In-cloud and clear sky relative humidity wrt ice (RH$_{ice}$, top and middle panel), and clear sky water vapor volume mixing ratio (H$_2$O, bottom panel). The field campaigns are listed in Table 4, data evaluation methods and detection ranges of the parameters are described in Appendix A. Note that the data points are plotted in the order of the colors from yellow to blue; frequencies of occurence of the parameters can be seen in Figure 6.





| Typical characteristics of cirrus types in the initial stage | | | | |
|---|---|---|---|---|
| ORIGIN | IWC | $N_{ice}$ | $R_{ice}$ | weather system |
| *slow updraft* (mostly *heterogeneous ice nucl.*) | | | | |
| IN-SITU | low | few | large | frontals systems (WCBs) |
| LIQUID | high | more | larger | |
| *fast updraft* (mostly *homogeneous ice nucl.*) | | | | |
| IN-SITU | high | many | small | gravity waves, convection |
| LIQUID | high | more | small & larger | |

IWC high/low: above/below the IWC median; $N_{ice}$ few/more/many: below/in-between/above the 10 and 90% $N_{ce}$ percentiles (see Figure 5).

$R_{ice}$ small/large/larger: ice particles $\lesssim$ 20µm dominate the PSD / ice particles $\gtrsim$ 20µm dominate the PSD, max. size several hundred µm diameter / ice particles $\gtrsim$ 20µm dominate the PSD, max. size up to thousand µm diameter (PSD: particle size distribution).

**Table 1.** Typical characteristics of cirrus types in the initial stage.

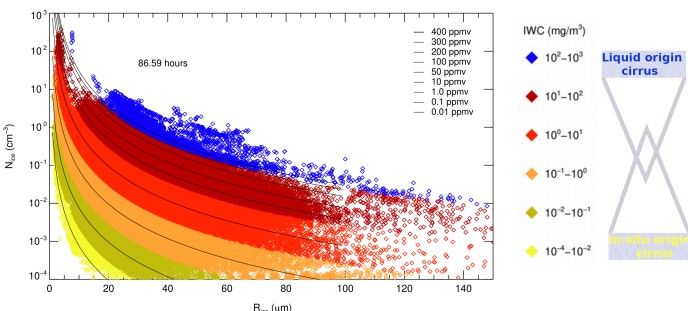

**Figure 4.** Relation between ice crystal concentration $N_{ice}$ and mean mass radius $R_{ice}$, color coded by the ice water content IWC in mg/m$^{-3}$, from ~87 h of cirrus cloud observations ($R_{ice}$ is calculated by dividing IWC/$N_{ice}$). The thin black lines are isolines of IWC in ppmv (in the order of the legend). The scheme at the right side illustrates the partitioning of the clouds between liquid and in-situ origin: the thickest cirrus (blue points) are of liquid origin, the thinnest (yellow points) of in-situ origin. As the thickness decreases, the portion of liquid origin cirrus becomes smaller and smaller while more and more in-situ origin cirrus appear.





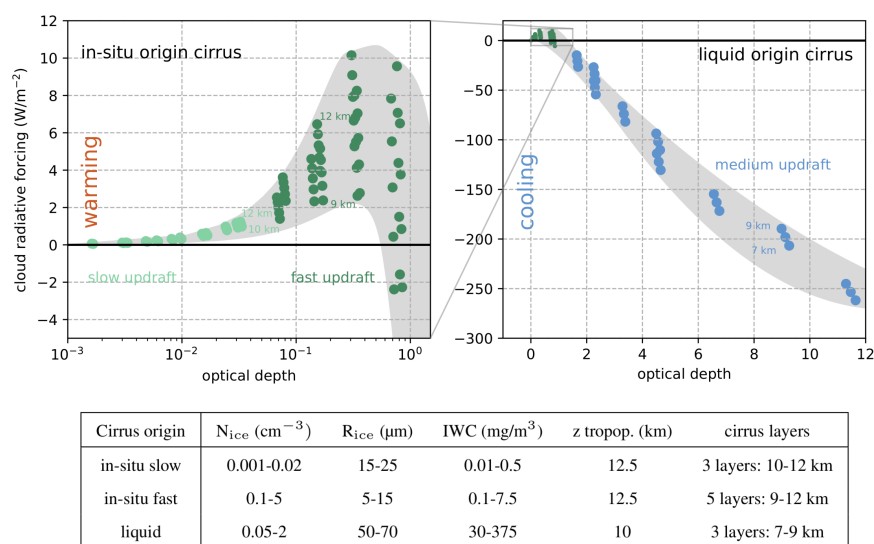

| Cirrus origin | $N_{ice}$ (cm$^{-3}$) | $R_{ice}$ (µm) | IWC (mg/m$^3$) | z tropop. (km) | cirrus layers |
|---|---|---|---|---|---|
| in-situ slow | 0.001-0.02 | 15-25 | 0.01-0.5 | 12.5 | 3 layers: 10-12 km |
| in-situ fast | 0.1-5 | 5-15 | 0.1-7.5 | 12.5 | 5 layers: 9-12 km |
| liquid | 0.05-2 | 50-70 | 30-375 | 10 | 3 layers: 7-9 km |

**Figure 5.** Simulated radiative forcing versus optical depth for exemplary in-situ slow and fast updraft (light and dark green dots) as well as liquid (blue dots) origin cirrus; the idealized scenarios are summerized in the table.

Description of the idealized scenarios: a temperature profile is prescribed representative for midlatitude conditions, the surface temperature is set to 288 K (at p = 1000 hPa, p exponentially decreases with height). The vertical profile of relative humidity over ice is also establish, with a saturated layer of 1000 m thickness, constant subsaturation below the layer, and a strong decrease in humidity towards the stratosphere. A cirrus cloud with constant ice mass and number concentration is placed in the saturated layer. From the measurements (see Figure 4), typical values of $N_{ice}$, $R_{ice}$ and IWC are chosen for the three cirrus types (see table above). The vertical profiles are adjusted with respect to tropopause height (z tropop.) and placement of the cirrus layers in accordance with the cirrus types. For the radiative transfer calculations, the well-known two-stream radiative transfer model for ice particles by Fu and Liou (1993), with 6 bands in the solar and 12 bands in the thermal infrared regime, is used. The simulations are realized for a geographic latitude of $\varphi = 50°$, solar surface albedo of 0.3, infrared surface emissivity of 1 and solar constant S = 1340 W/m$^2$; we assume equinox conditions (e.g. end of March) at local time t = 12 h (for these settings, we use the modified model of Joos et al., 2014). The net cloud radiative forcing (CRF) is calculated using the fluxes at the top of atmosphere in the short wave and long wave ranges in comparison with a clear sky case.



# CIRRUS GUIDE II – CLIMATOLOGIES

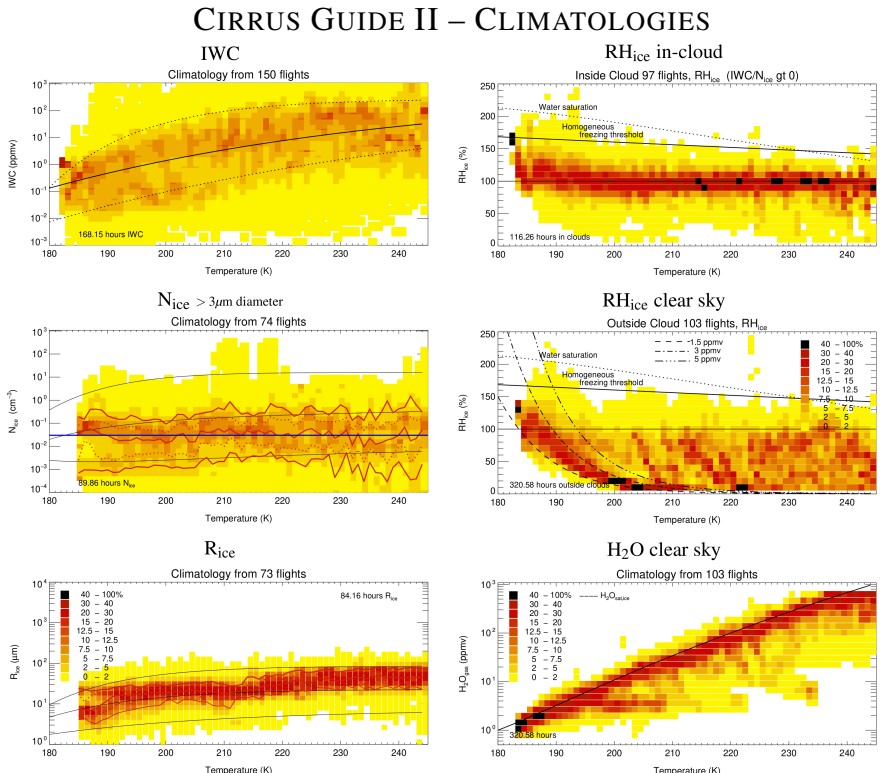

**Figure 6.** Frequencies of occurrence in dependence on temperature, binned in 1K intervals of: Ice Water Content (**IWC**; top left panel; black solid/dotted lines: median, min/max IWC of the core IWC band of Schiller et al. (2008)). Ice crystal number ($N_{ice}$, size $> 3$ $\mu$m diameter, middle left panel; red lines: 10, 25, 50, 75, 90% $N_{ice}$ percentiles; blue line: fit through median $N_{ice}$ values (median $N_{ice}$: 0.03 cm$^{-3}$; 10, 25, 75, 90% percentiles: 0.002, 0.007, 0.102, 0.3 cm$^{-3}$); black lines: minimum/middle/maximum $N_{ice}$ of Krämer et al. (2009)). Mass mean radius (**$R_{ice}$**: calculated from IWC/$N_{ice}$, bottom left panel; red lines: 10, 25, 50, 75, 90% $R_{ice}$ percentiles; black lines: minimum/middle/maximum $R_{ice}$ of Krämer et al. (2009)). In-cloud and clear sky relative humidity wrt ice (**RH**$_{ice}$, top and middle right panel), and water vapor volume mixing ratio (**$H_2O$**, bottom right panel). The field campaigns included in the data analysis are listed in Table 5. Data evaluation methods and detection ranges of the parameters are described in Appendix A. Note that for $N_{ice}$ -and thus $R_{ice}$- the hours spent in clouds is less than in  Figure 3. The reason is that for the calculation of data frequency distributions only measurements covering the same detection range are used (see also Appendix A2).



SCHILLER ET AL. (2008) & KRÄMER ET AL. (2009)

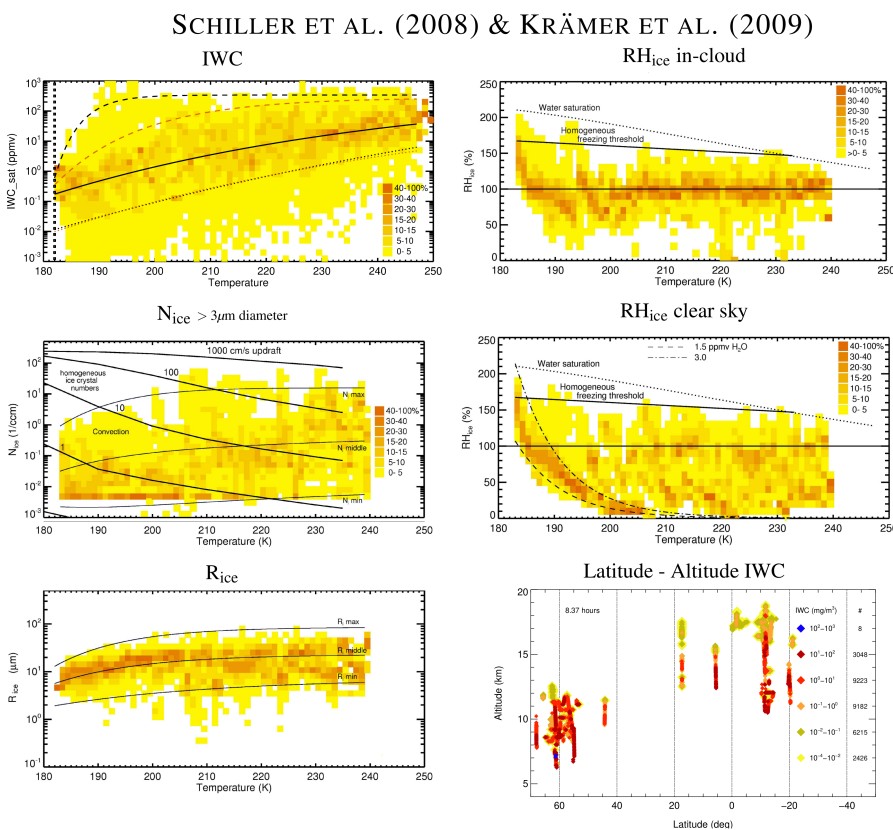

**Figure 7.** Frequencies of occurrence in dependence on IWC, $N_{ice}$, $R_{ice}$ and $RH_{ice}$ in cirrus and clear sky, binned in 1K intervals. The plots are adapted from Schiller et al. (2008) and Krämer et al. (2009); included field campaigns: Ape Theseo 1999 (Geophysica), Envisat 2002-2 (Geophysica), Envisat 2003-1 (Geophysica), Envisat 2003-2 (Geophysica), Euplex 2003 (Geophysica), Cirrus 2003 (GFD Learjet), Cirrus 2004 (GFD Learjet), Scout 2005 (Geophysica), Troccinox 2005 (Geophysica), Cirrus 2006 (GFD Learjet) (see also Table 5). In addition, the distribution of IWC (color code) with latitude and altitude is shown.


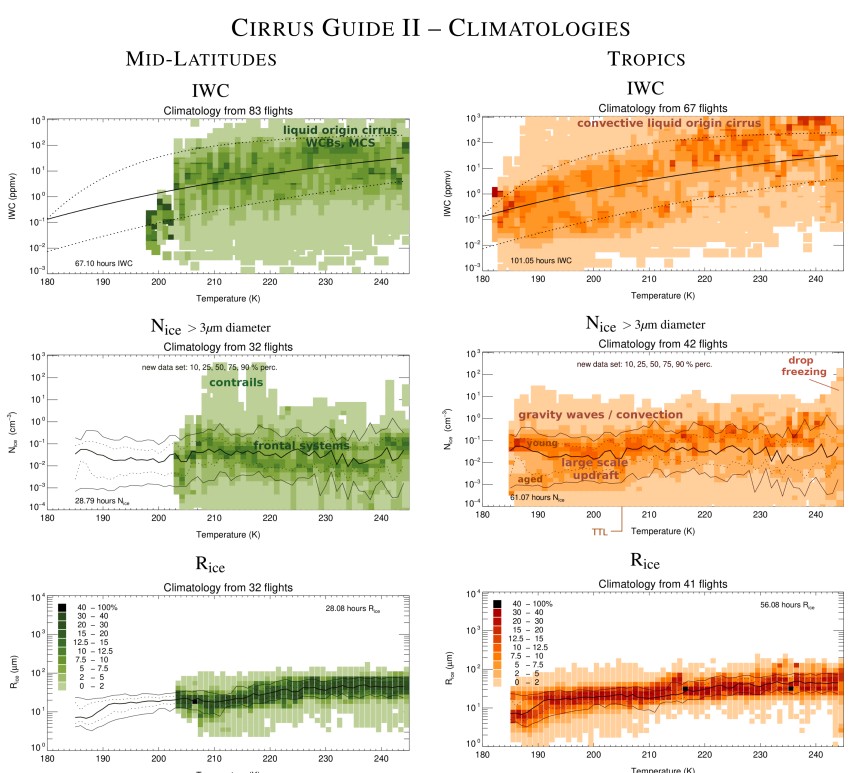

**Figure 8.** Same as Figure 6 (left column), but for mid-latitudes and tropics (WCBs: Warm Conveyor Belts; MCS: Mesoscale Convective Systems; TTL: Tropical Tropopause Layer). The field campaigns are listed in Table 5.



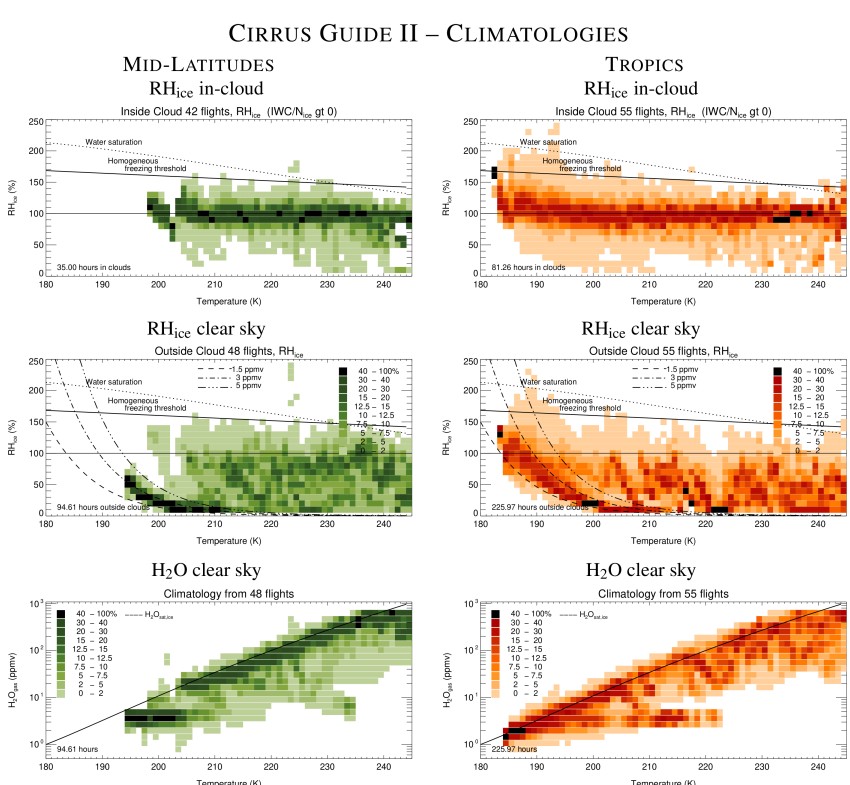

**Figure 9.** Same as Figure 6 (right column), but for mid-latitudes and tropics. The field campaigns are listed in Table 5.



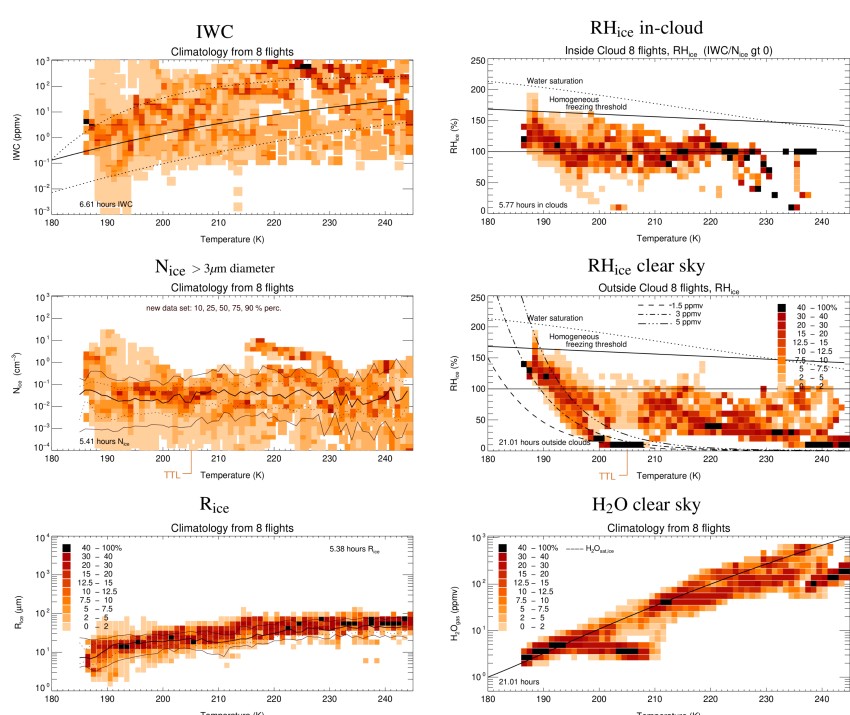

**Figure 10.** Same as Figure 6, but for the field campaign StratoClim 2017 above the Asian monsoon.



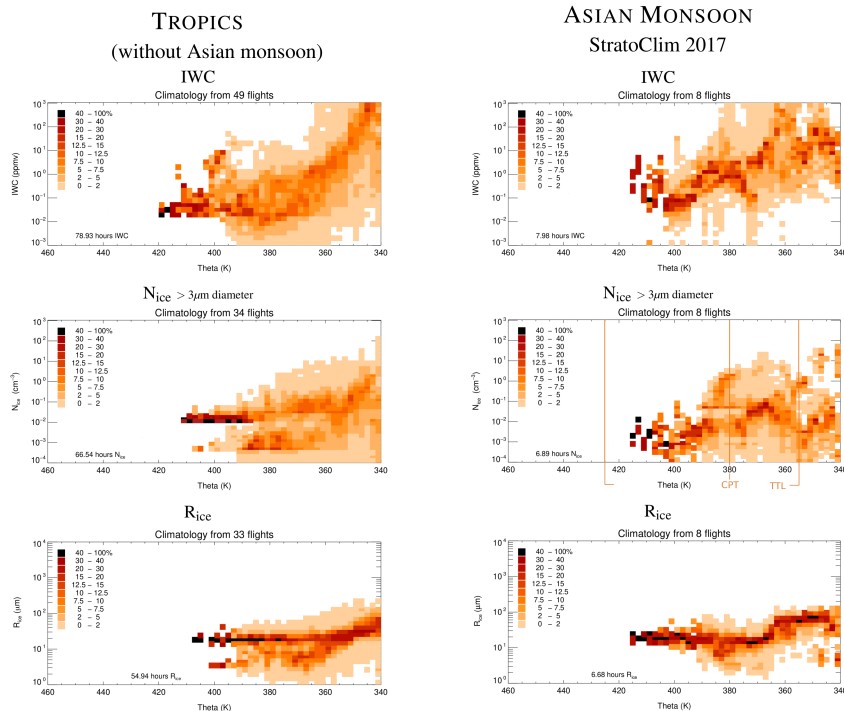

**Figure 11.** Climatology of cirrus clouds in dependence on potential temperature Θ (left panels: tropics without Asian monsoon, right panels: Asian monsoon). In the middle panel, the range of the TTL (after Fueglistaler et al., 2009) and the cold point temperature (CPT, derived from the observed temperature profiles) are marked; the corresponding altitude range is ~14 to 20 km. Note that in the tropics 355–330 K ≈ 235–275 K (-38–0 C) and that the TTL spans from about 420–355 K. The field campaigns are listed in Table 5.



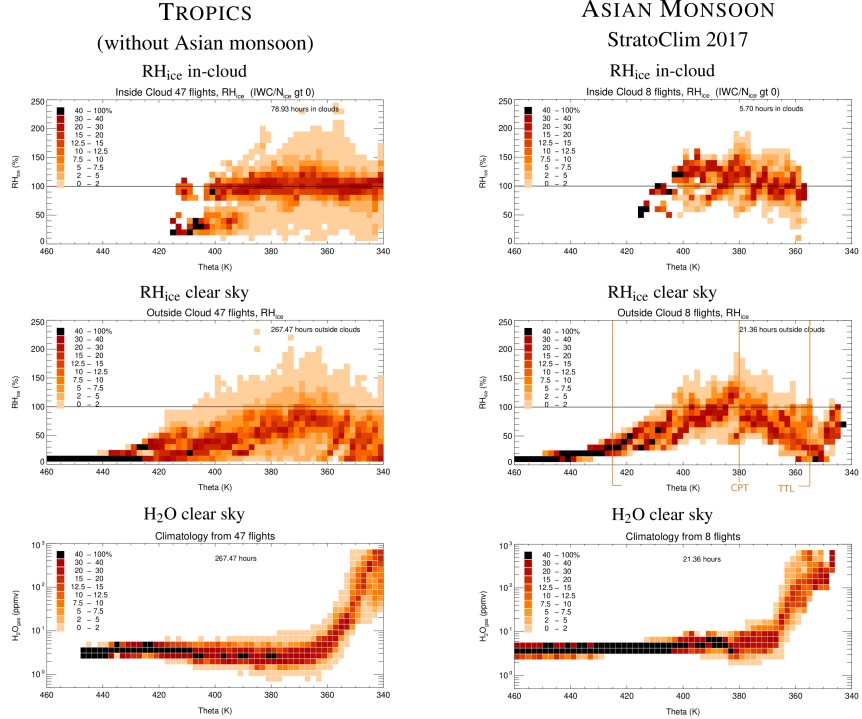

**Figure 12.** Climatology of humidity in dependence on potential temperature Θ (left panels: tropics without Asian monsoon, right panels: Asian monsoon). In the middle panel, the range of the TTL (after Fueglistaler et al., 2009) and the cold point temperature (CPT, derived from the observed temperature profiles) are marked; the corresponding altitude range is ∼14 to 20 km. Note that in the tropics 355–330 K ≈ 235–275 K (-38–0 C) and that the TTL spans from about 420–355 K. The field campaigns are listed in Table 5.



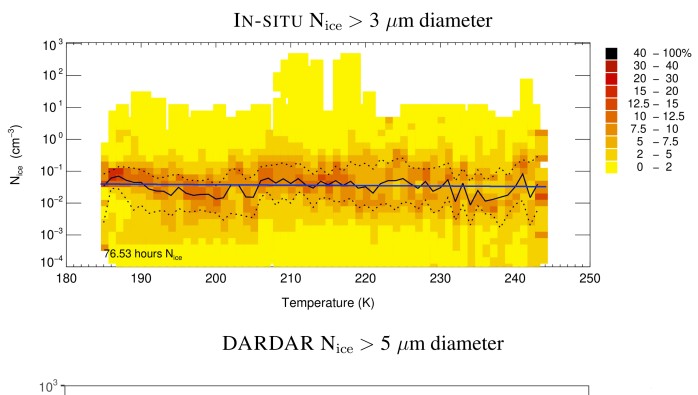

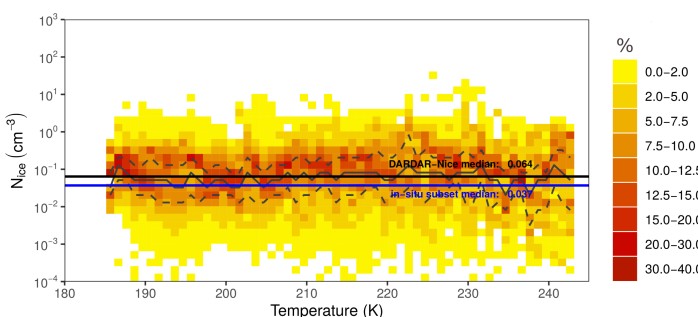

**Figure 13.** $N_{ice}$ – temperature climatologies from (top panel) in-situ measurements during the field campaigns COALESC2011, ATTREX2014, ACRIDICON2014, ML-CIRRUS2014 and STRATOCLIM2017, (bottom panel) satellite remote sensing, applying the algorithm DARDAR-Nice $f(N_0^\star, D_m)$ to $N_0^\star$, $D_m$ derived from the in-situ observations. The black contours represent the 25, 50 and 75th percentiles. Straight black line: DARDAR-Nice average median $N_{ice}$ (0.064 cm$^{-3}$), straight blue line: in-situ average median $N_{ice}$ (0.037 cm$^{-3}$). The excess of $N_{ice}$ by a factor of 1.73 in DARDAR is caused by the retrieval method, for more information see text.

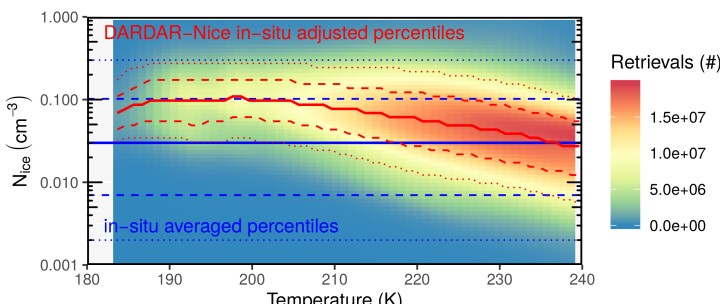

**DARDAR $N_{ice} > 5$ $\mu$m diameter**

10 years climatology (2006 – 2016)

**Figure 14.** $N_{ice}$ climatology from 10 years satellite remote sensing observations using DARDAR-Nice. The data was collected twice a day, at midday and at midnight. Thick black lines: 25, 50 and 75[th] percentiles; thin plain cyan line: DARDAR-Nice median $N_{ice}$; thin dotted cyan line: DARDAR-Nice median $N_{ice}$ adjusted to the offset factor of 1.73 between DARDAR-Nice and the in-situ observations; blue line: in-situ average median $N_{ice}$ (from Figure 6. Temperature intervals containing less than 10 retrievals were excluded (results non-sensitive to exact threshold).

**Figure 15.** Percentiles of the $N_{ice}$ - temperature climatologies. Red: global DARDAR-Nice percentiles adjusted to the in-situ observations (by the offset factor of 1.73, see Figure 12). The 10/90[th], 25/75[th] and 50[th] percentiles are indicated by dotted, dashed and plain lines, respectively. Blue: similarly, in-situ average percentiles. The solid lines denote the 25, 50 and 75% percentiles recommended for intercomparison with other data sets (the corresponding numbers are listed in Table 2). The shaded background indicates the occurrence of retrievals, i.e. cirrus clouds. The cirrus appear most often at higher temperatures above about 225 K.





| Temperature (K) | in-situ adj. DARDAR-Nice $N_{ice}$ (cm$^{-3}$) | | | | |
|---|---|---|---|---|---|
| | 10% perc. | 25% perc. | median | 75% perc. | 90 % perc. |
| 180-190 | 0.034 | 0.051 | 0.087 | 0.149 | 0.243 |
| 190-200 | 0.033 | 0.055 | 0.010 | 0.173 | 0.274 |
| 200-210 | 0.028 | 0.051 | 0.091 | 0.163 | 0.265 |
| 210-220 | 0.017 | 0.034 | 0.068 | 0.133 | 0.239 |
| 220-230 | 0.011 | 0.022 | 0.049 | 0.102 | 0.195 |
| 230-240 | 0.007 | 0.015 | 0.033 | 0.068 | 0.130 |
| Temperature all | In-situ $N_{ice}$ (cm$^{-3}$) | | | | |
| | 10% perc. | 25% perc. | median | 75% perc. | 90 % perc. |
| | 0.002 | 0.007 | 0.03 | 0.102 | 0.3 |

**Table 2.** DARDAR-Nice (adjusted to the in-situ observations by the offset factor of 1.73, see Figure 13) and in-situ observation $N_{ice}$ percentiles for intercomparison with global models or other data sets (see Figure 15).

| Region | in-situ adj. DARDAR-Nice $N_{ice}$ (cm$^{-3}$) medians | | | |
|---|---|---|---|---|
| | all Temp. | T-int weighted | most frequent | T-range |
| Arctic | 0.036 | 0.023 | 0.016 | 230-242 K |
| Mid-lat North | 0.057 | 0.040 | 0.030 | 227-242 K |
| Tropics | 0.070 | 0.067 | 0.074 | 197-221 K |
| Mid-lat South | 0.057 | 0.043 | 0.031 | 227-242 K |
| Antarctica | 0.050 | 0.045 | 0.029 | 222-242 K |
| Global | 0.056 | 0.046 | 0.031 | 224-242 K |

**Table 3.** DARDAR-Nice median $N_{ice}$ (adjusted to the in-situ observations by the offset factor of 1.73, see Figure 13) for five regional latitude bands: Arctic: 90°– 67.7°, Mid-lat North: 67.7°– 23.3°, Tropics: 23.3°– -23.3°, Mid-lat South: -23.3°– -67.7°, Antarctica: -67.7°– -90° as well as global median $N_{ice}$. All Temp.: median without temperature binning; T-int weighted: median using the relative occurrence of each temperature interval as a weight; most frequent: median considering those temperature intervals that contain 50% of the total $N_{ice}$ occurrence, T-range denotes the respective minimum and maximum temperatures.





## Appendix A: Methods

### A1 Field campaigns and instrumentation

The field campaigns, locations and deployed instruments are listed in Table 4, which is an extended version of the respective Table in the Cirrus Guide: Part I. For a brief description of the instruments already introduced there, we refer to Krämer et al. (2016). Other instruments used during START 2008, CONTRAST 2014, ATTREX 2014, POSIDON 2016 and LTU 2012-2018 are briefly introduced here.

ATTREX 2014 and POSIDON 2016: IWC and $RH_{ice}$ are measured with the dual channel NOAA-$H_2O$ instrument, a dual channel TDL hygrometer which detects both $H_2O_{tot}$ (= $H_2O_{gas}$ + IWC) and $H_2O_{gas}$ by using a forward and a backward directed inlet. The detection limit of the IWC is 0.03 ppmv ($\approx$ 0.003 mg/m$^3$). Cloud particle size distributions (PSDs) and thus $N_{ice}$ are recorded by a FCDP and a 2D-S. The size range of the FCDP is 1–50 μm diameter and the 2D-S counts cloud particles between 25–3005 μm.

Start-08 and CONTRAST: IWC and $N_{ice}$ are both derived from PSD measurements by a CDP and a 2D-C. The size range of the CDP is 2–50 μm diameter and the 2D-C counts cloud particles between 60–1100 μm.

LTU 2012-2018: IWC and $N_{ice}$ are both derived from PSD measurements by the Balloon-borne Ice Cloud particle Imager (B-ICI), detecting ice particle sizes between 10 and 1200 μm diameters. Note that the measurements are not from aircraft, but from balloons launched at Kiruna, Sweden.

### A2 Data evaluation methods

The data evaluation methods, detection ranges and data quality criteria of the measurements of IWC, $N_{ice}$, $R_{ice}$, $RH_{ice}$ and $H_2O$ are described in the following. The instruments measuring the respective parameters are listed in Table 4, the estimated uncertainties are mentioned in the Table caption. For the calculation of frequency distributions of the data, only those campaigns where the measured parameters cover the same detection range are used, since mixing of data with differing detection ranges influences the calculation of percentiles. Thus, the list of analyzed campaigns varies for each parameter, since not all parameters are available for the same detection ranges in all campaigns, or the data quality is considered as suspicious. In Table 5 the available parameters are listed for each campaign. The resulting data coverage of the parameters over the entire temperature range is shown in Figure 16.





### A2.1 Ice water content (IWC)

In the earlier studies of Schiller et al. (2008) and Luebke et al. (2013), the IWC was detected as
difference between total water ($H_2O_{tot}$: water vapor + evaporated ice crystals) and gas phase water
($H_2O_{gas}$); this product is named here IWC($H_2O$). Krämer et al. (2016) also determined the IWC by
integrating the ice particle size distributions (PSDs) measured by cloud spectrometers in cases where
no total water measurements were available; we call this IWC(PSD). A detailed description of both
methods is given in Afchine et al. (2018). Also, Afchine et al. (2018) demonstrated good agreement
between the two methods under reliable sampling conditions. Agreement between IWC($H_2O$) and
IWC(PSD) is also a criterion for the quality of the IWC data (see also Thornberry et al., 2017), as
well as for the measurements of $H_2O$ and the PSDs (for the data quality of $H_2O$ and $N_{ice}$ see also the
following subsections).

Unlike in the earlier studies, we here combine the two methods, IWC($H_2O$) and IWC(PSD). In
case both measurements are available, IWC($H_2O$) is used as first choice for each flight second. But
if IWC($H_2O$) is zero or missing, IWC(PSD) is used. This combination of IWCs can fill times of
measuring failures or signals beyond detection limits of one instrument and also compensates for
some of the inhomogeneities in cirrus. If only one instrument is available, the respective IWC is
taken.

As discussed by Schiller et al. (2008) and Afchine et al. (2018), the lower detection limit of
IWC($H_2O$) depends on the temperature and can not be clearly defined. Data points below the lower
dotted line drawn in the IWC-T parameter space in Figure 16 (upper left panel) represent obser-
vations where the difference between both the $H_2O_{tot}$ and $H_2O_{gas}$ measurements is not significant
to unambiguously identify a cloud. This is where the ratio $H_2O_{enh}/H_2O_{gas} < 1.07$; note here that
$H_2O_{enh}$ is the originally measured total water quantity, which is enhanced due to the sampling char-
acteristics of the system (for more detail see Schiller et al., 2008; Afchine et al., 2018). The lower
detection limit of IWC(PSD) is 0.05 ppmv ($\approx$ 0.005 mg/m$^{-3}$) for NIXE-CAPS and 0.01 ppmv ($\approx$
0.001 mg/m$^{-3}$) for the combination of FCDP and 2D-C.

### A2.2 Ice crystal number ($N_{ice}$)

From the campaigns where cloud particle size distributions (PSDs) are measured, we chose those
where cloud particles between 3 and $\sim$1000 μm diameter are recorded when calculating $N_{ice}$ (see
Table 5). We calculate $N_{ice}$ as the sum of ice crystal concentrations over all size bins larger than
3 μm on a 1 Hz time resolution, because this time span represents around 200 m flight distance
- which might already not resolve cirrus inhomogeneities. Thus, when averaging $N_{ice}$ over longer
flight times, cloud free segments might influence the ice crystal concentrations.

Two instruments are needed to be deployed to cover the complete cirrus ice particle size range,
one for smaller and the other for larger cloud particles. The PSDs from NIXE-CAPS (CAS-Depol
+ CIPgs) and FCDP + 2D-S are merged between 20 and 25 μm to avoid overlap of particle sizes,



those of CDP + 2D-C at 55 μm. In earlier campaigns where only FSSP measurements (3–30 μm) are
available, we calculate $N_{ice}$ without the larger particles, since they contribute only a negligible part
to $N_{ice}$ (these campaigns are marked as y* in Table 5). However, for the calculation of occurrence
frequencies only the campaigns covering the whole size range are used.

*Concentrations and frequencies of occurrence of small ice crystals: the effect of limited sampling
volumes*

$N_{ice}$ is given by the number of cloud particle events recorded in the cloud volume that is sampled.
Thus, the lowest concentration that can be detected is when only 1 particle is recorded. For the
particle imaging probes (CIPgs, 2D-S, 2D-C, used size range $\sim$ 20-1000 μm) that results to $\sim 10^{-4}$
$cm^{-3}$, while for the light scattering probes (CAS-Depol, FCDP, CDP, used size range $\sim$ 3-20 μm)
it reaches the higher value of $\sim$0.015 $cm^{-3}$. The difference in the recorded concentration range
is caused by the differing sampling volumes of the two instrument types. As a consequence, $N_{ice}$
concentrations $\lesssim$ 0.015 $cm^{-3}$ contain only ice particles $\gtrsim$ 20 μm. Lower concentrations of smaller
particles can not be detected with current particle measurement techniques (see also Krämer et al.,
2016; Baumgardner et al., 2017).

Figure 17 (left panel) shows exemplarily $N_{ice}$ in dependence on temperature, measured with
NIXE-CAPS (NIXE-CAPS combines a CAS-Depol and a CIPgs) during the field campaign ML-
Cirrus 2014, color coded by their frequencies of occurrence. CAS-DPOL samples 47.5$cm^{-3}$@190m/s
aircraft cruising speed in 1 second while CIPgs probes about 1000-16000$cm^{-3}$ in dependence on
particle size in the same time interval and aircraft speed (Costa et al., 2017). Thus, one particle
event in the CAS-DPOL corresponds to a higher concentration as in the CIPgs. The red line in the
left panel of Figure 17 represents the lowest detectable $N_{ice}$ of CAS-DPOL and the blue line the
same for CIPg. A notable feature in the figure is that directly above the lowest detectable $N_{ice}$ of
CAS-DPOL (red line) the $N_{ice}$ frequencies jump to a higher level than below. The reason for this
peak is an effect caused by the small sampling volumes of these instruments, which is often called
'bad statistics'. It concerns instruments whose lowest $N_{ice}$ detection limit is above the naturally oc-
curring particle concentrations. When clouds with concentrations smaller than the lowest detectable
$N_{ice}$ are probed, these concentrations appear in the sampling volume as 'single particles events' (in
the size range $\lesssim$ 20 μm) and are assigned to the minimum detectable concentration of the respective
instrument, since the occurrence of one particle per time unit can not be related to the true larger air
volume it belongs to. Therefore, the data sets of this study are cleared from the artificial concentra-
tion feature at the instrument's detection limit by excluding $N_{ice}$ caused by one single particle in the
respective instruments.

The effect of this correction on the occurrence frequencies can be seen in Figure 17 (right panel),
which shows the same data set as the left panel, but with 'single particle events' removed: the sharp
edge at the lowest detectable $N_{ice}$ of the CAS-DPOL has disappeared, instead more lower concen-





trations from the CIPg become visible, which were hidden by the artificial concentrations from the CAS-DPOL 'single particle events' before.

'Single particle events' are less common if the measurements are averaged over larger time periods. The considered time interval can be adapted according to the required spatial resolution of the measurements. However, as outlined above, for this study we choose to show the lowest time interval (1 s) to achieve the highest possible spatial resolution.

*Ice crystal shattering*

Fragmentation of large ice crystals at the cloud probes housings distorted a correct recording of the ice crystal concentrations in earlier times. Nowadays, new inlets and also postprocessing algorithms based on the interarrival times of the crystals in the sampling volume (Korolev and Field, 2015) have minimized this effect. Nevertheless, older data sets might be contaminated by small artifacts of shattered ice particles and needs to be marked in the data set and not used in the data analyses. Shattering can be recognized when looking at the $N_{ice}$ frequencies of occurrence plotted versus the temperature, as can be seen in Figure 18, where an example of a data set including shattered ice crystals is shown. Clearly, high frequencies of occurrence appear at $N_{ice}$ concentrations between 10-100 cm$^{-3}$ for all temperatures. This ice particle 'mode' is not present in data sets not influenced by shattered ice crystals, as can be seen when comparing Figures 18 and 17.

### A2.3 Water vapor and Relative Humidity wrt ice (H$_2$O and RH$_{ice}$)

Water vapor measurements where, as ice crystal concentrations, under discussion in the last decade (Peter et al., 2006). After some efforts to improve the quality of the measurements (e. g. Fahey et al., 2014), the water vapor instruments today provide high-precision data (see Rollins et al., 2014; Meyer et al., 2015; Thornberry et al., 2015; Buchholz et al., 2013; Kaufmann et al., 2018).

Nevertheless, we check the H$_2$O data quality for each campaign, including the recent ones, best by comparison of several H$_2$O instruments as done by e.g. Kaufmann et al. (2018). In case this is not possible, the frequencies of occurrence of in-cloud RH$_{ice}$ are used to evaluate the quality of the measurements: inside of cirrus clouds, at temperatures $\gtrsim$ 200 K, the frequencies should center around saturation, as can be seen in Figure 6 (upper right panel). Figure 19 gives an example of a data set with a bias in the in-cloud RH$_{ice}$. We should note that such a distortion might also be caused by a shift in the temperature measurements, which we, however, have not yet observed. In any case, such data sets are not used for the data analysis.





| Aircraft | Locations | IWC | RH_ice | N_ice |
|---|---|---|---|---|
| Geophysica | Seychelles[1], Europe[2], Brazil[3], Australia[4], Africa[5] | FISH[a] | FLASH[c] | FSSP-100[i] |
| Geophysica | Nepal[14] | FISH[a] | FLASH[c] | NIXE-CAPS[m] |
| Learjet | Europe[6] | FISH[a] | OJSTER[d], SEAL[e] | FSSP-300[j] |
| BAe-146 | UK[9] | NIXE-CAPS[m] | – | NIXE-CAPS[m] |
| HALO | Europe[10], Brazil[11] | NIXE-CAPS[m] | SHARC[h] | NIXE-CAPS[m] |
| Balloon | Europe[15] | B-ICI[n] | – | B-ICI[n] |
| WB-57 | USA[7,16], Costa Rica[8] | CLH[b] | HWV[f], JLH[g] | CAPS[k], 2D-S[l] |
| WB-57 | Guam[13] | NOAA-$H_2O$[o] | NOAA-$H_2O$[o] | FCDP[p], 2D-S[l] |
| Global Hawk | Guam[12] | NOAA-$H_2O$[o] | NOAA-$H_2O$[o] | FCDP[p], 2D-S[l] |
| GV HIAPER | USA[17], Guam[19] | CDP[q], 2D-C[r] | VCSEL[s:] | CDP[q], 2D-C[r] |
| Learjet | USA[18] | 2D-S[l] | – | FSSP[i], 2D-S[l] |

1: APE-THESEO 1999
2: ENVISAT 2002, EUPLEX 2003, ENVISAT 2003
3: TROCCINOX 2005
4: SCOUT-$O_3$ 2005
5: AMMA 2006
6: CIRRUS 2003, CIRRUS 2004, CIRRUS 2006,
   AIRTOSS-ICE 2013
7: MidCix 2004, MACPEX 2011
8: TC-4 2007
9: COALESC 2011
10: ML-CIRRUS 2014
11: ACRIDICON-CHUVA 2014
12: ATTREX 2014
13: POSIDON 2016
14: StratoClim 2017
15: LTU 2012-2015
16: CRYSTAL-FACE 2002
17: START 2008
18: SPARTICUS 2010
19: CONTRAST 2014

a: Lyman-α fluorescence hygrometer   (Schiller et al., 2008; Krämer et al., 2009; Meyer et al., 2015)
b: Tunable diode laser hygrometer   (Luebke et al., 2013)
c: Lyman-α fluorescence hygrometer   (Sitnikov et al., 2007)
d: Tunable diode laser hygrometer   (Krämer et al., 2009)
e: Tunable diode laser hygrometer   (Buchholz et al., 2013)
f: Lyman-α fluorescence hygrometer   (Rollins et al., 2014)
g: Tunable diode laser hygrometer   (May, 1998)
h: Tunable diode laser hygrometer   (Meyer et al., 2015)
i: Light scattering cloud probe   (Baumgardner et al., 2017)
j: Light scattering cloud probe   (Baumgardner et al., 2017)
k: Light scattering and optical imaging cloud probe   (Baumgardner et al., 2001)
l: Optical imaging cloud probe   (Lawson et al., 2006)
m: Light scattering and optical imaging cloud probe   (Meyer, 2012; Luebke et al., 2016)
n: Cloud particle Imager   (Wolf et al., 2018)
o: Tunable diode laser hygrometer   (Thornberry et al., 2017)
p: Light scattering cloud probe   (McFarquhar et al., 2007)
q: Light scattering cloud probe   (Baumgardner et al., 2017)
r: Optical imaging cloud probe   (Baumgardner et al., 2017)
s: Tunable diode laser hygrometer   (Zondlo et al., 2010)

The campaigns under 1-6 are described by Schiller et al. (2008), Krämer et al. (2009) and Finger et al. (2015), 7-8 by Luebke et al. (2013); Jensen et al. (2013), 9 Jones et al. (2012), 10 Voigt et al. (2017), 11 Wendisch et al. (2016), 12 Jensen et al. (2017), 13 https://espo.nasa.gov/posidon, 14 Stroh et al, ACP, in prep., 15 Wolf et al. (2018), 16 https://espo.nasa.gov/crystalface, 17 Pan et al. (2010), 18 Muhlbauer et al. (2014), 19 Pan et al. (2017) .

**Table 4.** Overview of campaigns and instruments. IWC: ice water content (uncertainties ∼20%), RH_ice: relative humidity over ice (uncertainties ∼10-15%), N_ice: ice crystal number concentration (uncertainties ∼10-100%); the sampling rate of all aircraft instruments is 1 Hz.





| Campaign | data sets[†] | IWC | $N_{ice}$ | $RH_{ice}$/$H_2O$ |
|---|---|---|---|---|
| **Arctic** | | | | |
| EUPLEX_2003 | SKL | y | y[⋆] | y |
| ENVISAT_2003_1 | SKL | y | y[⋆] | y |
| ENVISAT_2003_2 | SKL | y | y[⋆] | y |
| *LTU_2012-18* | | y | y[spe] | – |
| **Mid-latitude** | | | | |
| ENVISAT_2002_2 | SK | y | y[⋆] | y |
| CIRRUS_2003 | SKL | y | y[⋆] | y |
| CIRRUS_2004 | SKL | y | y[⋆] | y |
| CIRRUS_2006 | SKL | y | y[⋆] | y |
| *START_2008* | | y | y[spe] | y |
| MIDCIX_2004 | Lk | y | y>15 | – |
| *SPARTICUS_2010* | | y | y>15 | – |
| MACPEX_2011 | k | y | y>15 | y |
| COALESC_2011 | k | y | y[spe] | – |
| AIRTOSS_2013 | k | y | y>15 | y |
| ML-CIRRUS_2014 | k | y | y[spe] | y |
| **Tropics** | | | | |
| APE_THESEO_1999 | SKl | y | y[⋆] | y |
| CRYSTAL-FACE_2002 | L | y | – | – |
| TROCCINOX_2005 | SKL | y | y[⋆] | y |
| SCOUT_2005 | SKL | y | y[⋆] | y |
| TC-4_2007 | L | y | y>15 | – |
| *CONTRAST_2014* | | y | y[spe] | – |
| ACRIDICON_2014 | k | y | y[spe] | y |
| ATTREX_2014 | | y | y[spe] | y |
| POSIDON_2016 | | y | y[spe] | y |
| STRATOCLIM_2017 | | y | y[spe] | y |

**Table 5.** Cirrus Guide II data base for the IWC, $N_{ice}$ and $RH_{ice}$ climatologies. y: quality checked measurements; –: no/questionable measurements; an overview of each campaign is given in the Supplementary Material; y[spe]: $N_{ice}$ is corrected for single particle events (spe, see Section A2.2); y[⋆]: $N_{ice}$ from size limited detection range (3 - 30 µm → contains $\gtrsim$ 90% of the ice particles), not spe corrected; y>15: $N_{ice}$ from size limited detection range (> 15 µm → underestimating the full ice particle concentration), not included in the data analysis presented here; the campaigns written in *italics* are not contained in the occurrence frequencies. [†] S K L k: campaigns contained in the data sets of Schiller et al. (2008); Krämer et al. (2009); Luebke et al. (2013); Krämer et al. (2016) (for S&K see Figure 7).

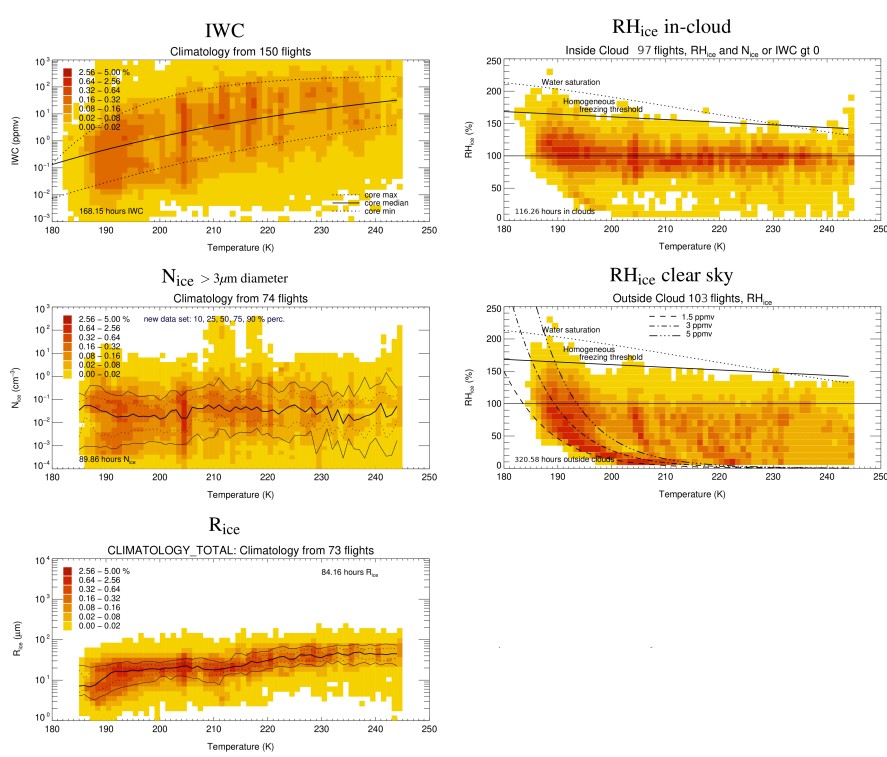

**Figure 16.** Data coverage of the Cirrus Guide II in situ data base: IWC, $N_{ice}$, $R_{ice}$, in-cloud and clear sky $RH_{ice}$ frequencies of occurrence in dependence on temperature.



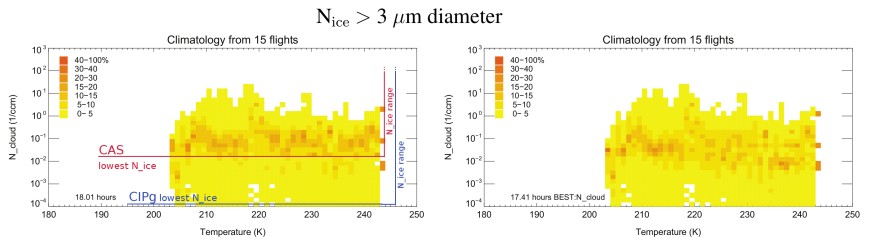

**Figure 17.** $N_{ice}$ frequencies of occurrence for ML-Cirrus 2014; left: original data set, right: corrected for 'single particle events'. A further example of an uncorrected data set can be seen in Figure 7, (middle left panel). For more details see text.

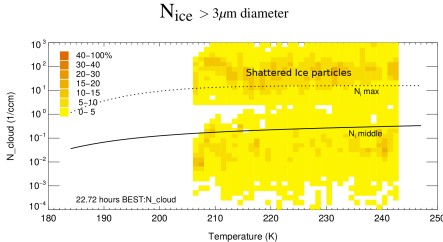

**Figure 18.** Example of $N_{ice}$ measurements biased by shattering of large ice particles, visible in the frequencies of occurrence: high frequencies appear at $N_{ice}$ concentrations between 10-100 $cm^{-3}$ for all temperatures, which are not present in undisturbed measurements (see Figure 17). The black lines denote the middle and maximum $N_{ice}$ lines from Krämer et al. (2009).

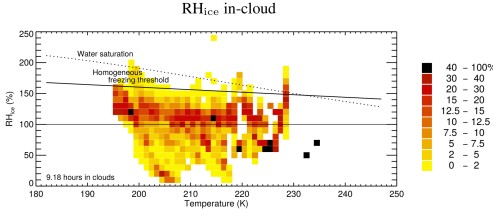

**Figure 19.** Example of biased $H_2O$ measurements, visible in the in-cloud $RH_{ice}$ frequencies: inside of cirrus clouds, at temperatures $\gtrsim$ 200 K, the frequencies should center around saturation (see Figure 6 upper right panel).