# Peer review of "A Microphysics Guide to Cirrus – Part II: Climatologies of Clouds and Humidity from Observations"

_Atmospheric Chemistry and Physics, 2020_

## Referee Comment (RC1) · Anonymous Referee #1 · 17 Apr 2020

This manuscript presents a comprehensive review of airborne in-situ and satellite remote sensing climatologies of cirrus clouds and water vapour. It combines previous as well as new insitu databases that help clarify detailed properties of tropical and mid-latitude cirrus and their responses and across a very important altitude range 5-20km. The links to satellite-borne data sets offers a benchmark for the model community to identify and begin to improve uncertainties in cirrus feedbacks.

The scale of the database and attention to detail is an impressive, particularly with respect to the review and update of in situ database quality control for known and ongoing issues with respect to small ice quantification due to artefacts in measurements. This is

particularly important in assessing error contributions to small ice concentrations providing confidence in the interpretation of different cirrus generation mechanisms currently being discussed, in-situ and liquid origin processes. These are well described although there is still much to be understood. The summary section on characteristics and distribution of in-situ origin and liquid origin cirrus linked to the previous Kramer et al. publication is very useful and helps to clarify the very large, sometimes overwhelming data sets. Despite some of the issues with previous measurements/data sets I found Figure 4 e.g. very encouraging showing a consistent relation between cirrus ice crystal concentration and mean ice mass radius and ice water content IWC.

The limitations of the data sets are discussed in good detail ins section A2.2 which is an important point for data users to be mindful of due to the impact on the uncertainties on concentrations/counting statistics and frequencies of occurrence of small cirrus ice crystals due to the very different sample volumes of the different instruments used in the analysis. Improving instrument response for small ice crystals still remains a challenge for in situ instruments especially for non-grey-scale imaging probes so merging data sets under varying environments needs to be treated with caution. The key issues are however explained - integration times dictated by different instrument sample volumes might limit detection of cirrus spatio-temporal inhomogeneities and hence interpretation of formation mechanisms. It was good to see the possible effects of this discussed and also those due t ice shattering (Figure 18 e.g.) which provides a useful benchmark for new data sets to compared with.

Figures 5 and 6 presented a nice overview of the data sets and how the different formation mechanisms contribute (minor typo in the Figure 5 legend, "summery" should be summary) versus clear sky conditions and as a function of region.

Minor typo Figure 7, plate Nice vs T, Nice units given as 1/ccm. This should be changed to cm-3 to be consistent with previous and subsequent figures.

The final results are perhaps not surprising and consistent with previous - i.e. "across

all latitudes, the thicker liquid origin cirrus predominate at lower altitudes, while at higher altitudes the thinner in-situ cirrus prevail." However, this study does provide a comprehensive database with estimates of radiative forcing ranges constrained by well described uncertainties.

Whilst this paper is extremely long and comprehensive it would have been useful to height potential uncertainties in some satellite retrievals, particularly with regard lack of sub-cloud top processes but likely this is not an issue for many of the cirrus discussed here.

All in all this is an excellent and very comprehensive review and analysis of our understanding of cirrus.

---

## Referee Comment (RC2) · Anonymous Referee #2 · 30 Apr 2020

This article reports findings of cirrus microphysical properties (IWC, Nice, Rice) and humidity from a climatology constructed from a large amount of airborne measurements, covering latitudes from 20S to 75N. This impressive data archive has been carefully quality checked and includes 150 flights from 24 campaigns. This effort is a huge contribution to our field. It is extremely difficult to build an unbiased climatology from airborne measurements as they often exist only for specific regions and at specific seasons. Results are presented as function of altitude and temperature, also stratified by tropics and NH midlatitudes (Sections 3 and 4). The supplement presents separate results for each of the 15 campaigns added to the ones of an earlier publication. Special attention has been given to the TTL of the Asian monsoon, using recent

measurements of the StratoClim campaign (Section 5). A second important part of this article consists of using these measurements to rescale ice crystal number concentration retrieved from global satellite radar-lidar observations, which then allows to compare Nice of different latitude bands (Section 6), in the case that this scaling factor of 1.73 is valid over the whole IWC-T range and over the whole globe.

This is a highly important article and should be published after revision. The abstract, introduction and conclusions are well written. It is really not an easy task to synthesize so much information. The present form of the article, though already quite well synthesized, is long, with a multitude of figures. The article could gain in clarity by taking into account the following suggestions. In particular, as already in Part I the cirrus clouds have been classified as in-situ and liquid origin and here are further distinguished according to updraft strength, a presentation of the in-cloud properties Nice, Rice and RHice in the IWC-T space, instead of only in the T space, would be very helpful.

Major comments:

Sections 3 and 4 both present results of the airborne climatology, with many figures. In general 6 variables, 3 corresponding to cirrus microphysical properties (IWC, Nice, Rice) and 3 corresponding to humidity (in-cloud RHice, clear sky RHice and clear sky water vapour mixing ratio) are presented stratified by altitude, latitude and temperature in these two sections. I would merge these sections into one section (3. In-situ climatologies), with for example subsections 3.1 Latitude – altitude distributions (including description of Figure 2), 3.2 In-cloud properties stratified by T and by (IWC, T), 3.3 In-situ and liquid origin cirrus. There are long descriptions, and as the title includes the word 'Guide', the behaviour of Nice, Rice and RHice in the IWC-T space, instead of or in addition to in the T space alone, would probably be clearer in respect to the cirrus classification. Therefore this new section 3.3 should show IWC as function of T (as already in Figs. 6-10), but then the other in-cloud properties as function of IWC and T (median or mean in IWC/T intervals and also variability in IWC/T intervals). The presentation in the (IWC-T) space has several advantages: 1) one can probably better

distinguish the different types of cirrus and their the properties Nice, Rice and RHice, leading to a more quantitative Table 1, and 2) this would be a very useful synthesis for testing parameterizations in climate models, as recent bulk ice cloud schemes rely on both of these parameters, IWC and T (see for example Field et al., 2007; Furtado et al., 2015; Baran et al. 2016 or Figs. 4f-h of Stubenrauch et al., 2019). Merging some of the figures as follows:

1) Figure 2 should only show 3 of the 4 sub-figures which links altitude and temperature for different latitude bands (one could imagine to separate summer and winter midlatitudes). What are the different colors within one latitude band? The right panel seems to be only an example from one field campaign (perhaps one could move this to section 5 where the TTL is discussed, or as it is published one can just resume the conclusion in the text).

2) Figure 3 presents scatter plots of these 6 variables deduced from all measurements, as function of altitude and latitude. The information could be presented in a more quantitative way by building intervals in altitude (for example per km) and per 10° in latitude and plot then the averages or medians in these intervals (and in addition the variability within the intervals in a separate plot), instead of superposing each of the measurement which indeed shows the scatter but also leads to confusion as some of the points are below others. It looks to me that the most rare measurement values are plotted above so that one can see them (in blue). The comments on the figures are very interesting, but the color blue should only be used if they correspond to the color of the variable value.

3) Then in Section 4, these 6 variables are shown as function of T in Figures 6 to 9. Take out Figure 7 (earlier results, already in supplement as Figure 1), and build two figures, 6 and 7: new Figure 6 could present IWC as fct of T, and Nice, Rice and RHice in IWC-T space, for all, NH midlatitudes and tropics, and new Figure 7 $H_2O$ clear sky as fct of T and perhaps RHice clear sky in $H_2O$-T space, for all, NH midlatitudes and tropics. One motivation of this analysis is certainly to verify that there

is a coherent relationship between the microphysical values and T, even if the tropics and midlatitudes include cirrus with a different range of these values. Therefore a joint discussion of all, NH midlatitudes and tropics will be easier to follow.

Section 3.1.1: As already in Part I, the authors classify cirrus according to their origin: in-situ or liquid origin; and they nicely summarize their characteristics by further distinguishing those meteorological situations with slow and fast updrafts. However, it is not clear to me from where the authors have the information on the updraft speed. Is this based on simultaneous measurements or on intuition? As this classification is one of the core findings, it is important to explain from where this information is obtained.

I would place Table 1 at the end of Section 3, so that the words 'low', 'few', 'large', etc. can be replaced by ranges (probably separately for tropics and midlatitudes). The authors show ranges in Figure 5, but these are only for midlatitudes, (probably at equinox conditions, as they were used for a specific simulation).

Section 6: Section 6.1 presents an evaluation of the remote sensing lidar-radar retrieval method, by comparing Nice in the T space from in-situ PSD measurements of 5 campaigns and Nice in the T space, where in a first step N0* and Dm, assuming a modified Gamma function, have been determined from these in-situ PSD data to use them as a constraint in the satellite retrieval (Figure 13). An interesting finding of the comparison between the two Nice results is that the Nice overestimation from satellite retrieval can be partly explained by the fact that the in-situ PSDs often do not contain ice crystals with D < 20 micron, while the retrieval assumes a PSD including all size bins (lines 750-751). This bias should be larger at low T. However, in Section 2.2 it is written that for T > -50°C (220 K), an overestimation in DARDAR Nice is due to the inability of the modified Gamma distribution to match the frequently bi-modal shape of measured PSDs (lines 149-151). This statement means that at warmer T there is also an overestimation, but for a different reason. Should this not be discussed when considering Figure 13? And should then not follow, that a different scaling factor applies for T < 220 K and for T > 220 K? Unfortunately the logarithmic scale and the squeezed

y axis do not permit to see if two instead of one scaling factor over the whole T range would be better. Again, I suggest to present Nice also in the IWC-T space, especially since IWC is also available from the DARDAR retrieval. Same for Figures 14 and 15. Then it could be directly seen that the thinnest Ci at cold T are not detected by DAR-DAR, and perhaps even that different scaling factors would apply in different (IWC-T) intervals. For the analysis in Figure 13, N0* and Dm ware determined from the in-situ data and used as constraint for the satellite retrieval, rather than being constrained from radar-lidar measurements during usual retrievals (line 751). I do not completely understand this sentence: does this mean that for the comparison and the following scaling, results of a special radar-lidar retrieval were used or is the global climatology, presented in sections 6.2 and 6.3, also based on the lidar-radar retrieval with this (N0*, Dm) constraint ? If the usual retrieval is different, then one should also show Nice for the usual retrieval. Perhaps this only needs clarification in the text.

Once Nice adjusted by a constant factor 1.73, Nice is decreasing with increasing T in Figure 14, while Nice is constant with T for in-situ measurements. Again, the presentation of Nice as fct of IWC and T and its variability within the IWC-T intervals will perhaps give additional insight, in particularly if one distinguishes tropics, midlatitudes and polar regions.

Section 7: lines 952-953 (conclusions from Figure 5): The authors should make it very clear that the analysis in Figure 5 only serves as an illustration how this data archive can be used to determine cloud radiative effects. The presented radiative transfer calculations have only been undertaken for a specific situation: at noon, equinox, at $50°$ latitude, only representative for midlatitude conditions at a specific daytime. This should be clearly written in the conclusions. It is mentioned as a kind of footnote in the legend of Figure 5, but can be easily overseen. Also, there have been many studies on cirrus radiative effects published before, it might be interesting to compare with earlier results (for example Kienast-Sjögren et al., 2016 or Campbell et al., 2016).

Will this data archive be made available? I did not find a section about data availability.

Minor comments:

Abstract, lines 22/ 23: half of the cirrus are located in the lowest warmest cirrus layer; what is the warmest cirrus layer?

Section 2, lines 120 – 124: the data of 4 campaigns out of 24 campaigns are not used, because the data volume is too low or too high. If the data volume is too high, one could imagine to filter out cases randomly. It is a pity that the data are not used at all. Or did I understand something wrong? Are all following results based on the 20 campaigns?

Section 2.2: It was found that the assumption of a modified Gamma distribution for the PSD is only valid for $T < -50°C$, which limits the statistics very much, and which leads to a positive bias of Nice at warmer T, because the assumed PSD shape is not coherent with observed bimodal PSDs. Are there other assumptions in the retrieval which may lead to biases?

Figure 13: It is not clear which satellite retrieval statistics is used: only the regions and seasons of the 5 campaigns? This needs some explanation in the text.

Figure 1: are the airplane schemes necessary? It is nearly impossible to read the name of the campaigns. As there are no flights in the SH higher latitudes, one could use this space to write these names there, which allows to increase the size of the map.

Section 5: Figure 10 is already in the supplement. As this section concentrates on the Asian monsoon, the data of Figure 10 can be analysed in the IWC-T space as proposed above.

Title of Section 6: Global cirrus Nice climatology from satellite remote sensing (the regional data are included in the global)

Table 3: could one add Nice for tropics and NH midlatitudes from in-situ measurements?

Typo, line 188: distribution of cirrus

Typo, line 409: which can be seen

Typo in Table 2: median for 190K-200K: 0.100 instead of 0.010

References mentioned in 1. Paragraph of major comments:

Field, P. R., Heymsfield, A. J., & Bansemer, A. (2007). Snow size distribution parameterization for midlatitude and tropical ice clouds. J. Atmos. Sci., 64, 4346–4365, doi:10.1175/2007JAS2344.1.

Furtado, K., Field, P. R., Cotton, R., & Baran, A. J. (2015). The sensitivity of simulated high clouds to ice crystal fall speed, shape and size distribution. Q. J. R. Meteorol. Soc., 141, 1546-1559, doi:10.1002/qj.2457.

Baran, A. J., Hill, P., Walters, D., Hardiman, S.C., Furtado, K., Field, P.R., & Manners, J. (2016). The Impact of Two Coupled Cirrus Microphysics–Radiation Parameterizations on the Temperature and Specific Humidity Biases in the Tropical Tropopause Layer in a Climate Model. J. Climate, 29, 5299–5316, doi: 10.1175/JCLI-D-15-0821.1.

Stubenrauch, C. J., Bonazzola, M., Protopapadaki, S. E., & Musat, I. (2019). New cloud system metrics to assess bulk ice cloud schemes in a GCM. J. Advanc. Model. Earth Systems, 11, 3212–3234. https://doi.org/10.1029/2019MS001642.

References mentioned in last paragraph of major comments:

Campbell, J.R., S. Lolli, J.R. Lewis, Y. Gu, and E.J. Welton, 2016: Daytime Cirrus Cloud Top-of-the-Atmosphere Radiative Forcing Properties at a Midlatitude Site and Their Global Consequences. J. Appl. Meteor. Climatol., 55, 1667–1679, https://doi.org/10.1175/JAMC-D-15-0217.1

Kienast-Sjögren, E., Rolf, C., Seifert, P., Krieger, U. K., Luo, B. P., Krämer, M., and Peter, T.: Climatological and radiative properties of midlatitude cirrus clouds derived by automatic evaluation of lidar measurements, Atmos. Chem. Phys., 16, 7605–7621,

https://doi.org/10.5194/acp-16-7605-2016, 2016.
* * *

---

## Author Comment (AC1) · 30 Jun 2020

Answers to the comments on acp-2020-40

**A Microphysics Guide to Cirrus – Part II:**
**Climatologies of Clouds and Humidity from  Observations**

by Krämer et al.

First of all, we like to thank the two referees a lot for their very positive rating of the manuscript and also for the constructive comments that further improved it. We are aware that the paper is very long and comprehensive, but could still have been extended into several directions. It was not easy to find a good balance in summarizing earlier findings and adding new data analysis, and we are glad to see that  the work that went into the  study seems  to have led to a satisfactory result.
In the following, we answer point by point  the comments of the referees (colored in black), our answers are blue.

**Anonymous Referee #1**

This manuscript presents a comprehensive review of airborne in-situ and satellite re mote sensing climatologies of cirrus clouds and water vapour. It combines previous as well as new insitu databases that help clarify detailed properties of tropical and mid- latitude cirrus and their responses and across a very important altitude range 5-20km. The links to satellite-borne data sets offers a benchmark for the model community to identify and begin to improve uncertainties in cirrus feedbacks → We are glad  to read hat, this is what we had hoped to be able  to provide.

The scale of the database and attention to detail is an impressive, particularly with respect to the review and update of in situ database quality control for known and ongoing issues with respect to small ice quantification due to artefacts in measurements. This is particularly important in assessing error contributions to small ice concentrations providing confidence in the interpretation of different cirrus generation mechanisms currently being discussed, in-situ and liquid origin processes. These are well described although there is still much to be understood. The summary section on characteristics and distribution of in-situ origin and liquid origin cirrus linked to the previous Kramer et al. publication is very useful and helps to clarify the very large, sometimes overwhelming data sets. Despite some of the issues with previous measurements/data sets I found Figure 4 e.g. very encouraging showing a consistent relation between cirrus ice crystal concentration and mean ice mass radius and ice water content IWC.

The limitations of the data sets are discussed in good detail ins section A2.2 which is an important point for data users to be mindful of due to the impact on the uncertainties on concentrations/counting statistics and frequencies of occurrence of small cirrus ice crystals due to the very different sample volumes of the different instruments used in the analysis. Improving instrument response for small ice crystals still remains a challenge for in situ instruments especially for non-grey-scale imaging probes so merging data sets under varying environments needs to be treated with caution. The key issues are however explained - integration times dictated by different instrument sample volumes might limit detection of cirrus spatio-temporal inhomogeneities and hence interpretation of formation mechanisms. It was good to see the possible effects of this discussed and also those due to ice shattering (Figure 18 e.g.) which provides a useful benchmark for new data sets to compared with.

Figures 5 and 6 presented a nice overview of the data sets and how the different formation mechanisms contribute (minor typo in the Figure 5 legend, "summery" should be summary:  →

corrected) versus clear sky conditions and as a function of region. Minor typo Figure 7, plate Nice vs T, Nice units given as 1/ccm. This should be changed to cm-3 to be consistent with previous and subsequent figures. Fig. 7 is copied from Krämer et al., 2009, so changes are not possible now; for information: the Figure is moved to the supplementary material in the revised version.

The final results are perhaps not surprising and consistent with previous - i.e. "across all latitudes, the thicker liquid origin cirrus predominate at lower altitudes, while at higher altitudes the thinner in-situ cirrus prevail." However, this study does provide a comprehensive database with estimates of radiative forcing ranges constrained by well described uncertainties.

Whilst this paper is extremely long and comprehensive it would have been useful to height potential uncertainties in some satellite retrievals, particularly with regard lack of sub-cloud top processes but likely this is not an issue for many of the cirrus discussed here.

All in all this is an excellent and very comprehensive review and analysis of our understanding of cirrus.     Thanks again !!

**Anonymous Referee #2**

This article reports findings of cirrus microphysical properties (IWC, Nice, Rice) and humidity from a climatology constructed from a large amount of airborne measurements, covering latitudes from 20S to 75N. This impressive data archive has been carefully quality checked and includes 150 flights from 24 campaigns. This effort is a huge contribution to our field. It is extremely difficult to build an unbiased climatology from airborne measurements as they often exist only for specific regions and at specific seasons. Results are presented as function of altitude and temperature, also stratified by tropics and NH midlatitudes (Sections 3 and 4). The supplement presents separate results for each of the 15 campaigns added to the ones of an earlier publication. Special attention has been given to the TTL of the Asian monsoon, using recent measurements of the StratoClim campaign (Section 5). A second important part of this article consists of using these measurements to rescale ice crystal number concentration retrieved from global satellite radar-lidar observations, which then allows to compare Nice of different latitude bands (Section 6), in the case that this scaling factor of 1.73 is valid over the whole IWC-T range and over the whole globe.

This is a highly important article and should be published after revision. The abstract, introduction and conclusions are well written. It is really not an easy task to synthesize so much information. The present form of the article, though already quite well synthesized, is long, with a multitude of figures. The article could gain in clarity by taking into account the following suggestions. In particular, as already in Part I the cirrus clouds have been classified as in-situ and liquid origin and here are further distinguished according to updraft strength, a presentation of the in-cloud properties Nice, Rice and RHice in the IWC-T space, instead of only in the T space, would be very helpful.

Thank you very much for this suggestion, we have added a respective new Figure (new Figure 6) and  a new Section (new Section 4.1) to the manuscript.  Such a representation of the data was not included in the first version of the paper, since we had already discussed the structure of Ni and Ri in the IWC-T space in Part 1 of the paper, but not as percentiles in IWC-T intervals. We were surprised  how informative this new new analysis is and think that it  might become one of the more noted result of the paper.

**Major comments:**

0) Sections 3 and 4 both present results of the airborne climatology, with many figures. In general 6 variables, 3 corresponding to cirrus microphysical properties (IWC, Nice, Rice) and 3 corresponding to humidity (in-cloud RHice, clear sky RHice and clear sky water vapour mixing ratio) are presented stratified by altitude, latitude and temperature in these two sections. I would merge these sections into one section (3. In-situ climatologies), with for example subsections 3.1 Latitude – altitude distributions (including description of Figure 2), 3.2 In-cloud properties stratified by T and by (IWC, T), 3.3 In-situ and liquid origin cirrus.
There are long descriptions, and as the tile includes the word 'Guide', the behaviour of Nice, Rice and RHice in the IWC-T space, instead of or in addition to in the T space alone, would probably be clearer in respect to the cirrus classification. Therefore this new section 3.3 should show IWC as function of T (as already in Figs. 6-10), but then the other in-cloud properties as function of IWC and T (median or mean in IWC/T intervals and also variability in IWC/T intervals). The presentation in the (IWC-T) space has several advantages:
1) one can probably better distinguish the different types of cirrus and their the properties Nice, Rice and RHice, leading to a more quantitative Table 1, and
2) this would be a very useful synthesis for testing parameterizations in climate models, as recent bulk ice cloud schemes rely on both of these parameters, IWC and T (see for example Field et al., 2007; Furtado et al., 2015; Baran et al. 2016 or Figs. 4f-h of Stubenrauch et al., 2019).

**Response to 0:**
We see and also like the idea behind the proposed structuring, namely to first consider entire climatologies of median values of the parameters (and their deviations)  in the latitude-altitude and the T-IWC parameter space (and maybe also the frequency distributions of the parametrs versus temperature) and then to split the climatologies into the two cirrus types, in-situ origin and liquid origin.

What would be lost in such a version of the article would be the discussion of the microphysical processes and the identification of the types based on the representation in Figure 3 (individual data points). If one calculates medians in intervals, characteristic differences that are important for this type of interpretation disappear.
  We also like to note here that the two different types of cirrus can not be distinguished looking at medians of Nice, Rice and Rhice in the T-IWC space (see new Figure 6, for convenience added at the end of this document) because of the overlap of the types in the T-IWC space. For such an analyses it would be necessary to first split the data set into in-situ-origin and liquid-origin cirrus and then produce the median T-IWC plots (as done by Luebke et al., 2016 and Wolf et al., 2018, 2019 for specific field campaigns). Such a sorting of the database is difficult (because the trajectory based method does not work in convective systems) and an ongoing work in our group. We will be happy to present such plots and also a more quantitative Table 1 (some quantification is given there, see answer to point 5) ) in a future publication.

The division of the paper into sections 3 and 4 followed the idea of first discussing the microphysical processes and characteristics of the cirrus types and then moving on to climatologies of the frequencies of occurrence, which can also be used to test parameterizations in climate models or satellite retrievals.

As a synthesis of our aproach and the new and very informative ideas of the referee, we kept Sections 3 and 4, but
- added   altitude-latitude plots of interval percentiles of the 6 variables (Section 3) as supplementary material (new Figures S1 – S3),

- added T-IWC plots of Nice, Rice, RHice   of interval percentiles (Section 4) as supplementary material (new Figure S4); we further added  T-IWC plots of DARDAR-Nice medians (Section 6) to the supplementary material (new Figure S5),

- added  a new Subsection to Section 4 (4.1: The IWC-T parameter space: median Nice , Rice , Rhice), where the  T-IWC plots of Nice, Rice, RHice interval medians are discussed (new Figure 6, see also at end of this document).

Merging some of the figures as follows.

1) Figure 2 should only show 3 of the 4 sub-figures which links altitude and temperature for different latitude bands (one could imagine to separate summer and winter midlatitudes). What are the different colors within one latitude band? The right panel seems to be only an example from one field campaign (perhaps one could move this to section 5 where the TTL is discussed, or as it is published one can just resume the conclusion in the text).

 **Response to 1:**
 - We have not splitted the  data set into winter and summer because to our feeling the in-situ data base is still not large enough to proviede a view on seasons. This Figure should give an impression on the temperature, Theta and altitude  ranges of the different geographical regions, so we decided to use the Figure published earlier.

 - The colors are:   blueish colors for Arctic, greenish colors for mid-latitude and reddish                        colors for  tropical observations,  which is mentioned in the Fgure caption.

 - We agree with the referee and removed the right panel.

2) Figure 3 presents scatter plots of these 6 variables deduced from all measurements, as function of altitude and latitude. The information could be presented in a more quantitative way by building intervals in altitude (for example per km) and per 10◦ in latitude and plot then the averages or medians in these intervals (and in addition the variability within the intervals in a separate plot), instead of superposing each of the measurement which indeed shows the scatter but also leads to confusion as some of the points are below others. It looks to me that the most rare measurement values are plotted above so that one can see them (in blue). The comments on the figures are very interesting, but the color blue should only be used if they correspond to the color of the variable value.

**Response to 2**: To better explain the way we have plotted the data we have added new text at the beginning of Section 3.1:
,The way the data is presented here as individual points was chosen because the  entire range of measurements is visible. Although data overlap occurs in this type of display, it is possible to identify cirrus types and microphysical processes, especially based on extreme values. As additional overview information, we have created latitude-altitude intervals (0.5◦ latitude, 500 m altitude) and calculated the 25, 50 (median) and 75 % percentiles for all

variables. These additional altitude-latitude climatologies are shown in the supplementary material.'

and also in Section 3.1.1:
'*Microphysical characteristics:* In the new in-situ data set, containing advanced measurements and extended by several field campaigns in comparison to the earlier studies, some typical characteristics of the cirrus types and hints to ice nucleation mechanisms are visible. In the following, the cirrus types are briefly introduced, and, using Figures 3 and 4, the types and freezing mechanisms are discussed and summarized in Table 1.'

3)  Then in Section 4, these 6 variables are shown as function of T in Figures 6 to 9.

Take out Figure 7 (earlier results, already in supplement as Figure 1), (done)

and build two figures, 6 and 7:
- new Figure 6 could present IWC as fct of T, and Nice, Rice and RHice in IWC-T space, for all (done, **see Response to 0** and new Figure 6, also at the end of this document),
- NH midlatitudes and tropics (we have produced these plots, but they look very similar to the plot for all data -except of the temperature range- **\*\*,** so we decided not to show them to not further lengthen the paper), and
- new Figure 7 H2O clear sky as fct of T and perhaps RHice clear sky in H2O-T space, for all, NH midlatitudes and tropics. (also for the sake of brevity, we have not shown such plots since they do not provide additional information)

One motivation of this analysis is certainly to verify that there is a coherent relationship between the microphysical values and T (this is discussed now in new Subsection 4.1),
even if the tropics and midlatitudes include cirrus with a different range of these values. Therefore a joint discussion of all, NH midlatitudes and tropics will be easier to follow (see comment **\*\*** above).

4)  Section 3.1.1: As already in Part I, the authors classify cirrus according to their origin: in-situ or liquid origin; and they nicely summarize their characteristics by further distinguishing those meteorological situations with slow and fast updrafts. However, it is not clear to me from where the authors have the information on the updraft speed. Is this based on simultaneous measurements or on intuition? As this classification is one of the core findings, it is important to explain from where this information is obtained.

**Response to 4**:
Good point, thanks for noticing this - sometimes one is routine-blinded as a author … we note now that the division in slow and fast updrafts is at ~10cm/s, based on simulations presented by Kärcher and Lohmann (2002) and Krämer et al. (2016).

5)  I would place Table 1 at the end of Section 3, so that the words 'low', 'few', 'large', etc. can be replaced by ranges (probably separately for tropics and midlatitudes). The authors show ranges in Figure 5, but these are only for midlatitudes, (probably at equinox conditions, as they were used for a specific simulation).

**Response to 5**:
Since we have kept the structure of the Sections, we have not moved Table 1. Please note that ranges of 'low', 'few', 'large', etc. are given in the text below the Table:

*slow updraft:* $\lesssim$ 10 cm/s; *fast updraft:* $\gtrsim$ 10 cm/s (Kärcher and Lohmann, 2002, Krämer et al., 2016).

IWC high/low: above/below the IWC median (see Figure 7).

$N_{ice}$ few/more/many: below/in-between/above the 10 and 90% $N_{ce}$ percentiles (see Figure 7).

$R_{ice}$ small/large/larger:

        ice particles $\lesssim$ 20μm dominate the PSD /

        ice particles $\gtrsim$ 20μm dominate the PSD, max. size several hundred μm diameter /

        ice particles $\gtrsim$ 20μm dominate the PSD, max. size up to thousand μm diameter,

        PSD: particle size distribution.

6) Section 6: Section 6.1 presents an evaluation of the remote sensing lidar-radar retrieval method, by comparing Nice in the T space from in-situ PSD measurements of 5 campaigns and Nice in the T space, where in a first step N0* and Dm, assuming a modified Gamma function, have been determined from these in-situ PSD data to use them as a constraint in the satellite retrieval (Figure 13). An interesting finding of the comparison between the two Nice results is that the Nice overestimation from satellite retrieval can be partly explained by the fact that the in-situ PSDs often do not contain ice crystals with D < 20 micron, while the retrieval assumes a PSD including all size bins (lines 750-751). This bias should be larger at low T. However, in Section 2.2 it is written that for T > -50∘C (220 K), an overestimation in DARDAR-Nice is due to the inability of the modified Gamma distribution to match the frequently bi-modal shape of measured PSDs (lines 149-151). This statement means that at warmer T there is also an overestimation, but for a different reason. Should this not be discussed when considering Figure 13? And should then not follow, that a different scaling factor applies for T < 220 K and for T > 220 K? Unfortunately the logarithmic scale and the squeezed y axis do not permit to see if two instead of one scaling factor over the whole T range would be better.

**Response to 6 start:**
This is a very good point raised by the reviewer, which refers to an attempt to correct inherent biases in statistical comparisons of Ni from satellite (DARDAR-Nice) and in situ observations. Learning from preliminary comparisons by Sourdeval et al (2018) between the satellite (DARDAR-Nice) and a subset of the current in-situ dataset (5 campaigns) led the authors to identify two main effects that can lead to possible biases.

The first is due to the assumption of a mono-modal PSD in the satellite algorithm, which becomes limiting when ice particle growth processes are dominant, and leads to an overestimation of Ni in DARDAR-Nice. However, the -50°C threshold announced by Sourdeval et al (2018) is very conservative. While bi-modality indeed appears from this temperature, the impact on Ni is not substantial. Their study even showed a reasonably good agreement of the satellite Ni and co-located in situ observations down to T = -30°C (see Fig. 4 of that paper). It is therefore reasonable to consider that DARDAR-Nice is capable of retrieving Ni with only small biases due to bi-modality of the PSD for the entire range of cirrus temperatures observed here, while keeping in mind that an overestimation of Ni is possible for certain cloud types.

The second effect that of the continuity of PSD bins, that is assumed by the satellite algorithm from 5 um and is not necessarily guaranteed in the in-situ data. This concerns mainly small ice crystals, which significantly contribute to Ni, and leads to a higher Ni in the satellite retrievals. The understanding of the physical basis for these missing or empty bins is a very interesting question, which will require further investigation. For the moment,

this manuscript proposes to correct this issue by using a scaling factor applied on the satellite Ni.

It can be noted that both effects are not expected to affect the same temperature ranges, as cold temperature would lead to a higher satellite Ni due to missing/empty PSD bins and warmer would lead to higher satellite Ni due to PSD bi-modality. This leads us to agree with the reviewer that a temperature dependent correction would be optimal. However, considering the large uncertainties on our understanding of these effects, and probably the occurrence of other issues that might lead to other biases (e.g. the PSD shape assumed in the retrievals), we decided to use the simplest option of a single scaling factor over the entire temperature. This choice appears very reasonable from Fig. 13, especially considering the large Ni variations around the overall median. This choice still appears to us as being the most reasonable and adapted for this paper, until more can be understood on these effects.

The text in section 2.2 has been clarified to account for this comment:
„Good agreements were found, except for temperatures higher than about -50°C where an overestimation in DARDAR-Nice due to the inability of the modified gamma distribution to match the frequently bi-modal shape of the measured PSDs."
replaced by
„Good agreements were found, although it was noted that the inability of the modified gamma distribution to match the frequently bi-modal shape of the measured PSDs could lead to an overestimation of Ni in DARDAR-Nice. This typically occurs at temperature above -50°C and is expected to be cloud-type dependent, but Sourdeval et al (2018) showed that Ni still was in reasonable agreement with the in-situ (a factor of 2) down to T = -30°C, which should cover the entire cirrus temperature ranges in this study."

6) ctd. 1: Again, I suggest to present Nice also in the IWC-T space, especially since IWC is also available from the DARDAR retrieval. Same for Figures 14 and 15. Then it could be directly seen that the thinnest Ci at cold T are not detected by DARDAR, and perhaps even that different scaling factors would apply in different (IWC-T) intervals.

**Response to 6 ctd. 1:**
The display of Ni in a IWC-T space is a very good suggestion, as it clearly allows for a better visibility. This approach is now used in Fig. 6 of the revised manuscript and an equivalent for DARDAR-Nice is here shown. Due to limited space, this figure has been included in the supplements of the revised manuscript (Fig. S5). The overall agreement between DARDAR-Nice and the Cirrus Guide II data set of the distribution of Ni bins in the IWC-T leads us to believe that the first order approach of a single scaling factor in the IWC-T interval should be reasonable for the means of the study.

This following paragraph is added to Section 6:
„Note that an analysis of Ni in a IWC-T space, similarly to Figure 6, is shown in Fig. 5 of the supplementary material. This figure shows a good climatological agreement between the satellite product and the Cirrus Guide II data set, with very similar distribution of Ni in the IWC-T space. Differences could be attributed to lack of statistics in Figure~6 (noisy patterns). However, one notable difference is the slope of the IWC-T relation, which appears much flatter in the satellite product than in the in-situ data, as indicated by the density isolines."

[Figure]

**New Figure S5:** Similarly to Figure 6 of the revised manuscript; Median Nice in intervals in the IWC-T parameter space for DARDAR-Nice. Densities of occurrence are indicated by plain isolines.

6) ctd. 2: For the analysis in Figure 13, N0* and Dm ware determined from the in-situ data and used as constraint for the satellite retrieval, rather than being constrained from radar-lidar measurements during usual retrievals (line 751). I do not completely understand this sentence: does this mean that for the comparison and the following scaling, results of a special radar-lidar retrieval were used or is the global climatology, presented in sections 6.2 and 6.3, also based on the lidar-radar retrieval with this (N0*, Dm) constraint ? If the usual retrieval is different, then one should also show Nice for the usual retrieval. Perhaps this only needs clarification in the text.

**Response to 6 ctd. 2:** We thank the reviewer for notice this lack of clarification in the text, and we agree that further explanations are required. There is no incompatibility between the analysis in section 6.1 and those using actual retrievals in section 6.2. Unfortunately there is almost no co-incident flights between CALIPSO/CloudSat and the in-situ flights, and so an indirect method had to be used to identify biases due to incompatibilities between DARDAR-ice and the in situ data. In section 6.1 we assume that the N0* and Dm retrievals are ``perfect'' in the sense that the lidar-radar measurements would have had sufficient information to perfectly retrieve the in-situ-measured N0* and Dm parameters. There is no further difference to the usual retrieval method. This approach allows to by-pass the measurement sensitivity issues and directly identify incompatibility issues in the satellite algorithm, i.e. here the limits of using a monomodal shape and the lack of missing/empty bins below 25 um. Sourdeval et al (2018) completed this theoretical analysis with actual co-incident flights from SPARTICUS (not included in the in-situ data) and showed that its conclusions still hold when looking at actual retrievals.

Line 751 was completed by the following sentence: „This approach allows to identify inherent incompatibilities between the satellite retrieval assumptions and the in-situ

measurements, by assuming that the in-situ PSD parameters are perfectly constrained by the lidar-radar. Therefore, possible differences should only be attributed to other retrieval assumptions, such as the PSD shape. Sourdeval et al. (2018) showed that this approach is efficient for identifying algorithmic limitations while still being representative of actual satellite retrievals."

7) Section 6: Once Nice adjusted by a constant factor 1.73, Nice is decreasing with increasing T in Figure 14, while Nice is constant with T for in-situ measurements. Again, the presentation of Nice as fct of IWC and T and its variability within the IWC-T intervals will perhaps give additional insight, in particularly if one distinguishes tropics, midlatitudes and polar regions.

**Response to 7:**
Nice from the satellite dataset actually decreases with increasing T even without the adjustment factor 1.73, but this indeed shows a possible issue in the way the satellite data is scaled. As the reviewer indicates, this factor most likely depends on the temperature at least, but also on the IWC and more generally on the ice cloud regime and nucleation type. The „correction" applied in this manuscript clearly is a first-order attempt to make satellite and in-situ data more compatible under direct comparisons for the means of this study, and is already giving good results in section 6. We completely agree that this correction is limited and should further expanded, but this would require many more analyses that would make the manuscript too dense. We nevertheless keep in mind this important comment by the reviewer for a future study, as understanding of these difference and the „bias" between satellite and in situ data is an important step towards improving our general understanding of ice cloud remote sensing or even processes (e.g. are the missing/empty bins a satellite or in-situ issue?).

The following sentences are added at the end of section 6.1: „A correction that depends on temperature, and possibly IWC, might therefore be optimal but a simpler first-order correction of 1.73 for all T and IWC range should here be sufficient for the needs of this study and considering the multitude of processes that can lead to this bias. Future studies will be required to precisely understand such inherent differences between satellite and in-situ dataset."

On the other hand, the discrepancy between the DARDAR-Nice and in-situ Nice median can be caused by the underlying flight strategies of the in-situ measurements. In Section 6.2, we wrote:
‚We attribute the slightly increasing median Nice with decreasing temperature to homogeneous ice nucleation events, because homogeneous ice nucleation rates increase with decreasing temperature, but their appearance in space and time is transient, as discussed in Section 4.2.2. Thus, such events are difficult to find by research aircraft. Hence, these events are likely underrepresented in the aircraft observations.'

8) Section 7: lines 952-953 (conclusions from Figure 5): The authors should make it very clear that the analysis in Figure 5 only serves as an illustration how this data archive can be used to determine cloud radiative effects. The presented radiative transfer calculations have only been undertaken for a specific situation: at noon, equinox, at 50◦ latitude, only representative for midlatitude conditions at a specific daytime. This should be clearly written in the conclusions.    It is mentioned as a kind of footnote in the legend of Figure 5, but can be easily overseen.

**Response to 8 start:**
In the conclusions, we changed the sentence
'Finally, a first estimate of the radiative characteristics of typical idealized in-situ and liquid origin cirrus scenarios is given (Figure 5)' to
'Finally, a first estimate of the radiative characteristics of typical, specific idealized in-situ and liquid origin cirrus scenarios is given (Figure 5)'.
This is also discussed in the main text in Section 3.1.1., new page 12, lines 352-355.

Also, there have been many studies on cirrus radiative effects published before, it might be interesting to compare with earlier results (for example Kienast-Sjögren et al., 2016 or Campbell et al., 2016).

**Response to 8 ctd.:**
We added a comparison with Kienast-Sjögren et al., 2016 and Campbell et al., 2016, the new paragraph (page 12) reads now:
„ ..., the slow 'in-situ origin' cirrus have only small optical depth (τ ) between 0.001 - 0.05, resulting in a slight net warming effect of not larger than about 1.5 W/m2. The optical depth of fast 'in-situ origin' cirrus is larger (τ : 0.05 - 1), but most of them are also warming (2-10 W/m2). The thickest fast-updraft 'in-situ origin' cirrus at the lowest altitudes change the sign of their net forcing, they switch to a slight cooling effect. The reason is the warmer temperature at lower altitude that reduces the warming effect of the longwave infrared radiation. The results of the radiative forcing calculations for the slow and fast updraft cirrus are in agreement with investigations from lidar observations reported by Kienast-Sjögren et al. (2016) and Campbell et al. (2016), who observed cirrus with optical depth up to 1 and 3, respectively, and found a decreasing warming effect with decreasing optical depth. Campbell et al. (2016) even reported a slight cooling effect at the warmest observed cirrus. The 'liquid origin' cirrus, however, mostly found in the warmest cirrus layers, have large optical depths (τ : 1 - 12), which is larger than the range of cirrus optical depth reported in many studies (the maximum optical depth is often found to be 1-3, e.g. Sassen et al., 2008; Kienast-Sjögren et al., 2016; Campbell et al., 2016; Mitchell et al., 2018, ; note that this is likely because the lidar technique, often use to investiagate cirrus cloud optical properties, has restrictions in the detection of thicker ice clouds). A consequence of the large optical thickness is a quite strong net cooling effect (- 15 to -250 W/m2 ) of 'liquid origin' cirrus. These values are of the same order of magnitude as reported from direct measurements inside of cirrus clouds (Wendisch et al., 2007; Joos, 2019)."

9) Will this data archive be made available? I did not find a section about data availability.

**Response to 9:**
Yes, after acceptance of the paper, in the final version of the manuscript, a link will be included where the data can be downloaded.

**Minor comments:**

- Abstract, lines 22/ 23: half of the cirrus are located in the lowest warmest cirrus layer; what is the warmest cirrus layer?

We added the temperature range (224-242 K).

- Section 2, lines 120 – 124: the data of 4 campaigns out of 24 campaigns are not used, because the data volume is too low or too high. If the data volume is too high, one could imagine to filter out cases randomly. It is a pity that the data are not used at all. Or did I understand something wrong? Are all following results based on the 20 campaigns?

Yes, the following results are based on the 20 field campaigns. The reason not to include the two larger data sets was that during theses campaigns mainly similar meteorological situations with very thick liquid origin cirrus were sampled, which would have biased the statistical analysis. To include only some filghts would have been an option to include some of the data – we are sorry that we have not thought about that. On the other hand, we believe that this would not have changed the overall picture that we presented.

- Section 2.2: It was found that the assumption of a modified Gamma distribution for the PSD is only valid for T < -50∘C, which limits the statistics very much, and which leads to a positive bias of Nice at warmer T, because the assumed PSD shape is not coherent with observed bimodal PSDs. Are there other assumptions in the retrieval which may lead to biases?

There are indeed other PSD assumptions that might lead to biases between the satellite and in-situ dataset. Notably, the choice two fixed parameters was until now not discussed. We have included the following paragraph in Section 6.1:

„Other assumptions on the PSD shape by the satellite remote sensing method might also contribute to this bias. The PSD shape indeed is provided by 4 parameters, 2 of which are fixed and 2 are retrieved (see Section 2.2). Delanoe et al. (2014) showed that the two fixed PSD parameters defined by Delanoe et al. (2005) and used in DARDAR-Nice might lead to a too steep representation of the small ice mode (i.e. too high Nice) and should be updated in future algorithm versions. Also, the bi-modality of the PSD towards temperature where growth processes become important is not accounted for and usually leads to small positive Nice biases (Sourdeval et al. 2018). The cause of these assumptions are difficult to account for, as they most likely depend on the cloud-type and on the thermodynamical environment, but they should on a first order be reasonably captured by the 1.73 adjustment factor."

- Figure 13: It is not clear which satellite retrieval statistics is used: only the regions and seasons of the 5 campaigns? This needs some explanation in the text.

As indicated in the figure caption and in the second paragraph of section 6.1, only the five mentioned campaigns are used in Figure 13. As shown in Figure 1, these campaigns are representative of a tropical and mid-latitude ice cloud. The following sentence was included in Section 6.1 for better clarity:

„This section therefore focuses on these five campaigns, which are nevertheless representative of a wide range of mid-latitude and tropical ice clouds (see Figure 1 and caption of Figure 13)."

- Figure 1: are the airplane schemes necessary? It is nearly impossible to read the name of the campaigns. As there are no flights in the SH higher latitudes, one could use this space to write these names there, which allows to increase the size of the map.

The airplane pictures are not essential, however, we think it gives an of impression on the experimental work to operate cloud and water vapor instruments on so many different platforms.

We have enlarged the Figure to the maximum size and hope that zooming in might help to read the campaign names.

- Section 5: Figure 10 is already in the supplement. As this section concentrates on the Asian monsoon, the data of Figure 10 can be analysed in the IWC-T space as proposed above.

In Section 5, the special features of occurence frequencies appearing in Figure 10 (Asian Monsoon) in comparison to the entire tropical climatology are discussed. We repeated this Figure in the Supplementary Material for the sake of completeness, so that one do not need to flip back and forth between main paper and supplement when looking at single campaigns. Thus, think that it is useful to keep the Figure as is.

- Title of Section 6: Global cirrus Nice climatology from satellite remote sensing (the regional data are included in the global)

Changed.

- Table 3: could one add Nice for tropics and NH midlatitudes from in-situ measurements?

The median Nice in this Table are calculated for the entire spatial and temporal Nice-T parameter space shown in Figure 15. This is possible for remote sensing measuremenst which are able to scan the entire space. In-situ observations provide only a snapshot of the statistical Nice distribution, thus medians from the entire Nice-T are not representative (see also text on page 29, lines 978 ff).

- Typo, line 188: distribution of cirrus

Changed.

- Typo, line 409: which can be seen

Changed.

- Typo in Table 2: median for 190K-200K: 0.100 instead of 0.010

Changed.

References mentioned in 1. Paragraph of major comments:

Field, P. R., Heymsfield, A. J., & Bansemer, A. (2007). Snow size distribution parameterization for midlatitude and tropical ice clouds. J. Atmos. Sci., 64, 4346–4365, doi:10.1175/2007JAS2344.1.

Furtado, K., Field, P. R., Cotton, R., & Baran, A. J. (2015). The sensitivity of simulated high clouds to ice crystal fall speed, shape and size distribution. Q. J. R. Meteorol. Soc., 141, 1546-1559, doi:10.1002/qj.2457.

Baran, A. J., Hill, P., Walters, D., Hardiman, S.C., Furtado, K., Field, P.R., & Manners, J. (2016). The Impact of Two Coupled Cirrus Microphysics–Radiation Parameterizations on the Temperature and Specific Humidity Biases in the Tropical Tropopause Layer in a Climate Model. J. Climate, 29, 5299–5316, doi: 10.1175/JCLI-D-15-0821.1. Stubenrauch, C. J., Bonazzola, M., Protopapadaki, S.

E., & Musat, I. (2019). New cloud system metrics to assess bulk ice cloud schemes in a GCM. J. Advanc. Model. Earth Systems, 11, 3212–3234. https://doi.org/10.1029/2019MS001642.

References mentioned in last paragraph of major comments:

Campbell, J.R., S. Lolli, J.R. Lewis, Y. Gu, and E.J. Welton, 2016: Daytime Cirrus Cloud Top-of-the-Atmosphere Radiative Forcing Properties at a Midlatitude Site and Their Global Consequences. J. Appl. Meteor. Climatol., 55, 1667–1679, https://doi.org/10.1175/JAMC-D-15-0217.1

Kienast-Sjögren, E., Rolf, C., Seifert, P., Krieger, U. K., Luo, B. P., Krämer, M., and Peter, T.: Climatological and radiative properties of midlatitude cirrus clouds derived by automatic evaluation of lidar measurements, Atmos. Chem. Phys., 16, 7605–7621.

**New Figure 6:**

[Figure]

[Figure]

[Figure]

---

## Referee Report (RR1)

**Response to authors reply and revised version of ACP 2020-40 'A Microphysics Guide to Cirrus – Part II: Climatologies of Clouds and Humidity from Observations' by M. Krämer et al.**

The authors have sufficiently replied to the comments and improved the manuscript accordingly. Therefore I recommend publication after minor revision.

**Minor comments:**

Section 2.2 reads already better than the first version, still I have a question concerning lines 155-160: *Good agreements were found, although it was noted that the inability of the modified gamma distribution to match the frequently bi-modal shape of the measured PSDs could lead to an overestimation of Ni in DARDAR-Nice. This typically occurs at temperature above -50°C and is expected to be cloud-type dependent, but Sourdeval et al (2018) showed that Ni still was in reasonable agreement with the in situ (**within** a factor of 2) down to T = -30°C, which should cover the entire cirrus temperature ranges in this study.*

This looks like the agreements are within a factor of 2 for T > -50°C and in better agreement for T < -50°C. Is this true ? How much better would this be?

line 338, 339: 'restrictions in the detection of thicker ice clouds': lidar has no problems to detect thick clouds, τ can't be determined because of saturation; suggestion: 'restrictions in the τ **determination** of thicker ice clouds'

Section 6.2, lines 916- 918: *The data was collected twice a day, approximately at midday and at midnight (satellite equator-crossing time is 1.30 am/pm), from June 2006 to December 2016.* DARDAR-Nice uses combined CALIPSO-CloudSat, but since April 2011, CloudSat data are not available anymore at 1.30 am LT; since May 2012 CloudSat data are only available during daytime. So are the results presented in section 6.1 a mixture of 1.30 am/pm from 2006-2011 and of 1.30 pm from 2012-2016 ? Is there a day – night difference? Or is DARDAR-Nice using CALIPSO alone (with better sensitivity to thin cirrus during night)?

**Typos**

line 35: inside **of** the clouds : take out 'of'

line 155: measurement**s**: add 's'

lines 157 & 158: $N_{ice}$ instead of Ni

line 169: tropic**s**: add 's'

line 243: 'as anvils' instead of 'and anvils' ?

line 344: likely **to** warm: add 'to'

line 400: pattern**s**: add 's'

line 451: within **of** cirrus: take out 'of'

line 558: tops of **to** massive: take out 'to'

line 608: In addition, deep convection with fast updrafts that occasionally overshoot into the TTL. **Not a complete sentence**

line 963: Table **?? :** should be replaced by Tables **2 and 3**

lines 964-965: by Bacer et al., 2018; Penner et al., 2018; Righi et al., 2019).

should be replaced by: by Bacer et al. (2018), Penner et al. (2018) and Righi et al. (2019).

---

## Author Response (AR2)

Response to second review of acp-2020-40

**A Microphysics Guide to Cirrus – Part II:**
**Climatologies of Clouds and Humidity from Observations**

by Krämer et al.

The authors have sufficiently replied to the comments and improved the manuscript accordingly. Therefore I recommend publication after minor revision.

We would like to thank the referee again for the very careful review and suggestions that have improved the manuscript. We have hopefully addressed the remaining points in a satisfactory manner (in blue).

**Minor comments:**

- Section 2.2 reads already better than the first version, still I have a question concerning lines 155-160: *Good agreements were found, although it was noted that the inability of the modified gamma distribution to match the frequently bi-modal shape of the measured PSDs could lead to an overestimation of Ni in DARDAR-Nice. This typically occurs at temperature above -50°C and is expected to be cloud-type dependent, but Sourdeval et al (2018) showed that Ni still was in reasonable agreement with the in situ (**within** a factor of 2) down to T = -30°C, which should cover the entire cirrus temperature ranges in this study.*

This looks like the agreements are within a factor of 2 for T > -50°C and in better agreement for T < -50°C. Is this true ? How much better would this be?

Yes, this was the initial meaning of this paragraph, but it is true that the agreement does not necessarily significantly improve with decreasing temperatures (other issues can appear, light the "missing/empty bins" discussed later in the manuscript). Overall, based on Sourdeval et al (2018), we think that a factor of 2 would be the most reasonable minimum accuracy to provide the readers here for such intercomparisons, especially considering that in situ measurements also are not perfect. This paragraph have been edited to reflect this clarification:
*"This typically occurs at temperature above -50°C and is expected to be cloud-type dependent. Nevertheless, Sourdeval et al (2018) showed that the satellite Ni remains in reasonable agreement with the in situ (within a factor of 2) from T = -30°C down to -90°C, which should cover the entire cirrus temperature ranges in this study."*

- line 338, 339: 'restrictions in the detection of thicker ice clouds': lidar has no problems to detect thick clouds, it can't be determined because of saturation; suggestion: 'restrictions in the tau **determination** of thicker ice clouds'

Thanks for pointing this out; we agree with this comment. This sentence has been edited accordingly.

- Section 6.2, lines 916- 918: *The data was collected twice a day, approximately at midday and at midnight (satellite equator-crossing time is 1.30 am/pm), from June 2006 to December*

*2016.* DARDAR-Nice uses combined CALIPSO-CloudSat, but since April 2011, CloudSat data are not available anymore at 1.30 am LT; since May 2012 CloudSat data are only available during daytime. So are the results presented in section 6.1 a mixture of 1.30 am/pm from 2006-2011 and of 1.30 pm from 2012-2016 ? Is there a day – night difference? Or is DARDAR-Nice using CALIPSO alone (with better sensitivity to thin cirrus during night)?

The referee is correct in that the results presented here correspond to a mixture of day- and night-time retrievals during 2006-2011 and day-time only later on. DARDAR-Nice only provides retrievals if both the lidar and the radar are available in order to provide retrievals of consistent quality (even if Lidar-only retrievals could indeed be possible during night- time). This is now indicated in the revised manuscript.

Diurnal differences in DARDAR-Nice retrievals (or generally in Ni) are a very interesting question that is however difficult to tackle using a Lidar-radar dataset. This is because one first needs to disentangle the different sensitivity to Ni of the lidar / radar during day- an night-time (that can lead to differences in retrievals) from the real changes in Ni. This problem would definitely be out of the scope of the current manuscript, but has for instance been addressed recently by Wall et al (2020) that studies the lifetime of convective ice clouds using DARDAR-Nice. Differences between the night- and day-time Ni were found, but this study only focused on deep convective clouds in a narrow regions. Also, there is no specific indication on the time of the day in situ measurements were obtained and so it seems reasonable to mix day- and night-time satellite retrievals as well.

*"The global frequency distribution of Nice from ten years of satellite observations is shown in Figure 14. The data was collected twice a day, approximately at midday and at midnight (satellite equator-crossing time is 1.30 am/pm), from June 2006 to December 2016. It should still be noted that satellite retrievals are not continuously available throughout this period due to instrumental limitations and night-time retrievals are only available until 2011. This does not impact the conclusions presented here as diurnal cycles in Ni are not considered this study. Such variations are also difficult to quantify lidar/radar products, although a diurnal signature has been investigated in the DARDAR-Nice Ni data for tropical anvils (Wall et al., 2020). Overall, the global satellite data set consists of nearly $2 \times 10^{10}$ Nice retrievals. The color code represent frequencies of occurrence, the black contours the 25, 50 and 75th 920 percentiles."*

Wall, C., Norris, J. R., Gasparini, B., Smith, W. R., Thieman, M. M., Sourdeval, O.: Observational Evidence that Radiative Heating Modifies the Life Cycle of Tropical Anvil Clouds, J. Clim., pp. 1–68, doi:10.1175/JCLI-D-20-0204.1, 2020.

**Typos**

line 35: inside **of** the clouds : take out 'of'        Done

line 155: measurements: add 's'        Done

lines 157 & 158: $N_{ice}$ instead of Ni        Done

line 169: tropics: add 's'        Done

line 243: 'as anvils' instead of 'and anvils' ?
The sentence was changed to:  ,… in jet streams, mesoscale convective systems and anvils.'

line 344: likely **to** warm: add 'to'            Done

line 400: pattern**s**: add 's'                    Done

line 451: within **of** cirrus: take out 'of'        Done

line 558: tops of **to** massive: take out 'to'      Done

[revised manuscript text omitted]

---

## Author Response (AR3)

Answers to the editor comments on acp-2020-40

**A Microphysics Guide to Cirrus – Part II:**
**Climatologies of Clouds and Humidity from  Observations**

by Krämer et al.

Many thanks for your further revisions and for submitting the revised version. I am happy to accept your paper for publication in ACP, however, I recommend that you do some further technical corrections in order to facilitate the typesetting process.

Also many thanks for the smooth and quick review process of the long manuscript and also your helpful comments.

1) I found a few language issues in the new paragraph:

L922: "are not considered IN this study"                    Done

L923: "to quantify IN/FROM lidar/radar products"            Done

L925: "The color code represent" --> represents            Done

2) Of course the general language level is fine, but there are several places where a native speaker should be able to make things even more elegant. Just an example, in L35 "inside of clouds": why not just "in clouds"? You have several native speaker co-authors - would one of them be ready to do some language corrections? My impression is that this would be worth the effort, given the outstanding dataset and science you present in your paper.

One of the native speaking co-authors has revised the manuscript for language and also made some more small changes and technical corrections. I hope the paper is now in a  shape that it can be published.

3) Please carefully check your list of references, for instance for Dinh et al., there is a 2016 version in ACP (instead of the 2015 ACPD); some journal names are written with capital letters (anyway, always use abbreviations); some journal names are just given as "JGR" (please correct); and some references are incomplete (e.g., Righi et al.).

Thanks for pointing this out, I have checked the list and corrected a number of references.

[revised manuscript text omitted]